# Learn to Guide Your Diffusion Model

**Alexandre Galashov**
Google DeepMind
agalashov@google.com

**Ashwini Pokle**
Google
apokle@google.com

**Arnaud Doucet**
Google DeepMind
arnauddoucet@google.com

**Arthur Gretton**
Google DeepMind
gretton@google.com

**Mauricio Delbracio**
Google
mdelbra@google.com

**Valentin De Bortoli**
Google DeepMind
vdebortoli@google.com

## Abstract

Classifier-free guidance (CFG) is a widely used technique for improving the perceptual quality of samples from conditional diffusion models. It operates by linearly combining conditional and unconditional score estimates using a *guidance weight* $\omega$. While a large, static weight can markedly improve visual results, this often comes at the cost of poorer distributional alignment. In order to better approximate the target conditional distribution, we instead learn *guidance weights* $\omega_{c,(s,t)}$, which are continuous functions of the conditioning $c$, the time $t$ from which we denoise, and the time $s$ towards which we denoise. We achieve this by minimizing the distributional mismatch between noised samples from the true conditional distribution and samples from the guided diffusion process. We extend our framework to reward guided sampling, enabling the model to target distributions tilted by a reward function $R(x_0, c)$, defined on clean data and a conditioning $c$. We demonstrate the effectiveness of our methodology on low-dimensional toy examples and high-dimensional image settings, where we observe improvements in Fréchet inception distance (FID) for image generation. In text-to-image applications, we observe that employing a reward function given by the CLIP score leads to guidance weights that improve image-prompt alignment.

## 1 Introduction

Diffusion models (Sohl-Dickstein et al., 2015; Ho et al., 2020; Song et al., 2021b; Song and Ermon, 2019) produce high-quality synthetic data in areas such as images (Saharia et al., 2022; BlackForest-Labs, 2025), videos (Google, 2025), and proteins (Watson et al., 2023; Abramson et al., 2024). These models proceed by gradually adding Gaussian noise through a diffusion process, transforming the data distribution into a Gaussian distribution. The generative model is obtained by approximating the time-reversal of this noising process. Practically, this relies on a learned denoiser network which is obtained by minimizing a regression objective. A similar strategy can, in principle, be developed to sample from a conditional distribution by learning a conditional denoiser network.

Recently, Classifier-Free Guidance (CFG) (Ho and Salimans, 2022) has become a popular choice to address the task of conditional sampling. CFG modifies the generating process by incorporating a *guidance* term, calculated as the difference between conditional and unconditional denoiser networks. This difference is scaled by a *guidance weight*, $\omega$, such that $\omega = -1$ recovers unconditional generation whereas $\omega = 0$ corresponds to the conditional one. Empirically, it was shown that CFG can drastically improve performance of diffusion models compared to conditional generation (which only uses the conditional denoiser), especially for very high guidance weights (Saharia et al., 2022; Rombach et al., 2022a). (Chidambaram et al., 2024; Wu et al., 2024) argued that such a regime pushes the samples to the edge of support such that they become easy to classify.

Classifier-free guidance was initially justified as a method for sampling from a modified distribution $p_\omega(x_0|c) \propto p(x_0)^{-\omega} p(x_0|c)^{1+\omega} \propto p(x_0)p(c|x_0)^{\omega+1}$, which amplifies the conditioning term (Ho and Salimans, 2022). However, subsequent work has established that the standard CFG sampling procedure does not, in fact, produce samples from this target distribution (Du et al., 2023; Koulischer

et al., 2025a). Instead, CFG sampling shifts the generated samples towards the modes of the original conditional $p(x_0|c)$. While this discrepancy can be corrected with computationally intensive methods like Sequential Monte Carlo (Skreta et al., 2025; He et al., 2025) or Markov Chain Monte Carlo (Du et al., 2023; Moufad et al., 2025; Zhang et al., 2025), these approaches are often impractical for large-scale applications. More importantly, naive CFG *already* yields a significant quality boost and therefore such correction might not be required.

Many works, see e.g. Ho and Salimans (2022); Kynkäänniemi et al. (2024); Wang et al. (2024); Kim et al. (2025), report improved FID scores when using some level of guidance ($\omega > 0$). This observation is at odds with the original motivation of CFG. Given that the learned denoiser $\hat{x}_\theta(x_t, c)$ is only an approximation of the true denoiser $\mathbb{E}[x_0|x_t, c]$, this suggests that CFG acts as a *correction to this approximation* which leads to a better modeling of the target conditional distribution $p(x_0|c)$.

Based on this observation, we propose to *learn* a guidance weight $\omega$ in order to better approximate the conditional distribution $p(x_0|c)$. We generalize the CFG method and allow the guidance weight to be both conditioning and time dependent, i.e., $\omega_{c,(s,t)}$. This weight is used to guide the denoising process from time $t$ to time $s$. We learn $\omega_{c,(s,t)}$ by matching the distribution of samples from the true diffusion process and the distribution of samples of the guided one. Empirically, we show that our method allows us to learn $\omega_{c,(s,t)}$ that better approximates the target distribution compared to the unguided model. For image generation, this leads to a consistently lower FID (Heusel et al., 2017) than the unguided model or a model with a constant guidance weight.

We further develop our method in a setting where an additional reward function $R(x_0, c)$ is defined on samples from the diffusion model and a conditioning $c$, and we want to bias the samples from the model to the regions of high rewards $R(x_0, c)$. In text-to-image applications, we found that empowering our method with a reward function given by the CLIP score (Hessel et al., 2021), leads to guidance weights which improve both FID and image-prompt alignment.

The paper is organized as follows. In Section 2, we provide a short background on the necessary concepts. In Section 3, we describe our approach of learning guidance weights $\omega_{s,(s,t)}$. In Section 4, we present the related work on the topic. Finally, the experimental results are presented in Section 5.

## 2 BACKGROUND

**Notation.** We write $p_t$ for a probability distribution $p(x_t)$, and $p_{s|t}$ to denote a conditional distribution $p(x_s|x_t)$ for any $s, t$. We also use $p_{0|c}$ to denote $p(x_0|c)$, $p_{s|t,0}$ for $p(x_s|x_t, x_0)$, and $p_{s|t,c}$ for $p(x_s|x_t, c)$. We denote by $p_{0,c}$ a joint distribution $p(x_0, c)$. Finally, we write $\mathcal{N}(x; \mu, \Sigma)$ to denote the Gaussian density of argument $x$, mean $\mu$ and covariance $\Sigma$ and $\mathcal{N}(\mu, \Sigma)$ for the distribution.

**Diffusion models.** The goal of conditional diffusion models is to sample from a target conditional distribution $p_{0|c}$ on $\mathbb{R}^d$, where $c \sim p(c)$ is a conditioning signal (i.e. a text prompt) sampled from a conditioning distribution. We adopt here the Denoising Diffusion Implicit Models (DDIM) framework of Song et al. (2021a). Let $x_{t_0} \sim p_{0|c}$ and define for $0 = t_0 < \cdots < t_N = 1$ the process $x_{t_1:t_N} := (x_{t_1}, ..., x_{t_N})$ by

$$p(x_{t_1:t_N}|x_{t_0}) = p_{t_N|t_0}(x_{t_N}|x_{t_0}) \prod_{k=1}^{N-1} p_{t_k|t_{k+1},0}(x_{t_k}|x_{t_{k+1}}, x_{t_0}), \tag{1}$$

where, for $0 \leq s < t \leq 1$, we have

$$p_{s|t,0}(x_s|x_t, x_0) = \mathcal{N}(x_s; \mu_{s,t}(x_0, x_t), \Sigma_{s,t}). \tag{2}$$

Both the mean $\mu_{s,t}(x_0, x_t)$ and covariance $\Sigma_{s,t}$ depend on a *churn* parameter $\varepsilon \in [0, 1]$, which controls the amount of stochasticity. Their full expressions are given in (23) in Appendix B. Here, the mean and variance are selected so that for any $t \in [0, 1]$, we recover the *noising process*

$$p_{t|0}(x_t|x_0) = \mathcal{N}(x_t; \alpha_t x_0, \sigma_t^2 \mathrm{Id}), \tag{3}$$

for $\alpha_t$ and $\sigma_t$ such that $\alpha_0 = 1, \sigma_0 = 0$ and $\alpha_1 = 0, \sigma_1 = 1$, i.e. $p_{1|0}(x_1|x_0) = \mathcal{N}(x_1; 0, \mathrm{Id})$. More precisely for any $0 \leq s \leq t \leq 1$, we have

$$p_{s|0}(x_s|x_0) = \int p_{t|0}(x_t|x_0) p_{s|t,0}(x_s|x_t, x_0) \mathrm{d}x_t.$$

**Sampling with DDIM.** At inference time, $x_{t_0} \sim p_{0|c}$ is generated by starting from a Gaussian $x_{t_N} \sim \mathcal{N}(0, \mathrm{Id})$ and $x_{t_k} \sim p(\cdot | x_{t_{k+1}}, c)$ for $k = N - 1, ..., 0$, where for $0 \leq s < t \leq 1$

$$p_{s|t,c}(x_s | x_t, c) = \int p_{s|t,0}(x_s | x_t, x_0) p_{0|t,c}(x_0 | x_t, c) \mathrm{d}x_0, \tag{4}$$

where $p_{0|t,c}(x_0 | x_t, c)$ is a posterior distribution, see also (Song et al., 2021a) for discussion. Since we do not have access to $p_{0|t,c}$, we approximate it by $\delta_{\hat{x}_\theta(x_t,c)}$ for any $t \in [0, 1]$, where $\hat{x}_\theta(x_t, c) \approx \mathbb{E}[x_0 | x_t, c]$ is a neural network denoiser with parameters $\theta$ trained by minimizing the loss

$$\mathcal{L}(\theta) = \int_0^1 \lambda(t) \mathbb{E}_{(x_0,c) \sim p_{0,c}, x_t \sim p_{t|0}}[\|x_0 - \hat{x}_\theta(x_t, c)\|^2] \mathrm{d}t. \tag{5}$$

Here, $\lambda(t) \geq 0$ is a weighting function (see (Kingma et al., 2021)). This yields the following procedure to sample approximately from $p_{0|c}$, by starting from $x_{t_N} \sim \mathcal{N}(0, \mathrm{Id})$, we then follow

$$x_{t_k} \sim \mathcal{N}(\mu_{t_k, t_{k+1}}(\hat{x}_\theta(x_{t_{k+1}}, c), x_{t_{k+1}}), \Sigma_{t_k, t_{k+1}}), \qquad k = N - 1, ..., 0 \tag{6}$$

**Classifier-Free Guidance (CFG).** Let $\hat{x}_\theta(x_t, \varnothing) \approx \mathbb{E}[x_0 | x_t]$ be an unconditional denoiser learned using a regression objective (5) (where conditioning is omitted). CFG replaces $\hat{x}_\theta(x_t, c)$ when simulating approximately from $p_{0|c}$ by

$$\hat{x}_\theta(x_t, c; \omega) = \hat{x}_\theta(x_t, c) + \omega \Delta_\theta(x_t, c), \tag{7}$$

where $\omega$ is the *guidance weight* and $\Delta_\theta(x_t, c) = \hat{x}_\theta(x_t, c) - \hat{x}_\theta(x_t, \varnothing)$. In order to sample the data with CFG, we first sample $x_{t_N} \sim \mathcal{N}(0, \mathrm{Id})$ and then follow

$$x_{t_k} \sim \mathcal{N}(\mu_{t_k, t_{k+1}}(\hat{x}_\theta(x_{t_{k+1}}, c; \omega), x_{t_{k+1}}), \Sigma_{t_k, t_{k+1}}), \quad k = N - 1, ..., 0. \tag{8}$$

When $\omega = 0$, the sampling procedure (8) is equivalent to *conditional* sampling (6), while $\omega = -1$ corresponds to a procedure using only the unconditional denoiser, allowing to sample approximately from $p_0$. In practice, one sets $\omega > 0$ in order to emphasize conditioning.

In this paper, we generalize CFG and make *guidance weight* $\omega_{c,(s,t)}$ a function of conditioning $c$ and of two time-steps $s < t$, where $t$ is the *proposal* timestep from which we denoise towards a *target* timestep $s$ as in (4). Denoising following (8) corresponds to setting, $s = t_k$ and $t = t_{k+1}$. We denote $\boldsymbol{\omega} = (\omega_{c,(s,t)})_{s,t \in [0,1], t > s}$. In Algorithm 2, we describe the CFG sampling method with DDIM.

**Maximum Mean Discrepancy.** To optimize the guidance weights $\boldsymbol{\omega}$, we match the true diffusion distribution $p$ and the guided $p^{\boldsymbol{\omega}}$, see Section 3. We employ the Maximum Mean Discrepancy (MMD) (Gretton et al., 2012) with energy kernel (Székely and Rizzo, 2004; Sejdinovic et al., 2013),

$$\mathrm{MMD}_{(\beta,\lambda)}[p^{\boldsymbol{\omega}}, p] = \mathbb{E}_{p_{\boldsymbol{\omega}} \otimes p}[\|x - y\|_2^\beta] - \tfrac{\lambda}{2}(\mathbb{E}_{p^{\boldsymbol{\omega}} \otimes p^{\boldsymbol{\omega}}}[\|x - x'\|_2^\beta] + \mathbb{E}_{p \otimes p}[\|y - y'\|_2^\beta]), \tag{9}$$

for independent $x, x' \sim p^{\boldsymbol{\omega}}$ and $y, y' \sim p$, with $\beta \in (0, 2)$ and $\lambda \in [0, 1]$. We require $\lambda = 1$ for a valid MMD, however $\lambda < 1$ may be preferable for generative modeling (Bouchacourt et al., 2016; De Bortoli et al., 2025); other characteristic kernels may also be used (Sriperumbudur et al., 2010).

## 3 LEARNING TO GUIDE YOUR DIFFUSION MODEL

In this section, we introduce our method for learning the guidance weights, $\boldsymbol{\omega}$, assuming access to a pre-trained conditional denoiser, $\hat{x}_\theta(x_t, c)$, and an unconditional one, $\hat{x}_\theta(x_t, \varnothing)$. Our approach is based on enforcing *consistency conditions*— that any valid diffusion process must satisfy. We first derive a theoretically sound objective from *marginal consistency* condition, which is however impractical to optimize due to high variance. To overcome this, we introduce a stronger, more practical condition, which we call *self-consistency*. Enforcing *self-consistency* is sufficient for achieving *marginal consistency* and, crucially, results in a simple, low-variance objective. We formulate our complete approach for learning guidance weights based on this simplified approach. We present alternative approaches in Appendix C.

### 3.1 CONSISTENCY CONDITIONS

**Marginal consistency.** The process defined by (1) admits the marginal distribution for $s \in (0, 1]$

$$p_s(x_s) = \int p_{s|0}(x_s | x_0) p_{0,c}(x_0, c) \mathrm{d}x_0 \mathrm{d}c. \tag{10}$$

This marginal distribution can also be obtained for any $0 \leq s < t \leq 1$ via

$$p_s(x_s) = \int \int \int p_{s|t,c}(x_s|x_t,c)p_{t|0}(x_t|x_0)p_{0,c}(x_0,c)\mathrm{d}x_t\mathrm{d}x_0\mathrm{d}c, \tag{11}$$

with $p_{s|t,c}$ given by DDIM (4). The equality (11) states that the marginal at time $s$ can be obtained by sampling $(x_0, c) \sim p_{0,c}$, noising $x_0$ to $x_t|x_0 \sim p_{t|0}$ and then denoising it back to time $s$ with $p_{s|t,c}$.

We now consider a denoising mechanism relying on the guided denoiser approximation (7) which follows the construction in (11) to obtain a sample at time $s$. Again we sample $(x_0, c) \sim p_{0,c}$ and $x_t \sim p_{t|0}(\cdot|x_0)$. However, the denoising to time $s$ where $0 \leq s < t \leq 1$ is done with the guided denoiser approximation, so that the marginal distribution of the resulting sample at time $s$ is

$$p_s^{t,(\theta,\boldsymbol{\omega})}(x_s) = \int \int \int p_{s|t,c}^{(\theta,\boldsymbol{\omega})}(x_s|x_t,c)p_{t|0}(x_t|x_0)p_{0,c}(x_0,c)\mathrm{d}x_t\mathrm{d}x_0\mathrm{d}c, \tag{12}$$

where

$$p_{s|t,c}^{(\theta,\boldsymbol{\omega})}(x_s|x_t,c) = p_{s|t,0}(x_s|x_t, \hat{x}_\theta(x_t,c;\boldsymbol{\omega})). \tag{13}$$

The distribution $p_s^{t,(\theta,\boldsymbol{\omega})}$ (12) is typically not equal to $p_s$ (10) as we used a delta-function approximation (13) with a model $\hat{x}_\theta(x_t,c;\boldsymbol{\omega})$ instead of sampling from $p_{0|t,c}$ as required by (4). Nevertheless, we could attempt to find guidance weights $\boldsymbol{\omega}$ to satisfy *marginal consistency*, i.e. for all $0 \leq s < t \leq 1$

$$p_s^{t,(\theta,\boldsymbol{\omega})}(x_s) \approx p_s(x_s). \tag{14}$$

This could be achieved by minimizing the MMD (9)

$$\mathcal{L}_m(\boldsymbol{\omega}) = \mathbb{E}_{(s,t)\sim p(s,t)}[\mathrm{MMD}_{(\beta,\lambda)}[p_s^{t,(\theta,\boldsymbol{\omega})}(\cdot), p_s(\cdot)]], \tag{15}$$

where $p(s,t)$ is a distribution on $0 \leq s < t \leq 1$. This distribution is crucial for good empirical performance, and we discuss it in detail in Section 3.2. The gradient of (15) could suffer from high variance due to marginalization over $(x_0, c)$, however, which could make it challenging to optimize. We found that this approach did not work in practice. Below, we propose a simpler, lower variance method, albeit one that imposes more constraints on the guided backward scheme compared to (15).

**Self-consistency (conditioning on $(c, x_0)$).** For a fixed $(x_0, c) \sim p(x_0, c)$, we denote by

$$p_{s|0,c}^{t,(\theta,\boldsymbol{\omega})}(x_s|x_0,c) = \int p_{s|t,c}^{(\theta,\boldsymbol{\omega})}(x_s|x_t,c)p_{t|0}(x_t|x_0)\mathrm{d}x_t, \tag{16}$$

which is the term under integral in (12) depending on $(x_0, c)$. We consider a *self-consistency* condition

$$p_{s|0,c}^{t,(\theta,\boldsymbol{\omega})}(x_s|x_0,c) \approx p_{s|0,c}(x_s|x_0,c) = p_{s|0}(x_s|x_0), \tag{17}$$

where $p_{s|0,c}^{t,(\theta,\boldsymbol{\omega})}(\cdot|x_0,c)$ is given by (16) and $p_{s|0}(\cdot|x_0)$ is a *noising process* (3). We used a notation $p_{s|0,c}(x_s|x_0,c) = p_{s|0}(x_s|x_0)$ to highlight that $x_0 \sim p_{0|c}(\cdot|c)$. Intuitively, this condition means that as we start from $x_0|c$ and go through the *noising process* $x_t|x_0$ and then denoise with guidance to time $s < t$, the distribution at time $s$ should be the same as of the *noising process*.

The condition (17) is much stronger than (14). Indeed, if (17) is satisfied for every $(x_0, c) \sim p_{0,c}$, then by integrating it over $(x_0, c)$, we will get (14). The reverse is not true. Moreover, the lack of dependence on $x_t$ (and on $x_0$) in guidance weights $\omega_{c,(s,t)}$ makes it very unlikely for this condition to hold. However, it provides a learning signal for guidance weights and we will aim to satisfy it approximately.

For $(x_0, c)$ in the support of $p(x_0, c)$, we could approximately satisfy (17) by minimizing wrt $\boldsymbol{\omega}$, $\mathrm{MMD}_{(\beta,\lambda)}[p_{s|0,c}^{t,(\theta,\boldsymbol{\omega})}(\cdot|x_0,c), p_{s|0,c}(\cdot|x_0)]$, see (9). Averaging over all $(x_0, c) \sim p(x_0, c)$ leads to

$$\mathcal{L}_{\beta,\lambda}(\boldsymbol{\omega}) = \mathbb{E}_{(x_0,c)\sim p_{0,c},s,t\sim p(s,t)}[\mathrm{MMD}_{(\beta,\lambda)}[p_{s|0,c}^{t,(\theta,\boldsymbol{\omega})}(\cdot|x_0,c), p_{s|0,c}(\cdot|x_0)]]. \tag{18}$$

This approach does not suffer from high variance compared to (15), since both $c$ and $x_0$ are fixed. Furthermore, we found that using (18) works well in practice (see our experiments in Section 5).

## 3.2 LEARNING TO GUIDE

We present here our approach for learning guidance weights $\boldsymbol{\omega}$ based on the *self-consistency* loss (18). We also provide empirical evaluation of other approaches in Section 5.

**Guidance learning objective.** For $(x_0, c) \sim p_{0,c}(x_0, c)$, we sample $x_s \sim p_{s|0}(\cdot|x_0)$, i.e., $x_s \sim \mathcal{N}(\alpha_s x_0, \sigma_s^2 \mathrm{Id})$. We also sample $\tilde{x}_s(\boldsymbol{\omega}) \sim p_{s|0,c}^{t,(\theta,\boldsymbol{\omega})}$, i.e., sample $\tilde{x}_t \sim \mathcal{N}(\alpha_t x_0, \sigma_t^2 \mathrm{Id})$ then

$$\tilde{x}_s(\boldsymbol{\omega}) \sim \mathcal{N}(\mu_{s,t}(\hat{x}_\theta(\tilde{x}_t, c; \omega_{c,(s,t)}), \tilde{x}_t), \Sigma_{s,t}), \tag{19}$$

where $\mu_{s,t}$ and $\Sigma_{s,t}$ are given by DDIM (see (23) in Appendix B). The objective (18) can be written as

$$\mathcal{L}_{\beta,\lambda}(\boldsymbol{\omega}) = \mathbb{E}_{(x_0,c)\sim p_{0,c},s,t\sim p(s,t)}[\mathbb{E}[||\tilde{x}_s(\boldsymbol{\omega}) - x_s||_2^\beta] - \tfrac{\lambda}{2}\mathbb{E}[||\tilde{x}_s(\boldsymbol{\omega}) - \tilde{x}_s'(\boldsymbol{\omega})||_2^\beta]], \tag{20}$$

where we dropped terms not depending on $\boldsymbol{\omega}$, and with independent $\tilde{x}_s(\boldsymbol{\omega}), \tilde{x}_s'(\boldsymbol{\omega}) \sim p_{s|0,c}^{t,(\theta,\boldsymbol{\omega})}$.

**Simplified guidance learning objective.** A special case of (20) with $\beta = 2$ and $\lambda = 0$, leads to

$$\mathcal{L}_{\ell_2}(\boldsymbol{\omega}) = \mathcal{L}_{\beta,0}(\boldsymbol{\omega}) = \mathbb{E}_{(x_0,c)\sim p_{0,c},s,t\sim p(s,t)}\left[\mathbb{E}[||\tilde{x}_s(\boldsymbol{\omega}) - x_s||_2^2]\right]. \tag{21}$$

We found that (21) was very effective but more sensitive to hyperparameters than (20). This approach is however cheaper than (20) since it avoids quadratic complexity $O(m^2)$ of computing interaction terms. We refer the reader to Appendix D and to Algorithm 3 for more details on the use of (21).

**Guidance network.** The guidance weights $\omega_{c,(s,t)}^\phi = \omega(s,t,c;\phi)$ are given by a neural network with parameters $\phi$. We use ReLU activation at the end to prevent negative guidance weights. For more details, see Appendix F. We denote $\boldsymbol{\omega}^\phi = (\omega_{c,(s,t)}^\phi)_{s,t\in[0,1],t>s}$ and employ $\mathcal{L}_{\beta,\lambda}(\phi)$ instead of $\mathcal{L}_{\beta,\lambda}(\boldsymbol{\omega})$.

**Distribution $p(s,t)$.** We choose target time $s$ to be distributed as $s \sim \mathcal{U}[S_{\min}, 1 - \zeta - \delta]$. We define $\Delta t \sim \mathcal{U}[\delta, 1 - \zeta - s]$ and we let $t = s + \Delta t$. Here, $S_{\min}$ controls the minimal time, and $\zeta$ controls how close it gets to 1. The parameter $\delta$ controls the distance between time-steps and we found it to be very important, see Figure 2. Even though during inference $|t - s| \approx \frac{1}{T}$ is typically small ($T$ is a number of steps), we found that using larger $\delta \approx 0.1$ during training worked better in practice.

**Empirical objective.** We sample $\{x_0^i, c^i\}_{i=1}^n \overset{\text{i.i.d.}}{\sim} p_{0,c}$ from the training set and we additionally sample noise levels $\{s_i\}_{i=1}^n \overset{\text{i.i.d.}}{\sim} \mathcal{U}[S_{\min}, 1 - \zeta - \delta]$, as well as time increments $\{\Delta t_i\}_{i=1}^n \overset{\text{i.i.d.}}{\sim} \mathcal{U}[\delta, 1 - \zeta - s_i]$ and we let $t_i = s_i + \Delta t_i$. For each $(x_0^i, s_i)$, we sample $m$ "particles" $\{x_{s_i}^j\}_{j=1}^m \overset{\text{i.i.d.}}{\sim} p_{s_i|0}(\cdot|x_0^i)$ from the *noising process* (3), which defines the target samples. We also produce $m$ "particles" $\{\tilde{x}_{s_i}^j(\boldsymbol{\omega}^\phi)\}_{j=1}^m \overset{\text{i.i.d.}}{\sim} p_{s_i|0,c_i}^{t_i,(\theta,\boldsymbol{\omega}^\phi)}(\cdot|x_0^i, c_i)$ by first sampling $\{\tilde{x}_{t_i}^j\}_{j=1}^m \overset{\text{i.i.d.}}{\sim} p_{t_i|0}(\cdot|x_0^i)$ from the *noising process* (3) and then denoising with guidance and DDIM using (19). This defines the proposal samples. We expand loss function (20) as a function of guidance network parameters $\phi$ defined on the empirical batches as follows (where $\lambda \in [0,1]$ and $\beta \in (0,2)$, see Algorithm 1)

$$\hat{\mathcal{L}}_{\beta,\lambda}(\phi) = \tfrac{1}{n}\sum_{i=1}^n \left[\tfrac{1}{m^2}\sum_{j,k=1}^m ||\tilde{x}_{s_i}^j(\boldsymbol{\omega}^\phi) - x_{s_i}^k||_2^\beta - \tfrac{\lambda}{2}\tfrac{1}{m(m-1)}\sum_{j\neq k}||\tilde{x}_{s_i}^j(\boldsymbol{\omega}^\phi) - \tilde{x}_{s_i}^k(\boldsymbol{\omega}^\phi)||_2^\beta\right] \tag{22}$$

**Learning to guide with rewards.** CFG can be used to produce samples with high reward $R(x_0, c)$, set by a practitioner. Denoising with guidance from $t$ to $s$ as described by (19), gives us an approximation $\hat{x}_\theta(x_t, c; \omega_{c,(s,t)}^\phi)$ of clean data. We could use it to optimize guidance weights, by defining the loss $\mathcal{L}_R(\phi) = -\mathbb{E}_{(s,t)\sim p(s,t),x_0,c\sim p(x_0,c),x_t\sim p_{t|0}(\cdot|x_0)}\left[R\left(\hat{x}_\theta(x_t, c; \omega_{c,(s,t)}^\phi), c\right)\right]$. Directly minimizing this loss may lead to *reward hacking* (Skalse et al., 2022). Thus, we regularize this objective using $\hat{\mathcal{L}}_{\beta,\lambda}$. For reward weight $\gamma_R \geq 0$, we optimize $\mathcal{L}_{\text{tot}}(\phi) = \hat{\mathcal{L}}_{\beta,\lambda}(\phi) + \gamma_R \mathcal{L}_R(\phi)$ (see Algorithm 4).

## 4 RELATED WORK

**Classifier-Free Guidance.** Using guidance in the sampling of diffusion models was first investigated by Dhariwal and Nichol (2021). They proposed a classifier guidance method which linearly combines the unconditional score estimate and the input gradient of the log-probability of a (time-varying) classifier. To avoid training such classifier on noisy training data, Ho and Salimans (2022) proposed Classifier-Free Guidance (CFG), which linearly combines a conditional and unconditional denoisers. By varying the guidance weight, one is able to obtain high-quality samples. This approach has become prominent in the literature, see Adaloglou and Kaiser (2024) for a recent introduction.

---

**Algorithm 1** Learning to Guide

---

1: **Input:** Init. guidance parameters $\phi$; (frozen) denoiser $\hat{x}_\theta$; data distribution $p_0$; learning rate $\eta$; $\zeta > 0, S_{\min} > 0, \delta > 0$, b.s. $n$, n. of particles $m, \lambda \in [0,1], \beta \in [0,2]$, DDIM churn $\varepsilon \in [0,1]$.

2: **repeat**

3:     Sample batch of clean data and their conditionings $\{x_0^i, c^i\}_{i=1}^n \overset{\text{i.i.d.}}{\sim} p_{0,c}$.

4:     Sample $\{s_i\}_{i=1}^n \overset{\text{i.i.d.}}{\sim} \mathcal{U}[S_{\min}, 1 - \zeta - \delta], \{\Delta t_i\}_{i=1}^n \overset{\text{i.i.d.}}{\sim} \mathcal{U}[\delta, 1 - \zeta - s_i]$, let $t_i = s_i + \Delta t_i$

5:     (**True process**) Sample $m$ particles $\{x_{s_i}^j\}_{j=1}^m \overset{\text{i.i.d.}}{\sim} p_{s_i|0}(\cdot | x_0^i)$ from noising process (3)

6:     (**Guided process**) Sample $m$ particles $\{\tilde{x}_{t_i}^j\}_{j=1}^m \sim p_{t_i|0}(\cdot | x_0^i)$ from noising process (3)

7:     Compute guidance weights $\omega_i = \omega_{c_i,(s_i,t_i)}^\phi$ and $\hat{x}_\theta(\tilde{x}_{t_i}^j, c_i; \omega_i)$ using (7)

8:     Sample $\tilde{x}_{s_i}^j(\boldsymbol{\omega}^\phi) \sim p_{s_i|t_i,0}(\cdot | \tilde{x}_{t_i}^j, \hat{x}_\theta(\tilde{x}_{t_i}^j, c_i; \omega_i))$ from DDIM (2) with *churn* parameter $\varepsilon$

9:     (**Loss**) Compute loss

10:     $\hat{\mathcal{L}}_{\beta,\lambda}(\phi) = \frac{1}{n} \sum_{i=1}^n \left[ \frac{1}{m^2} \sum_{j,k=1}^m ||\tilde{x}_{s_i}^j(\boldsymbol{\omega}^\phi) - x_{s_i}^k||_2^\beta - \frac{\lambda}{2} \frac{1}{m(m-1)} \sum_{j \neq k} ||\tilde{x}_{s_i}^j(\boldsymbol{\omega}^\phi) - \tilde{x}_{s_i}^k(\boldsymbol{\omega}^\phi)||_2^\beta \right]$

11:     Update $\phi \leftarrow \phi - \eta \nabla_\phi \hat{\mathcal{L}}_{\beta,\lambda}(\phi)$

12: **until** convergence

13: **Output:** Optimized guidance network parameters $\phi$

---

CFG has been extended in many different directions, some of them proposing dynamic mixing strategies to control the guidance weight throughout the sampling process (e.g. Sadat et al. (2024a); Kynkäänniemi et al. (2024); Wang et al. (2024); Shen et al. (2024); Malarz et al. (2025); Li et al. (2024); Koulischer et al. (2025a;b); Xia et al. (2025); Sadat et al. (2024b); Zheng and Lan (2024)). Another line of research focuses on removing the need for the unconditional score component by using either an "inferior" version of the conditional model to provide a negative guidance signal (Karras et al., 2024; Adaloglou et al., 2025), by leveraging time-step information (Sadat et al., 2025), or by enabling the model to act as its own implicit classifier (Tang et al., 2025). Other approaches focus on fixing some of the limitations of classifier-free guidance; e.g., Chung et al. (2025) introduced CFG++, a modification to the standard CFG to address off-manifold issues, while Koulischer et al. (2025b) show that the true conditional distribution can be obtained by approximating a term corresponding to the derivative of a Rényi divergence. Very recently, Fan et al. (2025) introduced CFG-Zero*, a CFG variant for flow-matching models. It leverages an optimized scale factor and a zero-init technique (skipping initial ODE steps) to correct for early velocity inaccuracies, resulting in improved text-to-image/video generation. *We highlight that our approach could be seamlessly combined with any other guidance technique by simply replacing the definition of the guided denoiser (7).* Finally, there exists a large literature on correcting CFG with Sequential Monte Carlo (Skreta et al., 2025; He et al., 2025) or MCMC techniques (Du et al., 2023; Moufad et al., 2025; Zhang et al., 2025). These approaches are mostly orthogonal to other improvements of CFG, and can be combined with most of the dynamic mixing strategies (i.e. Malarz et al. (2025)).

CFG has also been reinterpreted as a *predictor-corrector* in Bradley and Nakkiran (2024) and analyzed theoretically in a variety of works (Fu et al., 2024; Wu et al., 2024; Chidambaram et al., 2024; Kong et al., 2024; Frans et al., 2025). Pavasovic et al. (2025) have shown that while CFG can overshoot the target distribution in low dimension, it can reproduce the target distribution in high dimensions.

Concurrently to our work, Papalampidi et al. (2025) introduced dynamic classifier guidance by selecting the guidance weight among a list of pre-determined guidance weights using online evaluators.

**Correcting distribution mismatch.** Like our method, the time-tuner approach (Xia et al., 2024) corrects the mismatch between the sampled and target conditional distributions. However, time-tuner specifically addresses backward solver discretization errors in low-NFE regimes by adjusting the denoiser's noise level. In contrast, our approach addresses the distributional mismatch caused by CFG to improve alignment and quality, independent of NFE.

**Learning CFG weight.** Azangulov et al. (2025) introduced an algorithm based on stochastic control to optimize the guidance weights. However, this algorithm is not scalable as it requires guided backwards trajectories to estimate the gradient. Concurrently to us, Yehezkel et al. (2025) introduced an annealing guidance scheduler that dynamically adjusts the guidance weight based on

the timestep and the magnitude of the conditional noise discrepancy. The method learns an adaptive policy to better balance image quality and alignment with the text prompt throughout the generation process. One of their losses is similar to an alternative approach we explored – guided score matching (see Appendix C.1). We found that optimizing such loss led to guidance weights equal to zero.

**Distillation.** Our method shares similarities with distillation approaches such as Diff-Instruct (Luo et al., 2023), Variational Score Distillation (Wang et al., 2023) and Moment Matching Distillation (MMD) (Salimans et al., 2024), which distill a diffusion model by approximately minimizing the KL divergence between the distilled generator and the pretrained teacher model. One could think about our method as a form of distillation where the KL divergence is replaced by a scoring rule, the pretrained teacher model is replaced by the true distribution, and the student is replaced by a guided diffusion. Our methodology could be extended to the distillation setting.

**Distributional approaches for diffusion models.** Our objective function was motivated by (De Bortoli et al., 2025), where (9) is used to learn $p(x_0|x_t, c)$ to obtain distributional diffusion models. Our method is also related to Inductive Moment Matching (IMM) (Zhou et al., 2025), where a generative model is trained by enforcing marginal consistency (14). Instead of directly optimizing for (14), their objective minimizes the distance between $p_s^{t,(\theta_n,\boldsymbol{\omega})}$ and $p_s^{r,(\theta_{n-1},\boldsymbol{\omega})}$, where $n$ is an iteration number and $s < r < t$ is an intermediate time between $s$ and $t$. However, they do not marginalize over $(x_0, c) \sim p_{0,c}$ as in (12); instead, they sample a batch of $(x_0, c)$ and use it for both $p_s^{t,(\theta_n,\boldsymbol{\omega})}$ and $p_s^{r,(\theta_{n-1},\boldsymbol{\omega})}$. This means they optimize an objective similar to self-consistency (17), where they use a batch of $(x_0, c)$ instead of single points.

## 5 EXPERIMENTAL RESULTS

In this section, we present our core experimental results on learning guidance weights. In Section 5.1, we provide results for image generation benchmarks – ImageNet $64 \times 64$ (Deng et al., 2009) and CelebA (Liu et al., 2015) with resolution $64 \times 64$. We provide results on text-to-image (T2I) benchmark - MS-COCO 2014 (Lin et al., 2014) (10K images) at $512 \times 512$ resolution in Section 5.2. We also refer the reader to Appendix G for a discussion about training and inference cost of our method.

**Ablations.** We refer the reader to Appendix H for ablations of our method. In Appendix H.1, we provide ablations over $\beta$ and $m$ on ImageNet $64 \times 64$ and in Appendix H.2, we provide ablation over conditioning in T2I. Finally, in Appendix H.3, we provide ablation over guidance network size and architecture on T2I.

**Additional results.** We refer the reader to Appendix I for additional results and comparisons. In Appendix I.1, we provide comparisons to clamp-linear schedule (Wang et al., 2024) on ImageNet $64 \times 64$, and in Appendix I.2, we add comparisons to limited interval guidance (LIG) (Kynkäänniemi et al., 2024) and clamp linear schedule on T2I. Moreover, in Appendix I.3, we add results on Stable Diffusion-v1.5 (Rombach et al., 2022b). Furthermore, we report 2D Mixture of Gaussians (MoG) results in Appendix I.4. Finally, in Appendix J, we provide an analysis of learned guidance weights in T2I and ImageNet $64 \times 64$.

### 5.1 IMAGE GENERATION

**Experimental setting.** We evaluate the performance of our method on ImageNet $64 \times 64$ and CelebA with resolution $64 \times 64$. As evaluation metrics, we use FID (Heusel et al., 2017) and Inception Score (IS) (Salimans et al., 2016). First, we pretrain diffusion models on ImageNet and CelebA. We then freeze these models and train guidance network $\omega_{c,(s,t)}^{\phi}$ via Algorithm 1. We also train a variant of our method where conditioning is omitted, i.e. $\omega_{(s,t)}^{\phi}$. On top of that, we also train the simplified $\ell_2$ method using Algorithm 3 using the same methodology. We report metrics based on $50k$ samples. Please refer to Appendix F for more details.

**Baselines.** We report performance of the unguided model as well as the model with a constant guidance. We further report performance of limited interval guidance (LIG) (Kynkäänniemi et al., 2024). Guidance scale and intervals were selected via grid search for the lowest FID (see Appendix F).

Table 1: **ImageNet 64x64**. We report FID and IS for different methods (the best are in **bold**).

| Method Name | Guidance Weight | FID ↓ | Inception Score (IS) ↑ |
|---|---|---|---|
| **Baselines** | | | |
| Unguided | $\omega = 0$ | 4.46 | 43.52 |
| Constant guidance | $\omega = 0.25$ | 2.40 | 66.72 |
| Limited interval guidance | $\omega(t) = 0.95$ for $t \in [0.2, 0.8]$ | **2.11** | **71.60** |
| **Learned guidance approaches** | | | |
| Self-consistency (20) | $\omega^{\phi}_{c,(s,t)}$ | **1.99** | 73.62 |
| Self-consistency (20) | $\omega^{\phi}_{(s,t)}$ | 2.07 | 76.7 |
| $\ell_2$ objective (21) | $\omega^{\phi}_{c,(s,t)}$ | 2.09 | 75.93 |
| $\ell_2$ objective (21) | $\omega^{\phi}_{(s,t)}$ | 2.10 | **77.55** |

**Results.** The results for ImageNet $64 \times 64$ are given in Table 1 and for CelebA in Table 2. In both cases, the LIG baseline outperforms the other baselines in terms of FID and Inception Score. Our self-consistency approach (20) leads to the best results, especially when the conditioning information is provided. This highlights the importance of adjusting guidance weights for different conditioning.

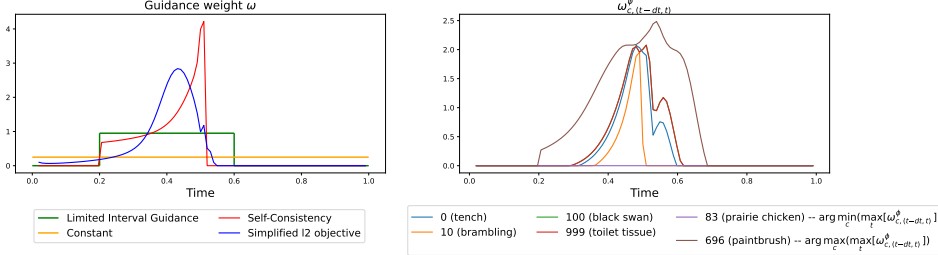

Figure 1: **Learned guidance weights on ImageNet 64x64**. **Left**, guidance weights $\omega^{\phi}_{(t-dt,t)}$ (conditioning-agnostic) for baselines as well as for self-consistency (20) and $\ell_2$ (21) objectives, where $dt = 1/100$. X-axis is time. **Right**, guidance weights $\omega^{\phi}_{c,(t-dt,t)}$ for specific ImageNet classes.

In Figure 1, left, we visualize the learned conditioning-agnostic guidance weights $\omega^{\phi}_{(t-dt,t)}$ with $dt = 1/100$, for self-consistency (20) and $\ell_2$ (21) approaches, as well as for the baselines. Learned guidance weights seem to be positive on a similar interval as LIG, but the shape of the weights is quite different. In Figure 1, right, we visualize guidance weights $\omega^{\phi}_{c,(t-dt,t)}$ for different classes of ImageNet, learned by self-consistency (20). First, we arbitrary choose some classes – 0 (tench), 10 (brambling), 100 (black swan) and 999 (toilet tissue). Then, we visualize the ones which achieve the largest and the lowest guidance weights, i.e. 83 (prairie chicken) = $\arg\min_c[\max_t \omega^{\phi}_{c,(t-dt,t)}]$ and 696 (paintbrush) = $\arg\max_c[\max_t \omega^{\phi}_{c,(t-dt,t)}]$. For the class 83 (prairie chicken), the guidance weight is zero. For the class 696 (paintbrush) it has quite a different behavior compared to others, being more aggressive and positive on a larger interval. Overall, this, variability highlights importance of adjusting guidance weights per conditioning.

**Ablations.** In Figure 2, we study impact of $\delta$ and $S_{\min}$ on ImageNet $64 \times 64$. We always use $\zeta = 0.01$ which introduces a small "safety margin" over the original diffusion model (i.e. $t \in [\zeta, 1-\zeta]$ instead of $t \in [0, 1]$). The self-consistency approach (17) is less sensitive to the parameters compared to $\ell_2$ (21). Overall, very small $\delta = 0.01$ leads to worse performance compared to larger ones, motivating us to train with large gaps ($\delta \approx 0.1$) between $s$ and $t$. This finding is surprising because during sampling $|s-t| \sim 0.01$ for 100 sampling steps. We hypothesize that a larger gap $|s-t|$ provides a more stable and informative gradient signal for the guidance network, which then successfully

Table 2: **CelebA** $64 \times 64$. We report FID and IS for different methods (the best are in **bold**).

| Method Name | Guidance Weight | FID $\downarrow$ | Inception Score (IS) $\uparrow$ |
|---|---|---|---|
| **Baselines** | | | |
| Unguided | $\omega = 0$ | 2.44 | 2.94 |
| Constant guidance | $\omega = 0.01$ | 2.45 | 2.94 |
| Limited interval guidance | $\omega(t) = 0.7$ for $t \in [0.0, 0.8]$ | **2.37** | **2.96** |
| **Learned guidance approaches** | | | |
| Self-consistency (20) | $w^{\phi}_{c,(s,t)}$ | **2.10** | **2.98** |
| Self-consistency (20) | $w^{\phi}_{(s,t)}$ | 2.28 | 2.97 |
| $\ell_2$ objective (21) | $w^{\phi}_{c,(s,t)}$ | 2.36 | 2.95 |
| $\ell_2$ objective (21) | $w^{\phi}_{(s,t)}$ | 2.33 | 2.95 |

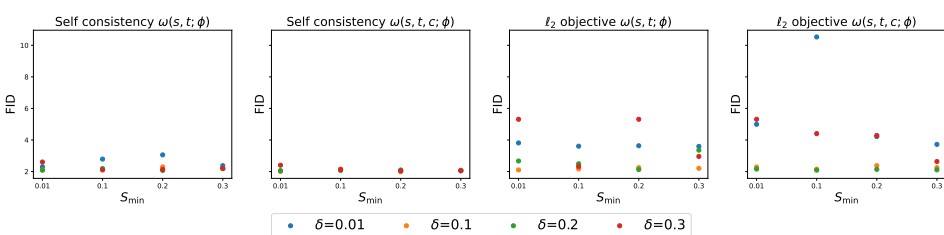

Figure 2: **Ablation over $\delta$ and $S_{\min}$ on ImageNet** $64 \times 64$. On the X-axis we report values of $S_{\min}$ and on Y-axis we show FID. Each column denotes a method while a color corresponds to a value of $\delta$.

generalizes to the small-step intervals used during inference thanks to the smoothness of the network $\omega^{\phi}_{c,(\cdot,\cdot)}$. The performance is not very sensitive to $S_{\min}$, but we found $S_{\min} = 0.2$ worked the best.

## 5.2 TEXT-TO-IMAGE GENERATION

We evaluate our method on MS COCO 2014 dataset at $512 \times 512$ resolution for text-to-image (T2I) task. We pretrain a 1.05B parameter flow matching model (Lipman et al., 2023) that uses a Multimodal Diffusion Transformer backbone (BlackForestLabs, 2025). We freeze it and train a guidance network $\omega^{\phi}_{c_{\text{CLIP}}, c_{\text{T5}}, (s,t)}$ via Algorithm 1, where $c_{\text{CLIP}}$ and $c_{\text{T5}}$ denote CLIP (Radford et al., 2021) and T5 (Raffel et al., 2020) embeddings of the text prompt. As baseline, we consider manually selected guidance weight $\omega$. Instead of an empty prompt $\varnothing$ for the unconditional term (7), we replace it with a fixed negative prompt $c_{\text{neg}} = $ "blurred, blurry, disfigured, ugly, tiling, poorly drawn". This 'negative guidance' setup is for both our learned model and the baseline. We also consider a setting with a reward function $R(x_0, c)$ given by CLIP score (Hessel et al., 2021) computed between image $x_0$ and prompt $c$. We train guidance network via Algorithm 4 with $\gamma_R = 10^5$. We report FID (Heusel et al., 2017) and CLIP Score using 10K samples. For more experimental details, see Appendix F.

**Results.** The quantitative results are summarized in Table 3. Our method outperforms unguided and guided baselines in terms of FID, which is consistent to image experiments. However, it achieves a lower CLIP score than a guided baseline. Adding CLIP score reward leads a similar CLIP score as guided baseline, but achieves lower FID. We provide qualitative results in Figures (4)-(6), see Appendix A. Our method generates images that are more realistic and better aligned with the text prompts. The learned guidance weights are shown in Figure 3. We observe high variability depending on the prompt.

## 6 DISCUSSION

In this paper, we presented an approach to learn CFG weights $\omega_{c,(s,t)}$ as a function of conditioning $c$ and times $s$ and $t$, using the self-consistency condition (17). This rather strong condition is motivated by a weaker marginal consistency condition (14), which is satisfied by the true backwards diffusion process. Our approach yields guidance weights that improve FID on image generation tasks – ImageNet $64 \times 64$ and CelebA $64 \times 64$. Our analysis reveals that guidance weights vary significantly de-

Table 3: **MS COCO** $512 \times 512$. FID and CLIP score for different methods (the best are in **bold**). We report the best qualitative results for the baselines: limited interval guidance (Kynkäänniemi et al., 2024) and clamp-linear schedule (Wang et al., 2024). We perform extensive hyperparameter tuning for each of these baselines as shown in Table 10 and Table 9

| Method Name | Guidance Weight | FID $\downarrow$ | CLIP Score $\uparrow$ |
|---|---|---|---|
| **Baselines** | | | |
| Unguided | $\omega = 0$ | **24.74** | 0.278 |
| Constant guidance | $\omega = 7.5$ | 31.2 | **0.306** |
| Limited interval guidance | $\omega(t) = 15.0$ for $t \in [0.1, 0.9]$ | 24.95 | 0.304 |
| Clamp-linear guidance | $\omega = 14.0$ | 27.14 | 0.305 |
| **Our approaches** | | | |
| Self-consistency (20) | $\omega^{\phi}_{c_{\text{CLIP}}, c_{\text{T5}, (s,t)}}$ | **18.01** | 0.295 |
| Self-consistency (20) + CLIP reward | $\omega^{\phi}_{c_{\text{CLIP}}, c_{\text{T5}, (s,t)}}$ | 28.37 | **0.306** |

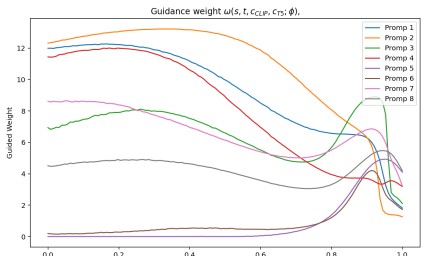

**Prompt 1**. Man performing stunt on a skateboard on a road.
**Prompt 2**. A street with cars and construction workers working.
**Prompt 3**. A bird is sitting on a branch among unfocused trees.
**Prompt 4**. A wooden table topped with four white bowls.
**Prompt 5**. A light brown horse's face is shown at close range.
**Prompt 6**. A living room with furniture, a fireplace, and a large scenic window.
**Prompt 7**. The large bird has a red face and black feathers.
**Prompt 8**. A small dog on TV behind the words: "What did I do wrong?"

Figure 3: **Learned guidance weights on MS COCO** $512 \times 512$ trained with self-consistency (20) and CLIP reward loss. Please refer to Figures 4 and 5 for the corresponding images.

pending on the conditioning information, implying that CFG with learnable, conditioning-dependent weights can improve conditional sampling performance.

We extended our methodology to text-to-image tasks with a reward function given by the CLIP score. We found that our approach leads to highly variable prompt-dependent guidance weights and visually provides better prompt alignment compared to baselines. Quantitatively, however, we found that the performance was close to a baseline with a manually selected guidance weight function.

Future work will focus on theoretical understanding of our objective function and its guidance solutions. Moreover, we will explore alternative reward functions for better prompt-image alignment, and investigate the impact of different guidance approaches. We hope our work motivates further research into how time- and conditioning-dependent guidance weights affect the sampled distributions.

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

ORGANIZATION OF THE APPENDIX

In Appendix A, we present the image samples for text-to-image experiment, see Section 5. Then, in Appendix B, we briefly cover the DDIM framework. Following that, in Appendix C, we discuss alternative approaches for learning guidance weights. In Appendix D, we discuss the simple approach based on $\ell_2$ objective (21) and in Appendix E, we discuss the extension of our method to reward-guided setting. In Appendix G, we discuss training and inference costs for the guidance network. Moreover, in Appendix F, we cover all the experimental details and in Appendix H, we present additional experimental results. Finally, in Appendix J, we provide analysis of the learned guidance weights in T2I and ImageNet $64 \times 64$.

## A  TEXT TO IMAGE SAMPLES

We provide qualitative results for the text-to-image experiments in Figure 4, Figure 5 and in Figure 6.

## B  DDIM FRAMEWORK

**Diffusion models.**  The goal of conditional diffusion models is to sample from a target conditional distribution $p_0(x|c)$ on $\mathbb{R}^d$, which we denote as $p_{0|c}$. Here $c$ is some user-specified conditioning signal (i.e., a class label or a text prompt) and $p(c)$ is a conditioning distribution.

We adopt here the Denoising Diffusion Implicit Models (DDIM) framework of Song et al. (2021a). Let $x_{t_0} \sim p_{0|c}$ and define the process $x_{t_1:t_N} := (x_{t_1}, ..., x_{t_N})$ by

$$p(x_{t_1:t_N}|x_{t_0}) = p(x_{t_N}|x_{t_0}) \prod_{k=1}^{N-1} p(x_{t_k}|x_{t_0}, x_{t_{k+1}}),$$

with $0 = t_0 < \cdots < t_N = 1$. For $0 \leq s < t \leq 1$, we let

$$p(x_s|x_0, x_t) = \mathcal{N}(x_s; \mu_{s,t}(x_0, x_t), \Sigma_{s,t}),$$

The mean and covariance of $p(x_s|x_0, x_t)$ are given by

$$\mu_{s,t}(x_0, x_t) = (\varepsilon^2 r_{1,2}(s, t) + (1 - \varepsilon^2)r_{0,1})x_t + \alpha_s(1 - \varepsilon^2 r_{2,2}(s, t) - (1 - \varepsilon^2)r_{1,1}(s, t))x_0,$$
$$\Sigma_{s,t} = \sigma_s^2(1 - (\varepsilon^2 r_{1,1}(s, t) + (1 - \varepsilon^2))^2)\mathrm{Id}, \tag{23}$$

with $r_{i,j}(s, t) = (\alpha_t/\alpha_s)^i (\sigma_s^2/\sigma_t^2)^j$; see (Song et al., 2021a) and Appendix F in (De Bortoli et al., 2025). The parameter $\varepsilon \in [0, 1]$ in (23) is a *churn* parameter which interpolates between a *deterministic* process ($\varepsilon = 0$) and a *stochastic* one ($\varepsilon = 1$). This ensures that for any $t \in [0, 1]$,

$$p(x_t|x_0) = \mathcal{N}(x_t; \alpha_t x_0, \sigma_t^2 \mathrm{Id}),$$

for $\alpha_t, \sigma_t$ such that $\alpha_0 = \sigma_1 = 1$ and $\alpha_1 = \sigma_0 = 0$. In particular, this guarantees that $p(x_1|x_0) = \mathcal{N}(x_1; 0, \mathrm{Id})$.

The data $x_{t_0} \sim p_{0|c}$ is generated by sampling $x_{t_N} \sim \mathcal{N}(0, \mathrm{Id})$ and $x_{t_k} \sim p(\cdot|x_{t_{k+1}})$ for $k = N - 1, ..., 0$, where for $0 \leq s < t \leq 1$

$$p(x_s|x_t, c) = \int_{\mathbb{R}^d} p(x_s|x_0, x_t)p(x_0|c, x_t)\mathrm{d}x_0.$$

## C  ALTERNATIVE APPROACHES FOR LEARNING GUIDANCE

We describe the alternative approaches for learning guidance weights and discuss the their potential downfalls.

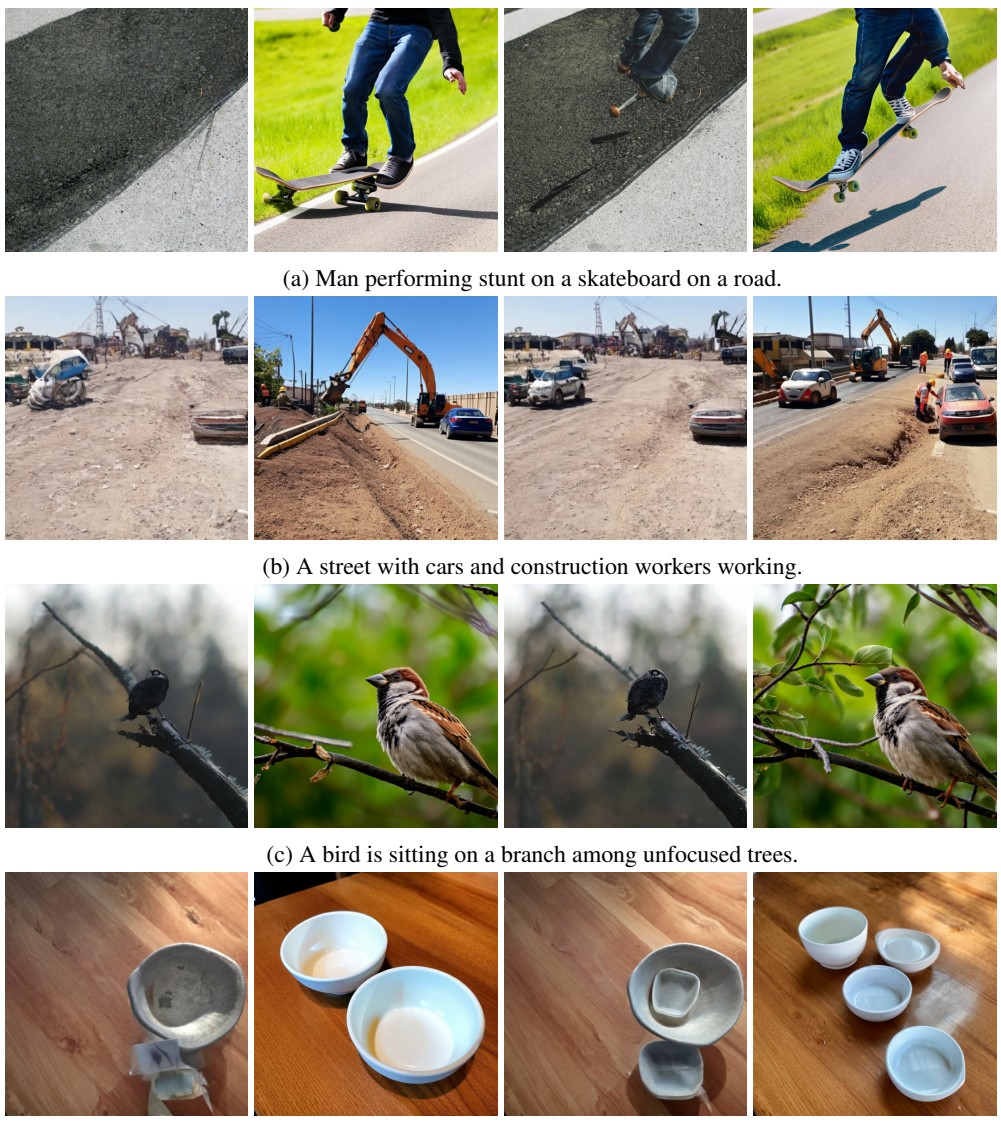

(a) Man performing stunt on a skateboard on a road.

(b) A street with cars and construction workers working.

(c) A bird is sitting on a branch among unfocused trees.

(d) A wooden table topped with four white bowls.

Figure 4: **T2I Results on MS-COCO. Part 1**. (left-to-right) We provide results of images generated from the given text prompt without CFG, with CFG $\omega = 7.5$, our method with self-consistency loss and our method with self-consistency loss and CLIP score reward.

## C.1 Guided score matching

The most straightforward approach for learning guidance weights is to simply use guided denoiser (7) in the score matching. This leads to the following objective

$$\mathcal{L}(\phi) = \int_0^1 \lambda(t) \mathbb{E}_{(x_0,c)\sim p(x_0,c), x_t \sim p_{t|0}(\cdot|x_0)}[\|x_0 - \hat{x}_\theta(x_t, c; \omega_{c,t}^\phi)\|^2]\mathrm{d}t, \tag{24}$$

where we omit the dependence of the guidance weights on $s$, i.e., $\omega_{c,(s,t)}^\phi = \omega_{c,t}^\phi$. However, this approach is doomed to fail, because at convergence, the solution to (24) satisfies the following property

$$\hat{x}_\theta(x_t, c; \omega_{c,t}^\phi) \approx \mathbb{E}[x_0|x_t, c], \tag{25}$$

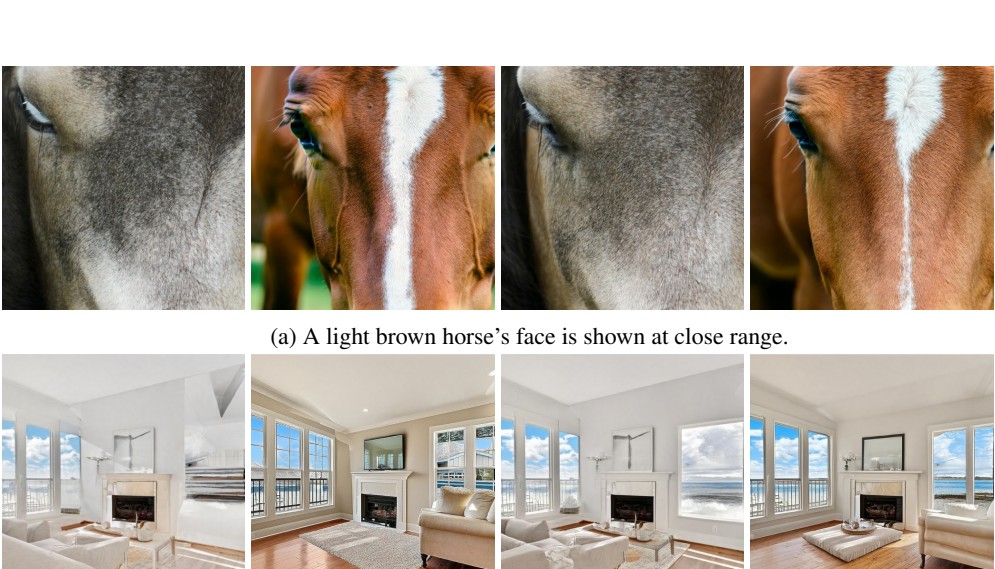

(a) A light brown horse's face is shown at close range.

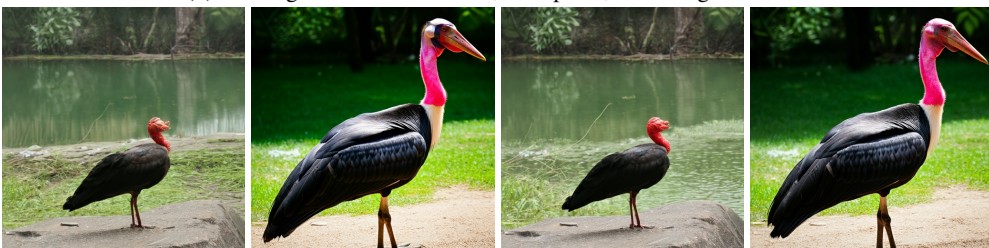

(b) A living room with furniture, a fireplace, and a large scenic window.

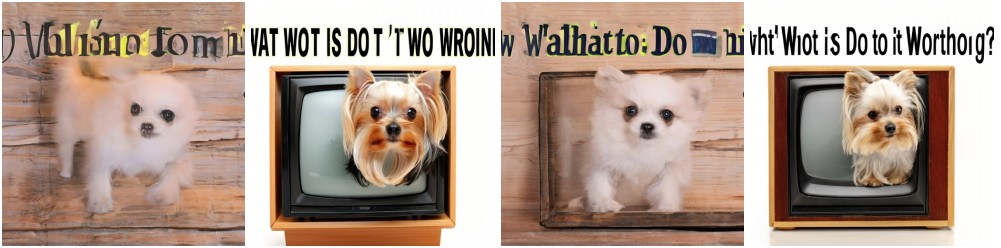

(c) The large bird has a red face and black feathers.

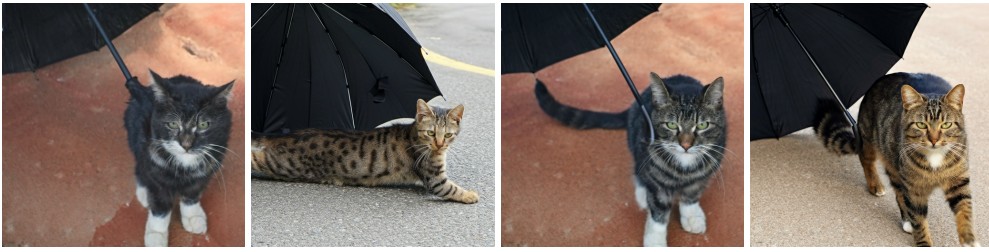

(d) A small dog on TV behind the words " What Did I Do Wrong?

(e) A cat walking under a black open umbrella.

Figure 5: **T2I Results on MS-COCO. Part 2 (Randomly selected)**. (left-to-right) We provide results of images generated from the given text prompt without CFG, with CFG $\omega = 7.5$, our method with self-consistency loss and our method with self-consistency loss and CLIP score reward.

when $x_t \sim p_{t|0}(x_t|x_0)$ is a sample from noising process (3). However, the condition (25) is also satisfied by the unguided score, when $x_t \sim p_{t|0}(x_t|x_0)$, i.e.

$$\hat{x}_\theta(x_t, c) \approx \mathbb{E}[x_0|x_t, c]. \tag{26}$$

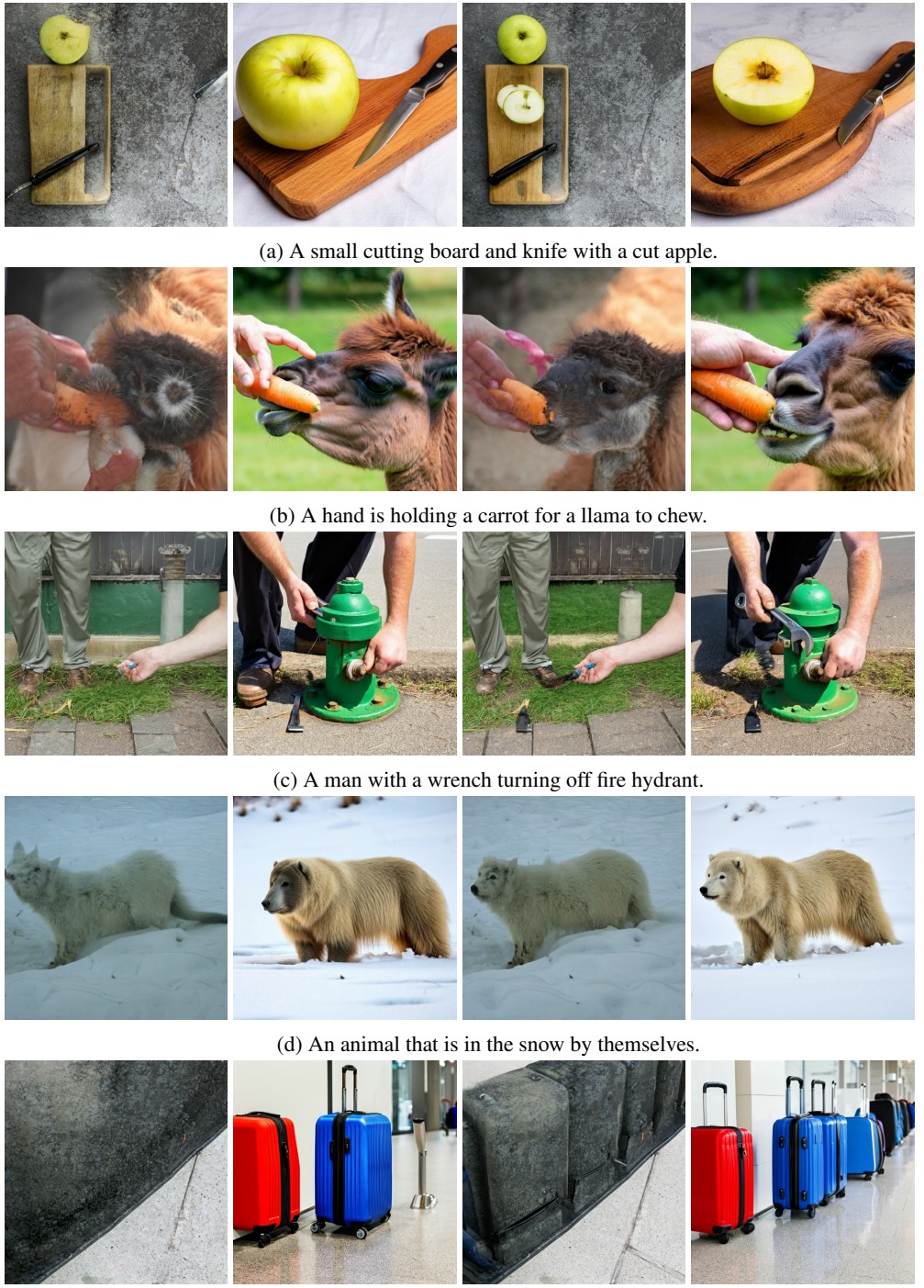

(a) A small cutting board and knife with a cut apple.

(b) A hand is holding a carrot for a llama to chew.

(c) A man with a wrench turning off fire hydrant.

(d) An animal that is in the snow by themselves.

(e) A row of red and blue luggage sitting at an airport.

Figure 6: **Additional T2I Results on MS-COCO. Part 3**. (left-to-right) We provide results of images generated from the given text prompt without CFG, with CFG $\omega = 7.5$, our method with self-consistency loss and our method with self-consistency loss and CLIP score reward.

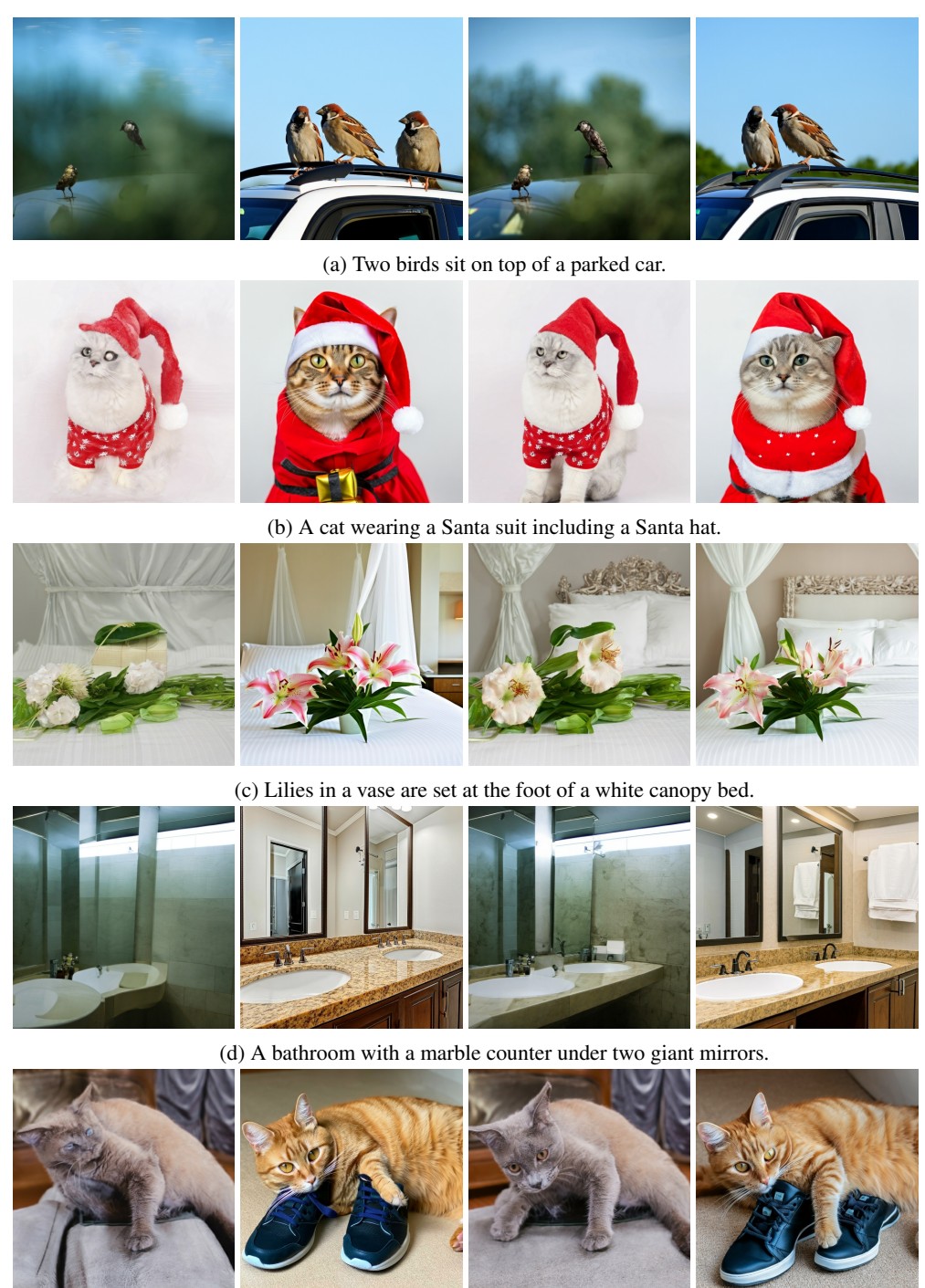

(a) Two birds sit on top of a parked car.

(b) A cat wearing a Santa suit including a Santa hat.

(c) Lilies in a vase are set at the foot of a white canopy bed.

(d) A bathroom with a marble counter under two giant mirrors.

(e) A cat laying on top of shoes on floor.

Figure 7: **Additional T2I Results on MS-COCO. Part 4**. (left-to-right) We provide results of images generated from the given text prompt without CFG, with CFG $\omega = 7.5$, our method with self-consistency loss and our method with self-consistency loss and CLIP score reward.

---

**Algorithm 2** CFG version of DDIM (sampling)

---

**Require:** $\{t_k\}_{k=0}^N$ with $t_0 = 0 < \cdots < t_N = 1$, churn parameter $\varepsilon$, conditioning $c$, guiding weights $\boldsymbol{\omega} = (\omega_{c,(s,t)})$, unconditional and conditional pretrained denoisers $\hat{x}_\theta(x_t, \varnothing)$, $\hat{x}_\theta(x_t, c)$

    Sample $x_{t_N} \sim \mathcal{N}(0, \mathrm{Id})$

    **for** $k \in \{N-1, \ldots, 0\}$ **do**

        Sample $z \sim \mathcal{N}(0, \mathrm{Id})$

        Set current guidance weight $\omega = \omega_{c,(t_k, t_{k+1})}$

        Compute $\hat{x}_0 = \hat{x}_\theta(t_{k+1}, x_{t_{k+1}}, c; \omega)$ using (7)

        Compute $\mu_{t_k, t_{k+1}}(\hat{x}_0, x_{t_{k+1}}), \Sigma_{t_k, t_{k+1}}$ using (23)

        Sample $x_{t_k} \sim \mathcal{N}(\mu_{t_k, t_{k+1}}(\hat{x}_0, x_{t_{k+1}}), \Sigma_{t_k, t_{k+1}})$

    **Return:** $x_0$

---

By equality of rhs of (26) and of (25), we have $\forall x_t \sim p_{t|0}(x_t|x_0), c \sim p(c), x_0 \sim p_{0|c}(\cdot|c), t \in [0,1]$,

$$\hat{x}_\theta(x_t, c; \omega_{c,t}^\phi) - \hat{x}_\theta(x_t, c) = \omega_{c,t}^\phi (\hat{x}_\theta(x_t, c) - \hat{x}_\theta(x_t, \varnothing)) \approx 0$$

We then assume that $\Delta_\theta(x_t, c) = \hat{x}_\theta(x_t, c) - \hat{x}_\theta(x_t, \varnothing) \neq 0$. It is a reasonable assumption since if $\Delta_\theta(x_t, c) = 0$, the CFG would not have had any effect. This implies that

$$\omega_{c,t}^\phi = 0, \quad \forall c \sim p(c), t \in [0,1] \tag{27}$$

The only regime where (27) will not hold is when

$$\hat{x}_\theta(x_t, c) \napprox \mathbb{E}[x_0|x_t, c]$$

In such a case, one could still find guidance weights for (25) to hold. But this is a regime of a very under-trained model. In our image experiments (see Section 5), we initially tried using this approach and found that it was leading to zero guidance weights (27).

**Connection to backward** KL. Another explanation for the behavior above is that as we train (24), we only observe $x_t \sim p(x_t|x_0)$. The form of (24) comes from optimizing backward KL

$$\mathrm{KL}[p_{[0,1]} || p_{[0,1]}^{\theta, \boldsymbol{\omega}}] = \int_0^1 \int \int ||x_0 - \hat{x}_\theta(x_t, c; \boldsymbol{\omega})||^2 p(x_t|x_0) p(x_0|c) \mathrm{d}x_0 \mathrm{d}x_t \mathrm{d}t, \tag{28}$$

where $p_{[0,1]}^{\theta, \boldsymbol{\omega}}$ is the distribution of all the trajectories obtained by running guided backwards diffusion, and $p_{[0,1]}$ is the distribution of all the trajectories from the true diffusion process.

When we use CFG during sampling, we actually may observe noisy samples $\hat{x}_t$ which do not coincide with any $x_t \sim p_{t|0}$ form the noising process (3). This hints that instead of backwards KL (28), we should actually aim to optimize the forward KL

$$\mathrm{KL}[p_{[0,1]}^{\theta, \boldsymbol{\omega}} || p_{[0,1]}]. \tag{29}$$

An approach based on optimizing (29) was considered in Azangulov et al. (2025). This approach, however, is very computationally expensive since it requires sampling full guided trajectory from the guided backwards process and then compute the gradients on it.

## C.2 CONDITIONAL CONSISTENCY

Here, we describe an approach similar to *marginal-consistency* (14), but which however could be more effective in practice in principle due to lower variance. We fix conditioning $c$ and we denote by

$$p_{s|0,c}^{t,(\theta, \boldsymbol{\omega})}(x_s|x_0, c) = \int p_{s|t,c}^{(\theta, \boldsymbol{\omega})}(x_s|x_t, c) p_{t|0}(x_t|x_0) \mathrm{d}x_t,$$

which is the term under integral in (12) depending on $(x_0, c)$. We now consider a *conditional consistency* condition where we fix $c$ and marginalize over $x_0$

$$\int p_{s|0,c}^{t,(\theta, \boldsymbol{\omega})}(x_s|x_0, c) p_{0|c}(x_0|c) \mathrm{d}x_0 = p_{s|c}^{t,(\theta, \boldsymbol{\omega})}(x_s|c) \approx p_{s|c}(x_s|c) = \int p_{s|0}(x_s|x_0) p_{0|c}(x_0|c) \mathrm{d}x_0.$$

The main difference with (14) is that we fixed conditioning $c$. We could use a similar loss

$$\mathcal{L}_c(\boldsymbol{\omega}) = \mathbb{E}_{c \sim p_c, (s,t) \sim p(s,t)}[\text{MMD}_{(\beta, \lambda)}(p_{s|c}^{t,(\theta, \boldsymbol{\omega})}(\cdot|c), p_{s|c}(\cdot|c))],$$

which can be optimized by drawing conditional samples $x_0|c$. While this approach is expected to have a lower variance than (15), since we do not need to sample $c$, it requires to be able to produce conditional samples which is not always available in practice during training (i.e., in text-to-image applications). In practice, we also found that it did not lead to good results. We hypothesize that it is due to still quite a high variance of the gradients of $\mathcal{L}_c(\boldsymbol{\omega})$.

## D   L2 APPROACH FOR LEARNING GUIDANCE

We describe here the algorithm based on the $\ell_2$ objective function (21)

$$\mathcal{L}_{\ell_2}(\boldsymbol{\omega}) = \mathbb{E}_{(x_0,c) \sim p_{0,c}, s, t \sim p(s,t)} \left[ \mathbb{E}\left[||\tilde{x}_s(\boldsymbol{\omega}) - x_s||_2^2\right]\right]$$

$$= \mathbb{E}_{(x_0,c) \sim p_{0,c}, s, t \sim p(s,t)} \left[ ||\mathbb{E}[\tilde{x}_s(\boldsymbol{\omega})] - \mathbb{E}[x_s]||_2^2 \right.$$
$$\left. + \text{Trace}(\text{Cov}(\tilde{x}_s(\boldsymbol{\omega}))) + \text{Trace}(\text{Cov}(x_s)) - 2\text{Trace}(\text{Cov}(x_s, \tilde{x}_s(\boldsymbol{\omega})))\right]$$

So, by minimizing this loss function, we not only attempt to match on averages the means of $\tilde{x}_s(\boldsymbol{\omega})$ while minimizing the expected sum of variances of the component of $\tilde{x}_s(\boldsymbol{\omega})$ and maximizing the sum of the covariance terms between $x_s$ and $\tilde{x}_s(\boldsymbol{\omega})$.

**Empirical objective.**   We sample $\{x_0^i, c^i\}_{i=1}^n \overset{\text{i.i.d.}}{\sim} p_{0,c}$ from the training set and we additionally sample noise levels $\{s_i\}_{i=1}^n \overset{\text{i.i.d.}}{\sim} \mathcal{U}[S_{\min}, 1-\zeta-\delta]$, as well as time increments $\{\Delta t_i\}_{i=1}^n \overset{\text{i.i.d.}}{\sim} \mathcal{U}[\delta, 1-\zeta-s_i]$ and we let $t_i = s_i + \Delta t_i$. For each $(x_0^i, s_i)$, we sample $x_{s_i} \sim p_{s_i|0}(\cdot|x_0^i)$ from the *noising process* (3), which defines the target samples. We then also produce $\tilde{x}_{s_i}(\boldsymbol{\omega}^\phi) \sim p_{s_i|0,c_i}^{t_i,(\theta,\boldsymbol{\omega}^\phi)}(\cdot|x_0^i, c_i)$ by first sampling $\tilde{x}_{t_i} \overset{\text{i.i.d.}}{\sim} p_{t_i|0}(\cdot|x_0^i)$ from the *noising process* (3) and then denoising with guidance and DDIM using (19). This defines the proposal samples. We expand loss function (21) as a function of guidance network parameters $\phi$ defined on the empirical batches as follows

$$\hat{\mathcal{L}}_{\ell_2}(\phi) = \frac{1}{n} \sum_{i=1}^n \left[||\tilde{x}_{s_i}(\boldsymbol{\omega}^\phi) - x_{s_i}||_2^2\right],$$

The full algorithm is given in Algorithm 3.

---

**Algorithm 3** Learning to Guide with $\ell_2$ loss

---

1: **Input:** Init. guidance parameters $\phi$; (frozen) denoiser $\hat{x}_\theta$; data distribution $p_0$; learning rate $\eta$; $\zeta > 0$, $S_{\min} > 0$, $\delta > 0$, b.s. $n$, DDIM churn $\varepsilon \in [0,1]$.
2: **repeat**
3:      Sample batch of clean data and their conditionings $\{x_0^i, c^i\}_{i=1}^n \overset{\text{i.i.d.}}{\sim} p_{0,c}$.
4:      Sample $\{s_i\}_{i=1}^n \overset{\text{i.i.d.}}{\sim} \mathcal{U}[S_{\min}, 1-\zeta-\delta]$, $\{\Delta t_i\}_{i=1}^n \overset{\text{i.i.d.}}{\sim} \mathcal{U}[\delta, 1-\zeta-s_i]$, let $t_i = s_i + \Delta t_i$
5:      (**True process**) Sample $x_{s_i} \sim p_{s_i|0}(\cdot|x_0^i)$ from noising process (3)
6:      (**Guided process**) Sample $\tilde{x}_{t_i} \sim p_{t_i|0}(\cdot|x_0^i)$ from noising process (3)
7:      Compute guidance weights $\omega_i = \omega_{c_i,(s_i,t_i)}^\phi$ and $\hat{x}_\theta(\tilde{x}_{t_i}, c_i; \omega_i)$ using (7)
8:      Sample $\tilde{x}_{s_i}(\boldsymbol{\omega}^\phi) \sim p_{s_i|t_i,0}(\cdot|\tilde{x}_{t_i}, \hat{x}_\theta(\tilde{x}_{t_i}, c_i; \omega_i))$ from DDIM (2) with *churn* parameter $\varepsilon$
9:      (**Loss**) Compute loss
10:      $\hat{\mathcal{L}}_{\ell_2}(\phi) = \frac{1}{n} \sum_{i=1}^n \left[||\tilde{x}_{s_i}(\boldsymbol{\omega}^\phi) - x_{s_i}||_2^2\right]$
11:      Update $\phi \leftarrow \phi - \eta \nabla_\phi \hat{\mathcal{L}}_{\beta,\lambda}(\phi)$
12: **until** convergence
13: **Output:** Optimized guidance network parameters $\phi$

---

# E LEARNING TO GUIDE WITH REWARDS

In this section, we describe how our guidance learning approach is extended to a setting, where a reward function $R(x_0, c)$ is available, see Algorithm 4 below.

---

**Algorithm 4** Learning to Guide with reward

---

1: **Input:** Init. guidance parameters $\phi$; (frozen) denoiser $\hat{x}_\theta$; data distribution $p_0$; learning rate $\eta$; $\zeta > 0, S_{\min} > 0, \delta > 0$, b.s. $n$, n. of particles $m$, $\lambda \in [0, 1], \beta \in [0, 2]$, DDIM churn $\varepsilon \in [0, 1]$. Reward function $R(x_0, c)$, reward loss weight $\gamma_R \geq 0$.

2: **repeat**

3:     Sample batch of clean data and their conditionings $\{x_0^i, c^i\}_{i=1}^n \overset{\text{i.i.d.}}{\sim} p_{0,c}$.

4:     Sample $\{s_i\}_{i=1}^n \overset{\text{i.i.d.}}{\sim} \mathcal{U}[S_{\min}, 1 - \zeta - \delta]$, $\{\Delta t_i\}_{i=1}^n \overset{\text{i.i.d.}}{\sim} \mathcal{U}[\delta, 1 - \zeta - s_i]$, let $t_i = s_i + \Delta t_i$

5:     (**True process**) Sample $m$ particles $\{x_{s_i}^j\}_{j=1}^m \overset{\text{i.i.d.}}{\sim} p_{s_i|0}(\cdot|x_0^i)$ from noising process (3)

6:     (**Guided process**) Sample $m$ particles $\{\tilde{x}_{t_i}^j\}_{j=1}^m \sim p_{t_i|0}(\cdot|x_0^i)$ from noising process (3)

7:     Compute guidance weights $\omega_i = \omega_{c_i,(s_i,t_i)}^\phi$ and $\hat{x}_\theta(\tilde{x}_{t_i}^j, c_i; \omega_i)$ using (7)

8:     Sample $\tilde{x}_{s_i}^j(\boldsymbol{\omega}^\phi) \sim p_{s_i|t_i,0}(\cdot|\tilde{x}_{t_i}^j, \hat{x}_\theta(\tilde{x}_{t_i}^j, c_i; \omega_i))$ from DDIM (2) with *churn* parameter $\varepsilon$

9:     (**Loss**) Compute loss

10:    $\hat{\mathcal{L}}_{\beta,\lambda}(\phi) = \frac{1}{n} \sum_{i=1}^n \left[ \frac{1}{m^2} \sum_{j,k=1}^m ||\tilde{x}_{s_i}^j(\boldsymbol{\omega}^\phi) - x_{s_i}^k||_2^\beta - \frac{\lambda}{2} \frac{1}{m(m-1)} \sum_{j \neq k} ||\tilde{x}_{s_i}^j(\boldsymbol{\omega}^\phi) - \tilde{x}_{s_i}^k(\boldsymbol{\omega}^\phi)||_2^\beta \right]$

11:    (**Reward loss**) Compute reward loss

12:    $\hat{\mathcal{L}}_R(\phi) = \frac{1}{nm} \sum_{i=1}^n \sum_{j=1}^m R(\hat{x}_\theta(\tilde{x}_{t_i}^j, c_i; \omega_i), c_i)$

13:    (**Total loss**) Compute total loss

14:    $\tilde{\mathcal{L}}_{tot}(\phi) = \hat{\mathcal{L}}_{\beta,\lambda}(\phi) + \gamma_R \hat{\mathcal{L}}_R(\phi)$

15:    Update $\phi \leftarrow \phi - \eta \nabla_\phi \tilde{\mathcal{L}}_{tot}(\phi$

16: **until** convergence

17: **Output:** Optimized guidance network parameters $\phi$

---

# F EXPERIMENTAL DETAILS

**Noise process.**   For all the experiments, we use recitifed flow noise process (1) where $\alpha_t = (1 - t)$ and $\sigma_t = t$.

**Guidance network architecture for MoG, ImageNet and CelebA.**   For guidance network $\omega(s, t, c; \phi) = \omega_{(c,(s,t))}^\phi$ we use a 6 layers MLP with hidden size of $64$ followed by an output layer transforming it to a dimension $1$. As an activation we use "GeLU". We also apply a "ReLU" activation after the last layer of the guidance network to ensure that the gudiance weights are non-negative. For 2D experiments, we do not use ReLU activation and allow the weights to be negative.

The time which is processed by the weight network is first converted to log SNR, i.e. $s \to \log \text{SNR}(s)$ and $t \to \log \text{SNR}(t)$. We do not use sinusoidal embedding as typically is done with diffusion models, since we found that it worked significantly worse. Then the two time-steps are concatenated and passed through to a 2-layer MLP with hidden dimension $256$ and an output dimension $512$ and GeLU activation. We also use a dropout with rate $0.3$ in the middle of this MLP.

**2D MoG.**   We use $4$ layer MLP with hidden dimension $64$ and GeLU activation for the denoiser network. The time $t$ for the backbone, is first embedded into log SNR and then we use sinusoidal embedding dimension $128$. We use adam optimizer with learning rate $10^{-4}$ and batch size $128$. We use maximum norm clipping for Adam to be $1$. When we do pretraining in the *well-trained* regime, we use $10k$ iterations, and when we do pretraining in the *under-trained* regime, we use $250$ iterations. We train guidance network for $1000$ iterations, with learning rate $5e-4$, norm clipping $1$ and Adam optimizer. We use $m = 32$ particles. We use batch size $128$. We use backwards DDIM (23) with $\varepsilon = 1$ in order to sample $x_s$ with guidance (19). To produce Fig. 15 and Fig. 16, we sample data with velocity sampler with $10$ steps. We use $4096$ samples in total.

**ImageNet.**   We pretrain the U-Net model (Ronneberger et al., 2015) similar to DDPM (Ho et al., 2020) which has channel dimension 192, 3 residual blocks, channel multiplier (1, 2, 3, 4), attention which is applied with (False, False, True, False) residual blocks (whenever it is True, we use attention), dropout which is applied at (False, True, True, True) residual blocks (whenever it is True, we use dropout). The dropout rate is 0.1. We also use sigmoid weighting (Kingma and Gao, 2023) with bias $b = 2$. The model predicts the velocity and the loss is based on velocity. We train it with batch size 128 for $7M$ iterations. We use Adam optimizer with learning rate $1e - 5$ and norm clipping 1. We use EMA decay of 0.9999.

For training guidance we reload the pretrained diffusion model with EMA parameters and freeze it. We train the guidance network for $100K$ iterations with batch size $B = 256$. We use $m = 4$ number of particles and $\beta = 1.75, \lambda = 1$ parameters. We use learning rate $1e - 5$ with adam optimizer and norm clipping 1. We use EMA decay of 0.9999. For self-consistency loss (20), in order to produce $\tilde{x}_s(\omega)$ in (19), we use DDIM sampler (23) with $\varepsilon = 1$. We sweep over $\delta \in [0.01, 0.1, 0.2, 0.3]$, $S_{\min} \in [0.01, 0.1, 0.2, 0.3]$ parameters. We use $\zeta = 1e - 2$ which is the same parameter as the safety parameter for the original diffusion model. This safety parameter in the original diffusion model just controls the time interval to be $(\zeta, 1 - \zeta)$ instead of $(0, 1)$, for numerical stability. To sample data, we use DDIM (19) with $\varepsilon = 0$ for all the experiments. During training, we track FID on a small subset of 2048 images. We observed that the performance over time could be unstable for some hyperparameters, therefore we use this training-time FID to keep the best checkpoint for every hyperparameter. We report the performance for the best hyperparameters based on FID, we use $50k$ samples for FID computation. On top of that, we also train the simplified $\ell_2$ method using Algorithm 3 using the same methodology, except that in order to produce $\tilde{x}_s(\omega)$ in (19), we use DDIM sampler (23) but with $\varepsilon = 0$, since it led to a better empirical performance.

**CelebA.**   We pretrain the U-Net model (Ronneberger et al., 2015) similar to DDPM (Ho et al., 2020) which has channel dimension 256, 2 residual blocks, channel multiplier (1, 2, 2, 2), attention which is applied with (False, True, False, False) residual blocks (whenever it is True, we use attention), dropout which is applied at (False, True, True, True) residual blocks (whenever it is True, we use dropout). The dropout rate is 0.2. We also use sigmoid weighting (Kingma and Gao, 2023) with bias $b = 1$. We do not transform time to $\log$ SNR for time embedding in the diffusion network. The model predicts the $x_0$ and the loss is based on velocity. We train it with batch size 128 for $300K$ iterations. We use Adam optimizer with learning rate $1e - 4$ and norm clipping 1. We use EMA decay of 0.9999.

For training guidance we reload the pretrained diffusion model with EMA parameters and freeze it. We train the guidance network for $100K$ iterations with batch size $B = 256$. We use $m = 4$ number of particles and $\beta = 1.75, \lambda = 1$ parameters. We use learning rate $1e - 5$ with adam optimizer and norm clipping 1. We use EMA decay of 0.9999. For self-consistency loss (20), in order to produce $\tilde{x}_s(\omega)$ in (19), we use DDIM sampler (23) with $\varepsilon = 1$. We sweep over $\delta \in [0.01, 0.1, 0.2, 0.3]$, $S_{\min} \in [0.01, 0.1, 0.2, 0.3]$ parameters. We use $\zeta = 1e - 2$ which is the same parameter as the safety parameter for the original diffusion model. This safety parameter in the original diffusion model just controls the time interval to be $(\zeta, 1 - \zeta)$ instead of $(0, 1)$, for numerical stability. To sample data, we use DDIM with $\varepsilon = 0$ for all the experiments. During training, we track FID on a small subset of 2048 images. We observed that the performance over time could be unstable for some hyperparameters, therefore we use this training-time FID to keep the best checkpoint for every hyperparameter. We report the performance for the best hyperparameters based on FID, we use $50k$ samples for FID computation. On top of that, we also train the simplified $\ell_2$ method using Algorithm 3 using the same methodology, except that in order to produce $\tilde{x}_s(\omega)$ in (19), we use DDIM sampler (23) but with $\varepsilon = 0$, since it led to a better empirical performance.

**Guidance network architecture for MS-COCO.**   For guidance network $\omega(s, t, c_{\text{CLIP}}, c_{\text{T5}}; \phi) = \omega^\phi_{c_{\text{CLIP}}, c_{\text{T5}}, (s,t)}$ we use a 6 layers MLP with hidden size of 512 followed by an output layer transforming it to a dimension 1. As an activation we use "GeLU". We also apply a "ReLU" activation after the last layer of the guidance network to ensure that the guidance weights are non-negative.

The time steps $t$ and $s$ are each processed by separate networks. We do not use sinusoidal embedding to get the time embeddings. We also do not convert time to $\log$ SNR as we did for ImageNet and CelebA. Each of the two time-steps are passed through to a 2-layer MLP with hidden dimension and output dimension of 256 and a SiLU activation. The two timestep embeddings are then concatenated.

We also pass CLIP and T5 embeddings through a similar MLP with an output dimension of 256 and 512 respectively. Both time embedding and CLIP embedding are concatenated to a dimension 512 and then concatenated with the text embedding. This is passed through the MLP layers described above.

**MS-COCO.** We pretrain a 1.05B parameter Flux text-to-image backbone (BlackForestLabs, 2025), which uses an architecture based on Esser et al. (2024) and consists of 30 Diffusion Transformer (DiT) blocks (Peebles and Xie, 2023) and 15 MMDiT blocks (Esser et al., 2024). We do not use any weighting, nor do we transform time with logSNR. The model is trained with the standard flow matching loss and predicts velocity. The CLIP embeddings used for training the model are extracted from the standard ViT-L/14 CLIP transformer architecture. The T5 embeddings are extracted from the standard T5-XXL architecture.

We load the pretrained flow matching model with EMA parameters and freeze it to train the guidance network. We train the guidance network for $50K$ iterations with a batch size of 256. We set the number of particles as $m = 4$, $\beta = 1.75$ and $\lambda = 1.0$. We use learning rate of $8e - 5$ with Adam optimizer. We set the Adam parameters $\beta = (0.9, 0.999)$, $\epsilon = 1e - 8$, and weight decay=0.01. For self-consistency loss (20), in order to produce $\tilde{x}_s(\boldsymbol{\omega})$ in (19), we use DDIM sampler (23) with $\varepsilon = 0$. We set $\delta = 0.2$, $S_{\min} = 0.01$ and safety parameter $\zeta = 1e - 2$ while training the model. To sample data from flow matching model, we use the standard Euler ODE solve in the time interval $[0, 1]$ and use 128 sampling steps. To select the best checkpoint, we tracked CLIP Score over $3K$ samples and then used the checkpoint corresponding to the best CLIP Score on this subset to compute CLIP Score over $10K$ samples.

## G    TRAINING AND INFERENCE COST OF GUIDANCE NETWORK

Training the guidance network involves training a lightweight MLP with a few layers. The cost of training it is insignificant compared to the cost of training a diffusion model. For example, in ImageNet experiments, we train a U-Net with $\sim 260$ M parameters for around 7M iterations with a batch size of 128, while the guidance network with $\sim 600$ K parameters is trained for up to 100 K with batch size 256. In text-to-image, the model has about 1B parameters and is trained for 1.6M steps with batch size of 2048, while guidance network has about 6M parameters which we train for 50K iterations with batch size 256.

At inference, we use a guidance network to produce a scalar $\omega$ which involves evaluating the MLP on precomputed $(t, s)$ and conditioning embeddings $c$. The computation cost is minimal compared to unrolling the diffusion model.

## H    ABLATIONS

In this section, we provide ablations of our method. We provide an ablation over $\beta$ and $m$ on ImageNet $64 \times 64$ in Appendix H.1. In Appendix H.2, we provide ablation of the conditioning signal in T2I signal, and in Appendix H.3, we demonstrate the impact of guidance network size and architecture on the performance.

### H.1    ABLATION OVER $\beta$ AND $m$ FOR SELF-CONSISTENCY LOSS

We ran an ablation study over $\beta$, the exponent applied to the $L_2$ norm in 20, and $m$, the number of particles used for the MMD computation for the self-consistency loss in 22. The ablation was conducted on ImageNet $64 \times 64$. The results are given in Table 4 and in Table 5. We see that the performance is not very sensitive to $m$, we see that $m >= 4$ works well. As for $\beta$, we see that values $\beta \in [1.1.75]$ lead to good results.

### H.2    CHOICE OF MLP CONDITIONING ON T2I

We investigate the effects of the choice of MLP conditioning for T2I task with our method. We find that T5-only embeddings gets slightly lower CLIP score than the other two conditioning configuration which is CLIP-only, and both T5 and CLIP, but it gets slightly better FID score. We provide qualitative

Table 4: Ablation over $\beta$ on ImageNet $64 \times 64$

| $\beta$ | FID | IS |
|---|---|---|
| 0.1 | 2.07 | 78.18 |
| 0.5 | 2.04 | 77.72 |
| 1.0 | 1.98 | 73.33 |
| 1.5 | 2.00 | 71.17 |
| 1.75 | 1.99 | 73.69 |

Table 5: Ablation over $m$ on ImageNet $64 \times 64$

| m | FID | IS |
|---|---|---|
| 2 | 2.00 | 70.66 |
| 4 | 1.99 | 73.69 |
| 8 | 1.99 | 73.02 |
| 16 | 2.00 | 77.11 |

results in Fig. 8 and quantitative results in Table 6. All the experiments use the same hyperparameters for the model architecture and training. Qualitatively, we find that in most of the cases, the images look very similar, however, in some cases, CLIP conditioning is helpful to get better text-aligned images.

Table 6: **MS COCO** $512 \times 512$. FID and CLIP score for different MLP condtioning (the best are in **bold**).

| Method Name | Guidance Weight | FID $\downarrow$ | CLIP Score $\uparrow$ |
|---|---|---|---|
| Self-consistency (20) | $\omega^{\phi}_{c_{\mathrm{CLIP}}, c_{\mathrm{T5}, (s,t)}}$ | **18.01** | 0.295 |
| Self-consistency (20) + CLIP Score reward | $\omega^{\phi}_{c_{\mathrm{CLIP}}, c_{\mathrm{T5}, (s,t)}}$ | 28.37 | **0.306** |
| Self-consistency (20) + CLIP Score reward | $\omega^{\phi}_{c_{\mathrm{CLIP}}, (s,t)}$ | 28.63 | **0.306** |
| Self-consistency (20) + CLIP Score reward | $\omega^{\phi}_{c_{\mathrm{T5}, (s,t)}}$ | 28.28 | 0.305 |

### H.3 GUIDANCE NETWORK SIZE AND ARCHITECTURE

Using T2I as testbed, we ran two ablations to explore design choices of MLP. First, we increased the size of the Dense layers in the MLP from 128 to 1024. This larger MLP has 12M parameters and we refer to this MLP as MLP-Wide. Second, we also introduced residual connections in the above MLP. We refer to this MLP architecture as Residual MLP-Wide. We summarize the results below in Table 7. We do not observe significant improvements in metrics upon increasing the number of parameters.

## I ADDITIONAL RESULTS

In this section, we provide additional experimental results and comparisons. We add a comparison to clamp-linear schedule (Wang et al., 2024) on ImageNet $64 \times 64$ in Appendix I.1. Moreover, in Appendix I.2, we compare our approach to limited interval guidance and clamp linear schedule in T2I regime. Furthermore, we add an additional experiment using Stable-Diffusion-v1.5 (Rombach et al., 2022b) model in Appendix I.3. Finally, we report 2-d results in Appendix I.4.

### I.1 COMPARISON TO CLAMP-LINEAR ON IMAGENET $64 \times 64$

We compare our method to the clamp-linear schedule (Wang et al., 2024), $w_t = \max(c, \frac{w_0 2t}{T})$, on ImageNet $64 \times 64$. We sweep over parameter $w_0 \in \{0.05, 0.1, 0.15, 0.2, 0.25, 0.3, 0.35, 0.4, 0.45, 0.5, 0.55, 0.6, 0.65, 0.7, 0.75, 0.8, 0.85, 0.9, 0.95, 1.0\}$

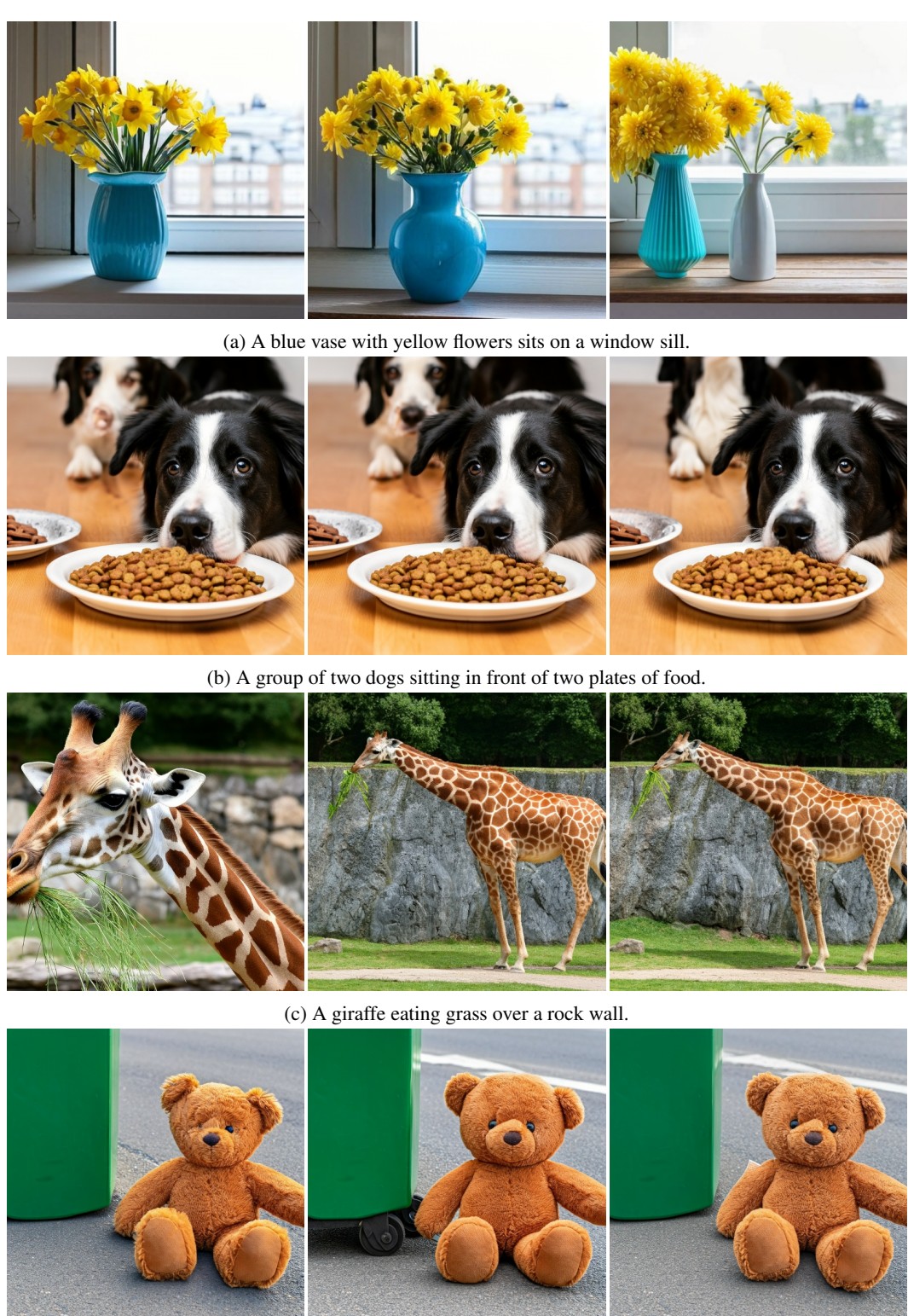

(a) A blue vase with yellow flowers sits on a window sill.

(b) A group of two dogs sitting in front of two plates of food.

(c) A giraffe eating grass over a rock wall.

(d) Stuffed teddy bear sitting next to garbage can on the side of the road.

Figure 8: **Choice of MLP conditioning**. **(left-to-right)** Images generated with our method with (left) both T5 and CLIP conditioning, (middle) CLIP conditioning, (right) T5 conditioning in the MLP. In some cases, we find that CLIP only conditioning is slightly better than having both T5 and CLIP conditioning, or T5-only conditioning.

| MLP | Dense layer size | Network size | FID 10K | CLIP 10K |
|---|---|---|---|---|
| MLP-Small | 128 | 6M | 28.37 | 0.306 |
| MLP-Wide | 1024 | 12M | 28.41 | 0.306 |
| Residual MLP-Wide | 1024 | 12M | 28.68 | 0.306 |

Table 7: Comparison of MLP model architectures and performance.

and $w_0 \in \{1.05, 1.1, 1.2, 1.3, 1.5, 2.0, 2.5, 3.0, 3.5, 4.0, 4.5, 5.0\}$, as well as over clamp parameter $c \in \{0.05, 0.1, 0.15, 0.2, 0.25, 0.3, 0.35, 0.4, 0.45, 0.5, 0.55, 1.0, 1.5, 2.0, 2.5\}$.

We report the results in a table below. Our method is the same as the one reported in Table 1 i.e., self consistency with all the conditioning information. We also report performance of constant guidance and limited-interval guidance. Results are given in Table 8. We observe that clamp-linear works worse than our method and baselines.

Table 8: Comparison to different methods on ImageNet $64 \times 64$

| Method | FID | IS |
|---|---|---|
| Ours | 1.99 | 73.62 |
| Clamp-linear | 2.24 | 76.34 |
| LIG | 2.11 | 71.60 |
| Constant guidance | 2.40 | 66.72 |

## I.2 COMPARISON AGAINST BASELINE METHODS ON MS COCO $512 \times 512$

We compare our method against classifier-free guidance (Ho and Salimans, 2022), limited interval guidance (Kynkäänniemi et al., 2024), and clamp-linear schedule (Wang et al., 2024). For each of these baseline methods, we perform extensive hyperparameter tuning as shown in as shown in Table 10 and Table 9 and select images from the hyperparameter setting that results in the best metrics in terms of CLIP Score. We find that our method has comparable or better images compared to these baselines in many cases as shown in Figs. 9 to 13. Along with the images, we also plot the guidance weight schedules obtained from our model and the baseline methods. We find that our guidance weight schedule can be similar to those of clamp-linear schedule in some cases such as Figs. 9 and 13. Quantitatively, we find that both the methods perform worse compared to our method and classifier-free guidance in terms of CLIP score.

Table 9: **MS COCO** $512 \times 512$. FID and CLIP score for limited interval guidance (Kynkäänniemi et al., 2024) (the best are in **bold**).

| Guidance Weight | Guidance Interval | FID ↓ | CLIP Score ↑ |
|---|---|---|---|
| 7.5 | $[0.1, 0.9]$ | 23.86 | 0.3033 |
| 7.5 | $[0.1, 0.2]$ | 24.09 | 0.3033 |
| 7.5 | $[0.2, 0.9]$ | **18.89** | 0.2979 |
| 7.5 | $[0.2, 0.8]$ | 18.96 | 0.2979 |
| 15.0 | $[0.1, 0.9]$ | 24.95 | **0.3042** |
| 15.0 | $[0.1, 0.8]$ | 25.10 | 0.3041 |
| 15.0 | $[0.2, 0.9]$ | 20.58 | 0.2992 |
| 15.0 | $[0.2, 0.8]$ | 20.57 | 0.2992 |

Table 10: **MS COCO** $512 \times 512$. FID and CLIP score for clamp-linear guidance schedule $w_t = \max(c_0, \frac{2w_0 t}{T})$ (Wang et al., 2024) (the best are in **bold**).

| Guidance Weight | $c_0$ | FID $\downarrow$ | CLIP Score $\uparrow$ |
|---|---|---|---|
| 7.5 | 1.0 | **21.99** | 0.3033 |
| 10 | 1.0 | 23.17 | 0.3039 |
| 14.0 | 1.0 | 24.15 | 0.3045 |
| 14.0 | 2.0 | 24.75 | 0.3049 |
| 14.0 | 4.0 | 27.14 | **0.3055** |
| 16.0 | 1.0 | 24.55 | 0.3047 |

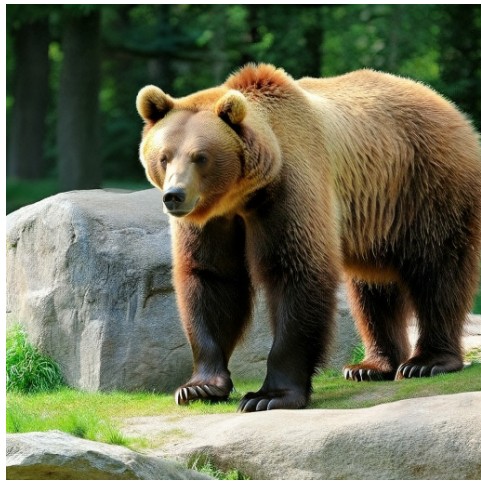
(a) CFG $w = 7.5$

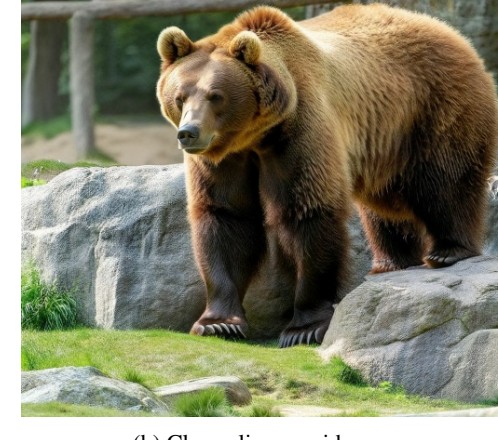
(b) Clamp-linear guidance

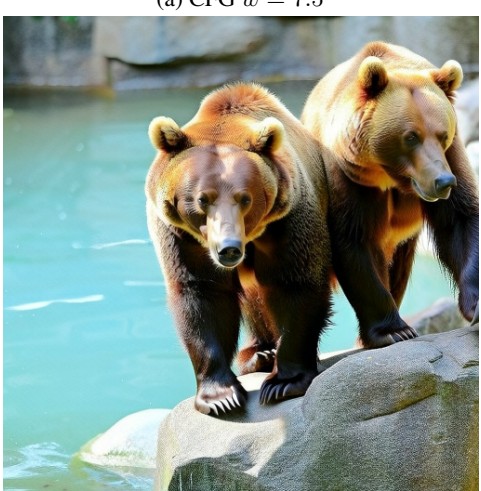
(c) Limited interval guidance

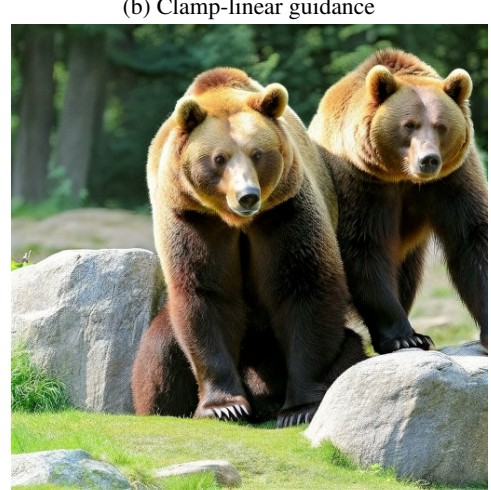
(d) Ours

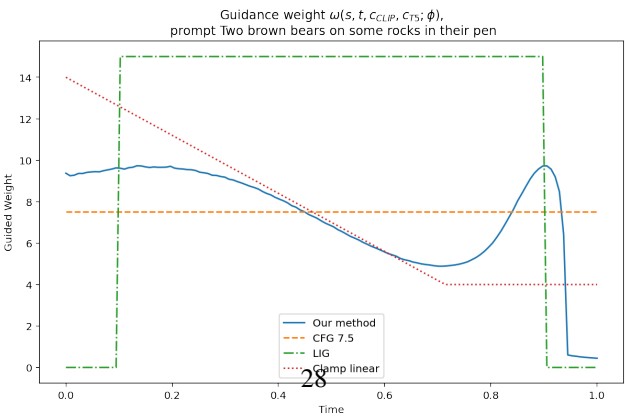
(e) Plot of Guidance weights

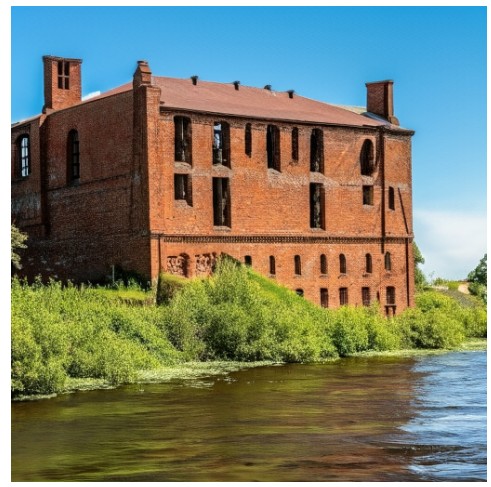

(a) CFG $w = 7.5$

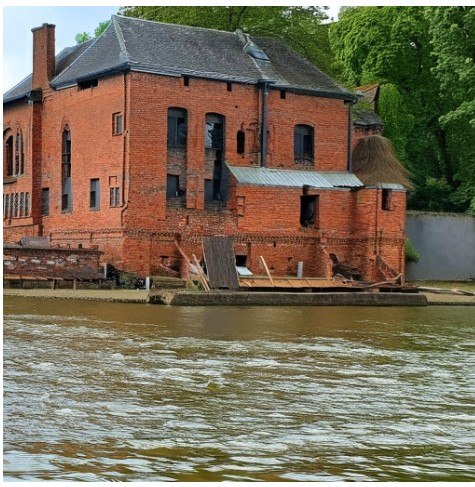

(b) Clamp-linear guidance

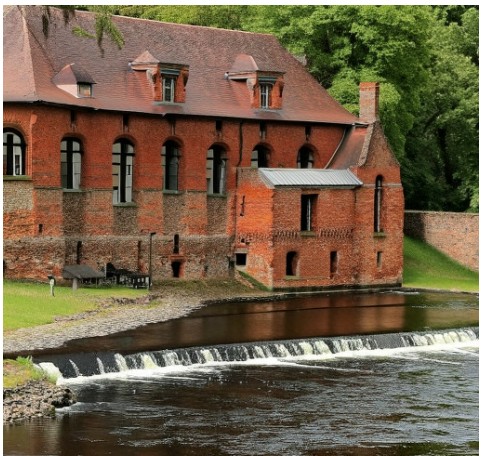

(c) Limited interval guidance

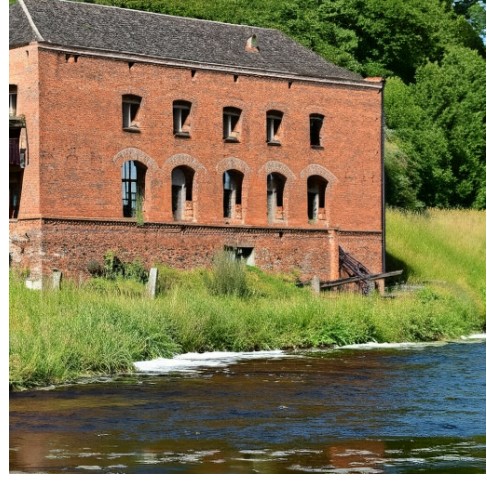

(d) Ours

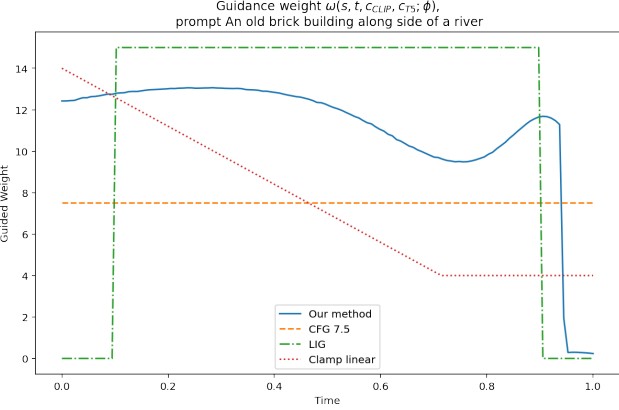

(e) Plot of Guidance weights

(f) **Text prompt**: An old brick building along side of a river.

Figure 10: **T2I Results on MS-COCO..** We provide results of images generated from the given text prompt with **(first row - left)** CFG $\omega = 7.5$, **(first row - right)** clamp linear, **(second row -left)** limited interval, and **(second row - right)** our method with self-consistency loss and CLIP score reward.

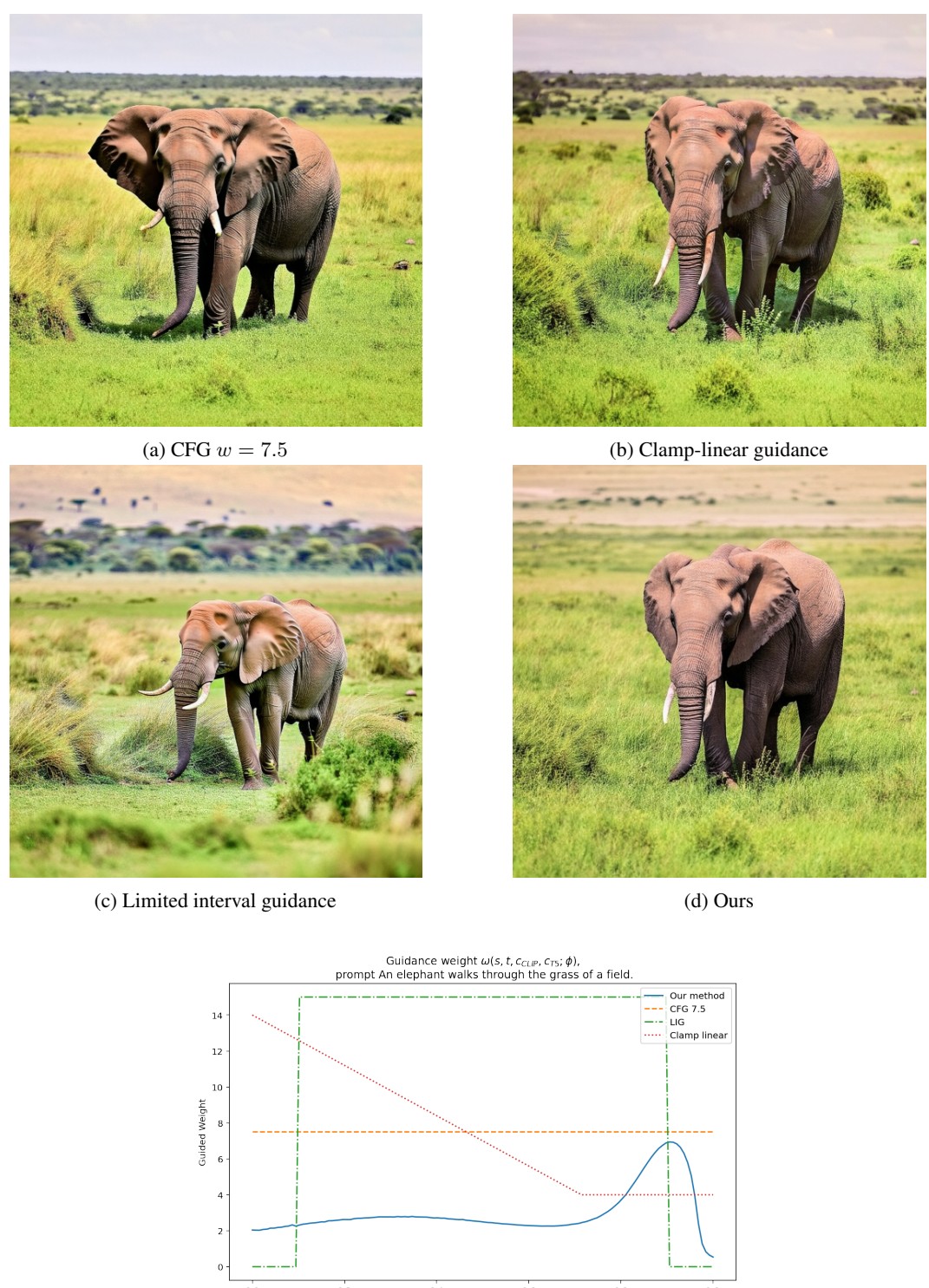

(a) CFG $w = 7.5$

(b) Clamp-linear guidance

(c) Limited interval guidance

(d) Ours

(e) Plot of Guidance weights

(f) **Text prompt**: An elephant walks through the grass of a field.

Figure 11: **T2I Results on MS-COCO.**. We provide results of images generated from the given text prompt with **(first row - left)** CFG $\omega = 7.5$, **(first row - right)** clamp linear, **(second row -left)** limited interval, and **(second row - right)** our method with self-consistency loss and CLIP score reward.

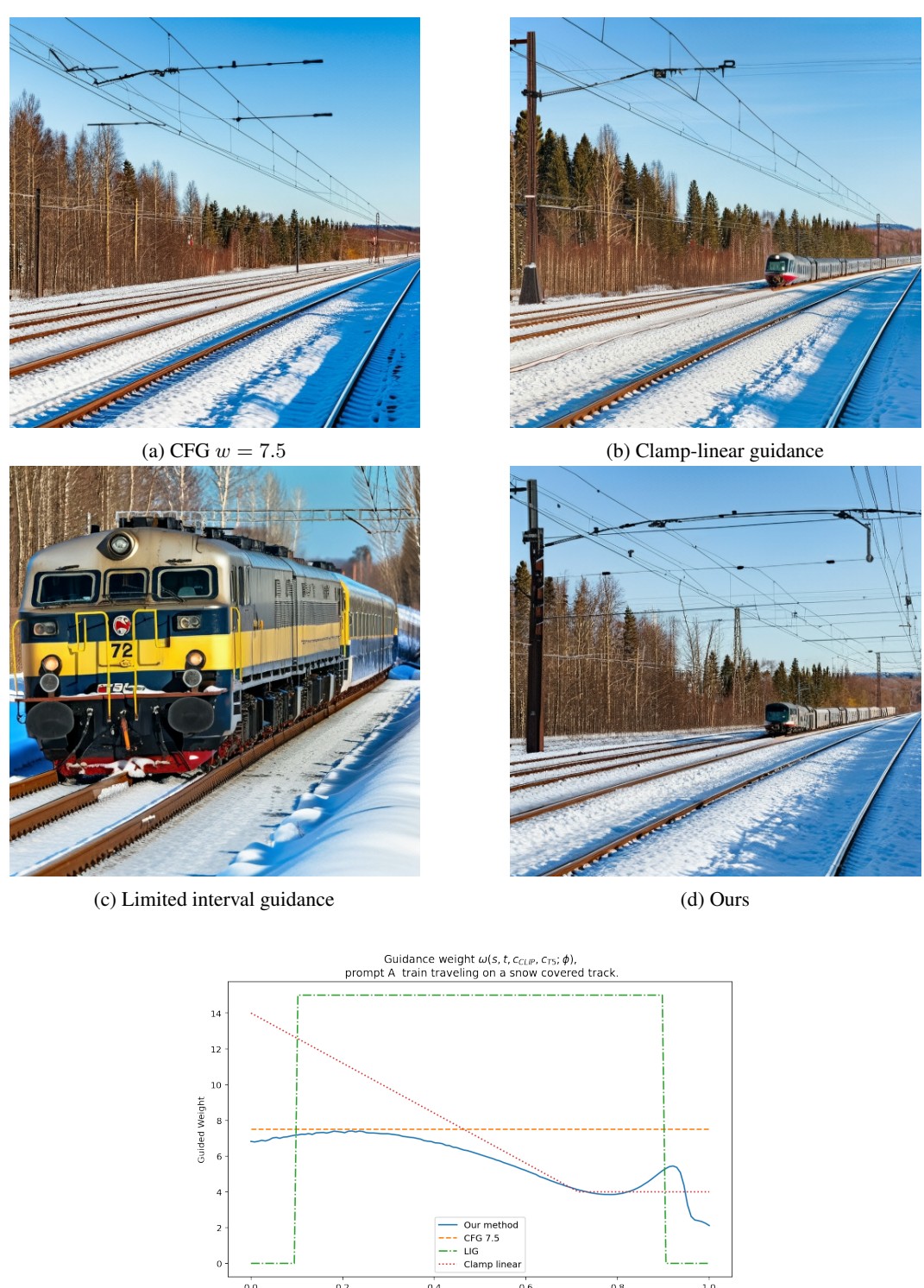

(a) CFG $w = 7.5$

(b) Clamp-linear guidance

(c) Limited interval guidance

(d) Ours

(e) Plot of Guidance weights

(f) **Text prompt**: A train traveling on a snow covered track.

Figure 12: **T2I Results on MS-COCO.**. We provide results of images generated from the given text prompt with **(first row - left)** CFG $\omega = 7.5$, **(first row - right)** clamp linear, **(second row - left)** limited interval, and **(second row - right)** our method with self-consistency loss and CLIP score reward.

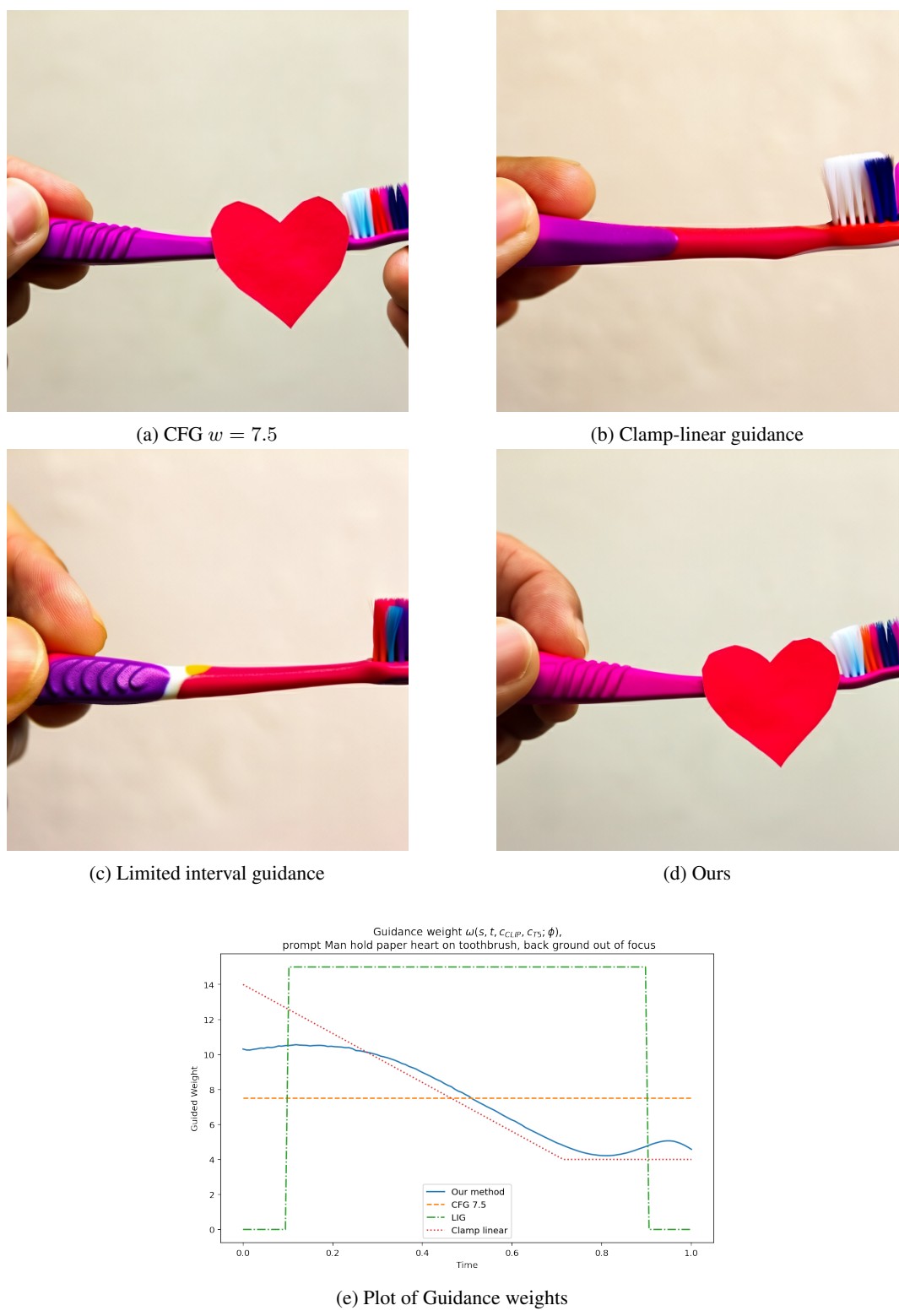

(a) CFG $w = 7.5$

(b) Clamp-linear guidance

(c) Limited interval guidance

(d) Ours

(e) Plot of Guidance weights

(f) **Text prompt**: Man hold paper heart on toothbrush, back ground out of focus.

Figure 13: **T2I Results on MS-COCO.**. We provide results of images generated from the given text prompt with **(first row - left)** CFG $\omega = 7.5$, **(first row - right)** clamp linear, **(second row -left)** limited interval, and **(second row - right)** our method with self-consistency loss and CLIP score reward.

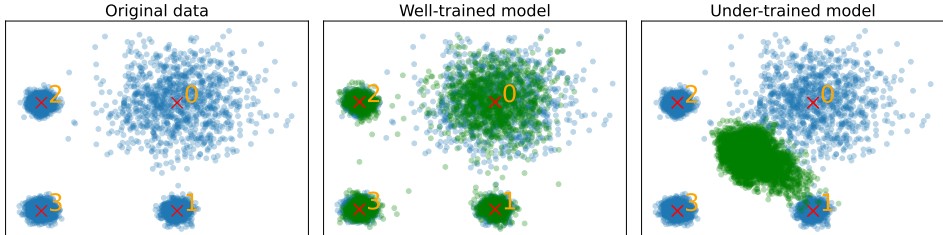

Figure 14: **Mixture of Gaussians data**. Red cross corresponds to the locations of the mean, while the numbers denote respective numbering of mixture components. The blue color corresponds to the data, while green color corresponds to the samples from diffusion model.

### I.3 EXPERIMENTS WITH STABLE DIFFUSION-V1.5

We run experiment using Stable Diffusion-v1.5 (SD-v1.5) (Rombach et al., 2022b) using a checkpoint from Hugging Face with Flax (Heek et al., 2024) backend. We use the same experimental setting (same guidance network architecture, same dataset, same batch size, same hyperparameters) as in T2I experiments reported in the main section of the paper, see Appendix F for more details. However, we do not use $T5$ conditioning (since it is not used in SD-v1.5) – we only use CLIP. Moreover, we use empty prompt instead of negative prompt, as we saw that constant guidance baseline led to better CLIP score with empty prompt (0.313) compared to negative prompt (0.304).

We train guidance network $\omega^{\phi}_{c_{\text{CLIP}}, c_{\text{CLIPseq}, (s,t)}}$ via Algorithm 1, where $c_{\text{CLIP}}$ is a CLIP (Radford et al., 2021) embedding and we additionally pass $c_{\text{CLIPseq}}$ which corresponds to non-pooled hidden state of CLIP, it has sequence dimension whereas $c_{\text{CLIP}}$ does not. Simiarly to the main paper, we also use a reward function $R(x_0, c)$ given by CLIP score (Hessel et al., 2021) computed between image $x_0$ and prompt $c$. We train guidance network via Algorithm 4 with $\gamma_R = 10^6$ (higher than for T2I runs in the main section of the paper). We report FID (Heusel et al., 2017) and CLIP Score using 10K samples. We train both variants for $40k$ iterations, and we report the performance at the end of the training. As baselines, we use constant guidance with $\omega = 0$ and $\omega = 7.5$.

The results are given in Table 11. We see that our method consistently leads to lower FID. We found that unlike for the results on MMDiT in the main section of the paper, we achieve a lower CLIP score compared to the constant guidance baseline. Drawing on our findings from ImageNet, we hypothesize that the performance of our method could be further improved by conducting a hyperparameter sweep for the distribution $p(s,t)$, rather than reusing the parameters from the MMDiT experiments.

| Method name | CLIP Score | FID |
|---|---|---|
| Constant guidance $\omega = 0$ | 0.271 | 33.203 |
| Constant guidance $\omega = 7.5$ | **0.313** | 23.479 |
| Self-consistency (20) $\omega^{\phi}_{c_{\text{CLIP}}, c_{\text{CLIPseq}}, (s,t)}$ | 0.294 | 21.670 |
| Self-consistency (20) $\omega^{\phi}_{c_{\text{CLIP}}, c_{\text{CLIPseq}}, (s,t)}$ + CLIP reward | 0.311 | **19.36** |

Table 11: Stable Diffusion-v1.5 results.

### I.4 2D MIXTURE OF GAUSSIANS (MOG)

We consider a 4 components MoG with the means $(10, 10)$, $(-10, 10)$, $(10, -10)$ and $(-10, -10)$, as well as the diagonal covariances $\sigma^2\text{Id}$ where the corresponding variances $\sigma^2$ are equal to $5, 1, 1, 1$. The data is visualized in Figure 14, left.

We study two regimes, *well-trained* and *under-trained*. In a *well-trained* regime, the diffusion model is trained for long enough in order to produce the samples close to the training distribution $p_{0|c}$, while in *under-trained*, it is trained for a small amount of iterations and produces samples far away from the training distribution. See Figure 14, center and right for corresponding samples. Please refer to Appendix F for all experimental details.

We train $\omega_{c,(s,t)}^{\phi}$ with Algorithm 1. The results are provided in Figure 15. In Figure 15, A, we visualize the MMD between samples from data distributions and samples produced by diffusion model with different constant guidance weights $\omega$. In the *well-trained* regime, as we increase the guidance weight $\omega$, it degrades the MMD, because it pushes the samples away from the distribution. What is intriguing is that in *under-trained* regime, MMD decreases as the guidance weight $\omega$ increases, achieving the lowest value with the learned guidance weight $\omega_{c,(s,t)}^{\phi}$. This finding highlights one of our main message – CFG can be used to correct the mismatch of the sampled distribution and the data distribution. In Figure 15, B, we visualize the learned $\omega_{c,(t-dt,t)}^{\phi}$, where $dt$ is a time discretization. In the *well-trained* regime, the learned guidance weights tend to be close to $0$, while in *under-trained* regime, the learned guidance weights reach quite high values. In Figure 16, we visualize the corresponding samples. This finding highlights that in an under trained regime, high guidance weights are required.

**A** **B**

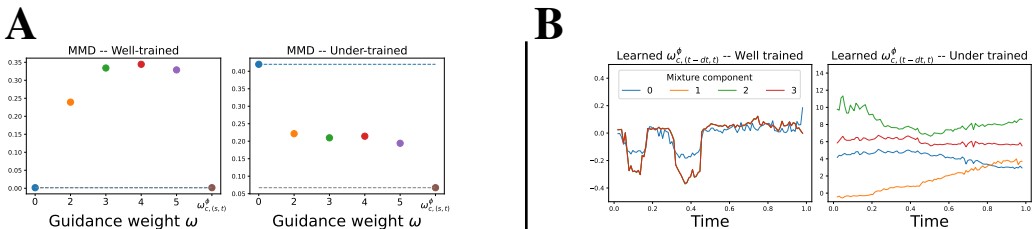

Figure 15: **A**: MMD between samples ($4096$) from data distribution and samples produced by a diffusion model with different guidance weight. Blue dashed line denotes the performance of unguided model, while brown dashed line denotes the performance of a model guided with learned $\omega_{c,(t-dt,t)}^{\phi}$.

**B**: learned guidance weight $\omega_{c,(t-dt,t)}^{\phi}$ from Algorithm 1 for different mixture components.

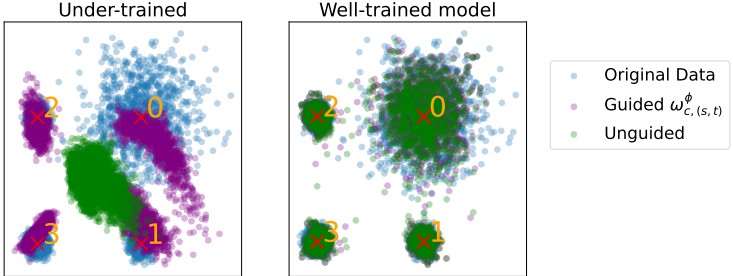

Figure 16: **Samples with and without guidance**. In blue, we show the original data, in green we show samples from unguided model while in purple we show samples from the guided model with learned guidance $\omega_{c,(s,t)}^{\phi}$ with Algorithm 1. First column corresponds to *under-trained* regime while second column corresponds to *well-trained* one.

## J    ANALYSIS OF GUIDANCE WEIGHTS

For the analysis of guidance weights, we evaluate guidance network $\omega_{c,(t-dt,t)}^{\phi}$ on all 10k prompts from COCO or 1000 classes from ImageNet, using $T = 128$ denoising steps for COCO, and $T = 100$ steps for ImageNet. We then compute average guidance weight per conditioning, i.e.

$$\bar{\omega}_c = \frac{1}{T}\sum_{t=1}^{T}\omega_{c,(t-dt,t)}^{\phi} \tag{30}$$

and also a sum of guidance weights per conditioning, i.e.,

$$\hat{\omega}_c = \sum_{t}\omega_{c,(t-dt,t)}^{\phi} \tag{31}$$

We then select top K (for K=10), median K (K after the median) and bottom K prompts / classes based on the values $\hat{\omega}_c$.

### J.0.1 ANALYSIS OF GUIDANCE WEIGHTS ON MS COCO

We visualize $\omega^{\phi}_{c,(t-dt,t)}$ for top, median and bottom K prompts in Figure 17. The results indicate that the guidance weights behave quite differently for these 3 categories. For bottom K, we see that guidance weights are high close to the noise and low throughout. For the median K guidance weights, we see that they are increasing as t approaches 0. For top K guidance weights, we see that they are generally high. This suggests that there is a very different behavior depending on the prompt. Moreover, in Figure 18, we show mean and standard deviation of $\bar{\omega}_c$ across prompts for a given prompt length. Overall, it seems that prompt length does not affect the magnitude of guidance weights that much.

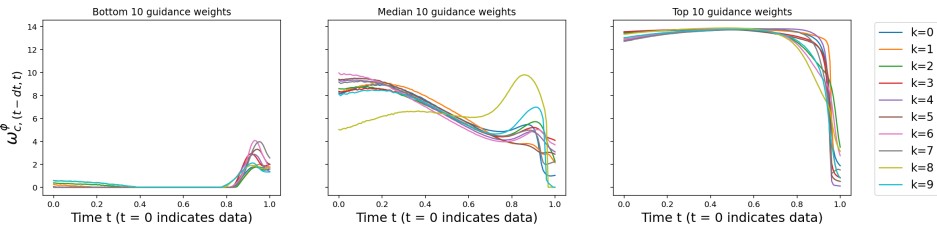

Figure 17: $\omega^{\phi}_{c,(t-dt,t)}$ for top, median and bottom K prompts based on the values of sum of per-prompt guidance weights $\hat{\omega}_c$ 31

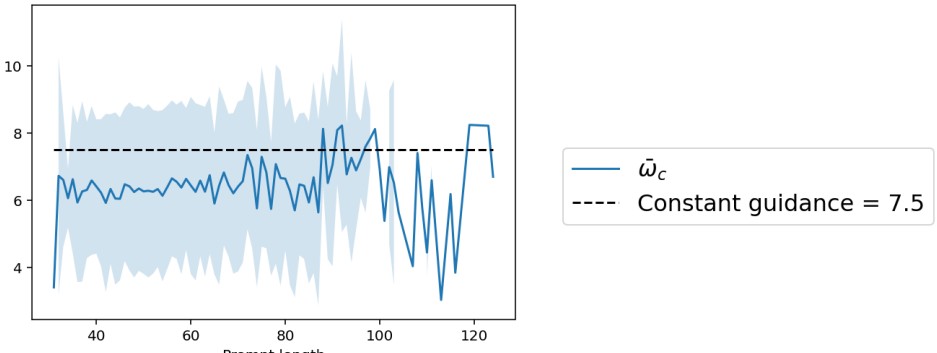

Figure 18: Average per-prompt guidance weights $\bar{\omega}_c$ 30 for different prompt lengths, shaded region indicate standard deviation.

Below we show the prompts for top, median and bottom K categories. See Table 12, Table 13, Table 14. It is hard to see if there is a specific category of prompts which leads to high or low guidance, so we hypothesize that the discrepancy is due to how the original diffusion model was trained.

### J.0.2 ANALYSIS OF GUIDANCE WEIGHTS ON IMAGENET

We visualize $\omega^{\phi}_{c,(t-dt,t)}$ for top, median and bottom K classes on ImageNet, see Figure 19.

Unlike in T2I, we see that guidance weights seem to have two distinct behaviors – for bottom K, they are generally small, while for median and top K they are generally large. We also see that they look like a form of Limited Interval Guidance (LIG) and clamp-linear guidance. The discrepancy to T2I could be attributed to a few things. First, in T2I, we add clip score reward to the loss which encourages prompt alignment. Second, in T2I, the diffusion model was trained on a vast amount of text-image pairs, such that each prompt receives an unequal number of images. In ImageNet, each class is seen the same amount of time.

Below, we show the class labels and their corresponding $\bar{\omega}_c$ for top, median and median K classes based on $\hat{\omega}_c$. See Table 15, Table 16 and Table 17.

Table 12: Top K Prompts with Highest average guidance weight ($\bar{\omega}_c$) 30

| k | Prompt | $\bar{\omega}_{\mathbf{c}}$ |
|---|---|---|
| 0 | an old rusty refrigerator is abandoned in a field. | 12.975 |
| 1 | An old truck with several years of paint schemes sits parked as if artwork. | 12.936 |
| 2 | A sky filled with lots of colorful flying kites. | 12.881 |
| 3 | A pile of junk sitting next to a curb on green grass. | 12.756 |
| 4 | Street art painted on the wall in an asian country. | 12.733 |
| 5 | OLD RUSTED BOXCARS STILL SITTING ON A SET OF TRACKS | 12.727 |
| 6 | A field full of people flying kites in a cloudy blue sky. | 12.630 |
| 7 | A crowded city filled with lots of people and traffic. | 12.607 |
| 8 | a large clock tower towering above a city under a cloudy blue sky. | 12.544 |
| 9 | Some roll down doors at a building painted with graffiti. | 12.494 |

Table 13: Median K Prompts by average guidance weight ($\bar{\omega}_c$) 30

| k | Prompt | $\bar{\omega}_{\mathbf{c}}$ |
|---|---|---|
| 0 | A group of men standing around a table of food in wooden room. | 6.477 |
| 1 | A room with a toilet, a door and shoes in it. | 6.475 |
| 2 | A boy in black shirt doing a trick on a skateboard. | 6.474 |
| 3 | a man stands in front of a flipped skate boarder | 6.474 |
| 4 | Oddly shaped homemade pizza about to be cut with pizza cutter | 6.474 |
| 5 | a silver toilet and some toilet paper and a green switch | 6.471 |
| 6 | Two cakes sitting on a class table near a candle. | 6.471 |
| 7 | A counter top in a kitchen topped with fruits and vegetables. | 6.470 |
| 8 | A park with many trees and benches at night. | 6.470 |
| 9 | A duck floating on a lake with gray and black feathers. | 6.470 |

Table 14: Bottom K Prompts with Lowest average guidance weight ($\bar{\omega}_c$) 30

| k | Prompt | $\bar{\omega}_{\mathbf{c}}$ |
|---|---|---|
| 0 | A group of kids that are playing soccer on a field. | 0.217 |
| 1 | A group of men playing a game of soccer on a field. | 0.243 |
| 2 | A group of people playing a game of soccer on top of a field. | 0.280 |
| 3 | A group of men who are playing basketball together. | 0.335 |
| 4 | Small boys are playing a softball game on grass. | 0.347 |
| 5 | A group of men on a field playing baseball. | 0.358 |
| 6 | Brown horse grazing alone in an open grassy field. | 0.392 |
| 7 | Men on a baseball diamond playing baseball, a man just swung | 0.393 |
| 8 | A group of soccer players scramble against each other to get the ball. | 0.413 |
| 9 | A group of children playing soccer on a field. | 0.415 |

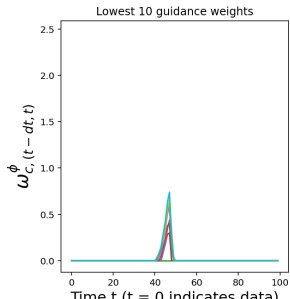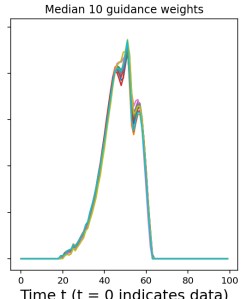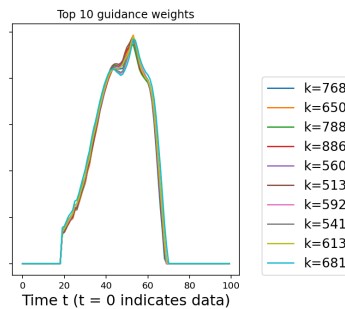

Figure 19: Guidance weights for bottom, median, top K on ImageNet.

Table 15: Top K Classes with Highest Mean Guidance Weight (mean $\omega$) 30

| k | Class id | Class name | mean $\omega$ |
|---|---|---|---|
| 0 | 768 | rugby_ball | 0.727 |
| 1 | 650 | microphone | 0.735 |
| 2 | 788 | shoe_shop | 0.735 |
| 3 | 886 | vending_machine | 0.738 |
| 4 | 560 | football_helmet | 0.743 |
| 5 | 513 | cornet | 0.745 |
| 6 | 592 | hard_disc | 0.748 |
| 7 | 541 | drum | 0.751 |
| 8 | 613 | joystick | 0.752 |
| 9 | 681 | notebook | 0.777 |

Table 16: Median K Classes by Conditional Guidance Weight ($\bar{\omega}_c$) 30

| k | Class id | Class name | $\bar{\omega}_{\mathbf{c}}$ |
|---|---|---|---|
| 0 | 345 | ox | 0.445 |
| 1 | 735 | poncho | 0.445 |
| 2 | 765 | rocking_chair | 0.444 |
| 3 | 607 | jack-o'-_lantern | 0.444 |
| 4 | 51 | triceratops | 0.447 |
| 5 | 660 | mobile_home | 0.447 |
| 6 | 565 | freight_car | 0.444 |
| 7 | 108 | sea_anemone | 0.447 |
| 8 | 200 | Tibetan_terrier | 0.443 |
| 9 | 255 | Leonberg | 0.443 |

Table 17: Bottom K Classes with Lowest Conditional Guidance Weight ($\bar{\omega}_c$) 30

| k | Class id | Class name | $\bar{\omega}_{\mathbf{c}}$ |
|---|---|---|---|
| 0 | 640 | manhole_cover | 0.000 |
| 1 | 138 | bustard | 0.000 |
| 2 | 83 | prairie_chicken | 0.000 |
| 3 | 353 | gazelle | 0.009 |
| 4 | 974 | geyser | 0.012 |
| 5 | 44 | alligator_lizard | 0.012 |
| 6 | 106 | wombat | 0.020 |
| 7 | 352 | impala | 0.021 |
| 8 | 276 | hyena | 0.024 |
| 9 | 209 | Chesapeake_Bay_retriever | 0.026 |

