# OpenReview forum: "Learn to Guide Your Diffusion Model"
_ICLR.cc/2026/Conference — ICLR 2026 Poster_

### Official Review · Reviewer_A5xC · 2025-10-22

**Soundness:** 2
**Presentation:** 3
**Contribution:** 3
**Rating:** 4
**Confidence:** 5

**Summary:**

The paper proposes to introduce the Maximum Mean Discrepancy (MMD) to quantitatively evaluate the distance between the CFG-guided distribution and the ground-truth one. By doing so, it is enabled to design both time-dependent and condition-dependent guidance weight, which is more flexible and controllable. To realize such a pipeline, the authors employ an auxiliary network for guidance weight prediction, and further discuss the influences under different objectives. Both qualitative and quantitative results confirm the superiority of the proposed method.

**Strengths:**

- The paper is well structured and easy to follow. All technical details are carefully discussed and clear.
- The motivation is intuitive and effective, alleviation the discrepancy between CFG-guided and the ground-truth distributions is capable of improving the performance of guided sampling.
- The further discussion about different objective is detailed and inspiring, encouraging future study for better training efficiency and synthesis performance.

**Weaknesses:**

- The whole pipeline of narrowing the distribution discrepancy by optimizing some scalars is not new. Beyond the main objective being directly motivated by prior works, similar idea of canceling mismatch (or "self-consistency") between forward diffusion and reverse denoising stages has also been proposed [1].

  [1] Towards More Accurate Diffusion Model Acceleration with A Timestep Tuner. Xia et al., CVPR 2024.

- The main concern about the proposed method is the optimality. There is no theoretical analysis about the optimality or convergence about the proposed objective. Then why the optimized guidance weight could guarantee the self-consistency? What further confirms my worries is in Appendix C.1. The most intuitive objective is the L2 norm in Eq. (23), which is consistent with the vanilla training loss of diffusion models, *i.e.*, the ELBO loss. However, as the authors themselves claim, such an objective leads to zero guidance weight. That is to say, **diffusion model converges well and both conditional and unconditional scores are accurate**, thus there is no need to employ CFG. Under this discussion, I am curious about the theoretical optimality of Eq. (20) or Eq. (21). Could the authors provide theoretical analyses under some trivial toy data to verify the correctness of the method? From my opinion, Eq. (21) is somewhat an upper bound of ELBO (*i.e.*, Eq. (23)). Given that ELBO itself is an upper bound of KL-divergence (by the native theory of diffusion models), the optimization of a looser bound may lead to meaningless results.

- What further weakens the soundness of the paper is the employment of an auxiliary network. To be honest, such a setting could lead to more flexible guidance weight given different conditions. However, what if the network fails to converge well and only predicts sub-optimal guidance weight? The accuracy issue is more severe under open-vocabulary setting, *i.e.*, text-to-image or text-to-video. Considering that the authors employ a light-weight network, such a concern is also crucial. Could the author provide some closed-form expressions or some empirical solutions avoiding the employment of an auxiliary network?

- The qualitative results of text-to-image is somewhat not convincing. The improvements in Figs. 4-7 are inconspicuous. This strengthens my concerns above.

**Questions:**

Beyond the Weaknesses part, I am curious about the general rule of the guidance weights. Current analyses in Fig. 3 is poor. Could the authors provide more quantitative analyses about the weights under different conditions and timesteps? For example, what if a text condition gradually becomes complex and a long caption? What about the mean or variance?

---

> ### Author Response · Authors · 2025-11-17
> **Rebuttal part 1**
>
> We thank Reviewer A5xC for their feedback. Please find our detailed answer below.
>
> > The whole pipeline of narrowing the distribution discrepancy by optimizing some scalars is not new...[1] Towards More Accurate Diffusion Model Acceleration with A Timestep Tuner. Xia et al., CVPR 2024.
>
> We thank the reviewer for bringing up this work. The method in [1] uses a different mechanism for cancelling mismatch which relies on essentially tuning the noise level of a denoiser, while in our work we focus on classifier free guidance. The primary focus of [1] is correcting discretization errors in the solver to enable fewer sampling steps, while our work focuses on optimizing guidance schedules of CFG. We will add this paper to the related work.
>
>
> > The main concern about the proposed method is the optimality.
>
> We would like to reiterate the motivation behind our method based on self-consistency. Our method is based on the principle of self-consistency which serves as a proxy for theoretically grounded but difficult to optimize marginal consistency.
>
> We however emphasize that enforcing self-consistency implies marginal consistency (since you can integrate over $(x_0, c)$). Marginal consistency condition was also considered in [1] where authors trained a distributional model to get a sample at time s from a sample at time t. Instead of directly optimizing the challenging marginal consistency objective (eq.15), they designed an alternative approach that guarantees this condition at convergence. They reported strong empirical results which provided motivation for marginal consistency conditions. This means that optimizing for self-consistency (i.e. eq.18, eq.20) is useful since it will allow us to achieve marginal consistency.
>
> While deriving a theoretical bound that guarantees our optimized guidance weights via Eq 20 strictly improves conditional sampling remains an open problem (common to many deep learning optimizations), we do observe empirically in 2D toy datasets, image and text-to-image settings, that using this objective function leads to guidance weights which outperform unguided and guided baselines.
>
> [1] Inductive Moment Matching, Linqi Zhou, Stefano Ermon, Jiaming Song, 2025
>
> > What further confirms my worries is in Appendix C.1. The most intuitive objective is the L2 norm in Eq. (23), which is consistent with the vanilla training loss of diffusion models, i.e., the ELBO loss. However, as the authors themselves claim, such an objective leads to zero guidance weight. That is to say, diffusion model converges well and both conditional and unconditional scores are accurate, thus there is no need to employ CFG.
>
> We would like to highlight that in eq.(23), we do denoising on the data coming from the forward diffusion process. As we argue in Appendix C.1, a reasonably pretrained model already would approximate the conditional expectation of clean data given a noisy sample $x_t$ from the forward process, which leads to zero guidance, since this conditional expectation is the minimizer of such loss. In practice, however, the reverse process imperfectly follows the reverse of the forward process, and our claim is that guidance helps to correct this. We also would like to cite a sentence from the introduction of our paper: “this suggests that CFG acts as a correction to this approximation which leads to a better modeling of the target conditional distribution p(x0|c)”.

---

> > ### Author Response · Authors · 2025-11-17
> > **Rebuttal part 2**
> >
> > > why the optimized guidance weight could guarantee the self-consistency? ... Under this discussion, I am curious about the theoretical optimality of Eq. (20) or Eq. (21). Could the authors provide theoretical analyses under some trivial toy data to verify the correctness of the method? From my opinion, Eq. (21) is somewhat an upper bound of ELBO (i.e., Eq. (23)). Given that ELBO itself is an upper bound of KL-divergence (by the native theory of diffusion models), the optimization of a looser bound may lead to meaningless results.
> >
> > The enforcing  of self-consistency comes from the formulation of our optimization objective eq. 18 and eq. 20. Our method optimizes this objective to learn guidance weights that satisfy self-consistency constraints. By definition, our objective in eq. 18, which is divergence between the forward distribution and guided reverse conditional distribution, both conditioned on $(x_0, c)$, is minimized when the two distributions match.
> >
> > **Evaluation on toy datasets**: We would like to point out that we also evaluate our approach on toy 2d MoG dataset, see appendix G. We observe that it allows us to learn guidance weights which improve MMD compared to unguided baselines and baselines with manually chosen constant guidance.
> >
> > **Theoretical analysis**: While deriving a theoretical bound that guarantees our optimized guidance weights via Eq 20 strictly improve conditional sampling remains an open problem (common to many deep learning optimizations), our empirical results suggest that optimizing this objective leads to improved performance on 2D toy datasets, image generation and text-to-image settings. Please see also our common response and appendix H for the additional experiments and analysis which we have included to the paper.
> >
> > > What further weakens the soundness of the paper is the employment of an auxiliary network...
> >
> > The lightweight auxiliary network allows us to take advantage of the additional conditional information which is available in the datasets. As our empirical results suggest, adding more conditional information leads to overall better performance. This highlights that employing an auxiliary network is actually a strength of our approach due to its flexibility. Our empirical results on 2d data, image and text-to-image settings demonstrate that using an auxiliary network with conditioning information leads to better results than baselines.
> >
> > > Could the author provide some closed-form expressions or some empirical solutions avoiding the employment of an auxiliary network?
> >
> > While we do not currently provide the closed form expressions for the guidance weights learned from our objective function, we provide extensive empirical evidence that our method performs better than unguided and guided baselines.
> >
> > > The qualitative results of text-to-image is somewhat not convincing. The improvements in Figs. 4-7 are inconspicuous...
> >
> > Please see our common response where we provided additional analysis of learned guidance weights on T2I and ImageNet. We hope our new results will convince you of the usefulness of our method.
> >
> > > Beyond the Weaknesses part, I am curious about the general rule of the guidance weights. Current analyses in Fig. 3 is poor. Could the authors provide more quantitative analyses about the weights under different conditions and timesteps? For example, what if a text condition gradually becomes complex and a long caption? What about the mean or variance?
> >
> > Please see our common response where we provided additional analysis of learned guidance weights on T2I and ImageNet. We have also updated the appendix to include this additional analysis. We hope our new results will convince you of the usefulness of our method.
> >
> > We hope the above addresses the reviewer’s concerns - if we have done so, we would be grateful if the reviewer might consider increasing their score.  We are happy to address any further questions.

---

> > > ### Comment · Reviewer_A5xC · 2025-11-19
> > >
> > > Thanks for the extensive clarifications and additional experiments. I think most of concerns are well addressed except for the Eq. (23) part. It is true that directly employing Eq. (23) with clean data and forward diffusion model will theoretically lead to zero guidance weight. As proposed in [1], they could use both forward diffusion and reverse denoising process to optimize and annihilate the distribution mismatch. So what I am currently curious about is that, what if the authors modify Eq. (23) with the denoised intermediate samples instead of the diffused ones?
> > >
> > > I am highly willing to raise my score if all my concerns are perfectly addressed.
> > >
> > > [1] Towards More Accurate Diffusion Model Acceleration with A Timestep Tuner. Xia et al., CVPR 2024.

---

> > > > ### Author Response · Authors · 2025-11-21
> > > > **Response part 1**
> > > >
> > > > > Thanks for the extensive clarifications and additional experiments. I think most of concerns are well addressed except for the Eq. (23) part. It is true that directly employing Eq. (23) with clean data and forward diffusion model will theoretically lead to zero guidance weight. As proposed in [1], they could use both forward diffusion and reverse denoising process to optimize and annihilate the distribution mismatch. So what I am currently curious about is that, what if the authors modify Eq. (23) with the denoised intermediate samples instead of the diffused ones? I am highly willing to raise my score if all my concerns are perfectly addressed. [1] Towards More Accurate Diffusion Model Acceleration with A Timestep Tuner. Xia et al., CVPR 2024
> > > >
> > > > We thank the reviewer again for bringing up the work [1]. We implemented a few alternatives to the method described in eq.23, based on your suggestions and based on [1]. We use an identical network architecture as our method, and we train this model on ImageNet-64x64. We discuss these methods below and report the corresponding results. Overall, only one variant (see below) led to performance close to limited-interval guidance (LIG), but still under-performed compared to our approach. This variant was also much more sensitive to hyperparameters compared to our self-consistency approach.
> > > >
> > > > For all the methods, we describe methodology for one sample, but in practice we always use mini batches.
> > > >
> > > > **Method 1 – simple modification of eq.23**
> > > >
> > > > We follow your suggestion “So what I am currently curious about is that, what if the authors modify Eq. (23) with the denoised intermediate samples instead of the diffused ones?”. We describe the method below.
> > > >
> > > > * Sample $(x_0,c) \sim p(x_0,c)$
> > > >
> > > > * Sample $t \sim U[0,1]$
> > > >
> > > > * Noise $x_0$ to time $t$ with $x_t \sim p(x_t | x_0)$
> > > >
> > > > * Denoise to $\hat{x_0} = denoiser(x_t, t)$
> > > >
> > > > * Renoise it to time $t$ again to get $\hat{x_t} \sim p(\cdot | \hat{x_0})$
> > > >
> > > > * Optimize an objective similar to eq.23, i.e. $||guideddenoiser(\hat{x_t}, t, \omega) - x_0||^2$
> > > >
> > > > We have conducted an extensive sweep over hyperparameters of $p(s,t)$ similar to our approach. We also swept over whether to use $\omega^\phi_{c,t}$ or $\omega^\phi_{c,(s,t)}$.
> > > >
> > > > **The results**: We tested method 1 and found that it learns guidance weights equal to zero.
> > > >
> > > > **Method 2 – more involved modification of eq.23**
> > > >
> > > > We again follow your suggestion as with method 1 and we do a partial denoising as you suggested. We describe the method below.
> > > >
> > > > * Sample $(x_0,c) \sim p(x_0,c)$
> > > >
> > > > * Sample $s,t \sim p(s,t)$ which is a similar distribution we used in our work
> > > >
> > > > * Noise $x_0$ to time $t$ with $x_t \sim p(x_t | x_0)$
> > > >
> > > > * Partially denoise without guidance with DDIM (with a churn parameter) to time $s$ with $\hat{x_s} \sim p(x_s | x_t, \hat{x_0} = denoiser(x_t, t))$
> > > >
> > > > * Optimize the loss similar to eq 23., i.e. $||guideddenoiser(\hat{x_{s}}, s, \omega) - x_0||^2$
> > > >
> > > > We have conducted an extensive sweep over hyperparameters of $p(s,t)$ similar to our approach. We also swept over whether to use $\omega^\phi_{c,t}$ or $\omega^\phi_{c,(s,t)}$. For DDIM, we also tried using a churn parameter equal to 0.0 or 1.0.
> > > >
> > > > **Results**: We found that it was important to include both timesteps, i.e., $\omega^\phi_{c,(s,t)}$. This variant with the best hyperparameters led to FID ~ 2.61. When we used $\omega^\phi_{c,t}$ , this led to guidance weights equal to zero. We visualize guidance in https://drive.google.com/file/d/13tpfge5HWq2pOO5-yNXbXB29glbb51FQ/view?usp=sharing. We see that the shape is such that a lot of guidance is applied closer to the noise, with a peak in the middle and zero guidance near the data. However, the best FID ~ 2.61 is much higher than for our method and for the baselines which we considered.

---

> ### Author Response · Authors · 2025-11-21
> **Response part 2**
>
> **Method 3 – Application of [1]**
>
> We have implemented a variant of the method described in [1]. Since Algorithm 1 from [1] is impractical, as it requires to do many denoising steps up to a given time $s$, we have implemented a variant proposed in Section 3.5 from [1], which is "Parallel Training Strategy of TimeTuner". We state here what we have implemented:
>
> * Sample $(x_0,c) \sim p(x_0,c)$
>
> * Sample $s,t \sim p(s,t)$ which is a similar distribution we used in our work
>
> * Noise $x_0$ to time $t$ with $x_t \sim p(x_t | x_0)$
>
> * Denoise with guidance and DDIM (with churn parameter) to time $s$, i.e., $\hat{x_s}(\omega) \sim p^{\omega}(x_s | x_t)$
>
> * Compute the loss $||denoiser(\hat{x_s}(\omega), s) - denoiser(x_t, t)||^2$. This loss is similar to eq.11 in [1].
>
> We have conducted an extensive sweep over hyperparameters of $p(s,t)$ similar to our approach. We also swept over whether to use $\omega^\phi_{c,t}$ or $\omega^\phi_{c,(s,t)}$. For DDIM, we also tried using a churn parameter equal to 0.0 or 1.0.
>
> **Results**: We found that $\omega^\phi_{c,t}$ worked better than $\omega^\phi_{c,(s,t)}$ for this method. The best performance across all hyperparameters for $\omega^\phi_{c,t}$ was FID = 2.13, while for $\omega^\phi_{c,(s,t)}$ it was FID = 2.51. The best parameters for $\omega^\phi_{c,t}$ were $S_{\min}=0.2$, $\delta=0.3$ and DDIM churn parameter equal to 1.0. This produced guidance weights similar to limited interval guidance (LIG) baseline on ImageNet (see https://drive.google.com/file/d/1MhUMenyEfuPZzyviD5MyK4lXfpxdchaP/view?usp=sharing). We also see that the guidance weights do not seem to vary much for different conditionings.
>
> However, we noticed that the method is very sensitive to hyperparameters compared to our self-consistency approach (see Figure 2 in our paper).  Across the sweep of $\min_{S} \in \{0.01, 0.1, 0.2, 0.3 \}$ and $\delta \in \{0.01, 0.1, 0.2, 0.3 \}$, most of the experiments led to guidance equal to zero, and only 3 combinations worked. We provide these combinations in the tables below with corresponding FIDs.
>
> **Hyperparameters which led to non-zero guidance weights with $\omega^\phi_{c,t}$**
>
> |Hyperparameters | FID |
> | -- | -- |
> | $S_{\min}=0.2,\delta=0.3$|  2.13 |
> | $S_{\min}=0.3,\delta=0.2$|  2.3 |
> | $S_{\min}=0.3,\delta=0.3$|  2.33 |
>
> **Hyperparameters which led to non-zero guidance weights with $\omega^\phi_{c,(s,t)}$**
>
> |Hyperparameters | FID |
> | -- | -- |
> | $S_{\min}=0.2,\delta=0.3$|  3.62 |
> | $S_{\min}=0.3,\delta=0.2$|  2.96 |
> | $S_{\min}=0.3,\delta=0.3$|  2.52 |
>
>
> Below, we show a summary of our experiments comparing the best-found hyperparameters for method 1, method 2 and method 3 to self-consistency and other baselines.
>
> | Method | FID | IS|
> | -- | -- | -- |
> | Self-consistency | 1.99 | 73.62|
> | Unguided baseline | 4.46 | 43.52|
> | Constant guidance| 2.4 | 66.72|
> | Limited interval guidance| 2.11 | 71.6|
> | Method 1  | 4.46 | 43.52 |
> | Method 2  | 2.61 | 65.47 |
> | Method 3  | 2.13 | 72 |
>
> Overall, method 3 (which is closely related to [1]) led to somewhat reasonable results, however it seems to be very sensitive to hyperparameters and it has overall worse performance than our self-consistency approach as well as being less robust. We will add these comparisons to the appendix of the paper.
>
> We hope that we have fully addressed your concerns, through the clarification regarding Equation 23 as requested. If we have done so, we would respectfully ask that you reconsider your score. We would be happy to address any further questions.

---

> > ### Comment · Reviewer_A5xC · 2025-11-22
> >
> > Thanks for the comprehensive additional experiments that provide more detailed clarifications of the proposed method. I highly appreciate the authors' great efforts. There is one point I hope to make it clear: The paper I mentioned before uses $x_T$ and denoises to each $x_t$, rather than one-step denoising and one-step diffusing. For example, for $0=t_0<t_1<\cdots<t_M=T$, the paper produces $x_{t_i}$ for $i<M$ by'
> >   1. Sample $x_{t_M}$
> >   2. Denoise $x_{t_M}$ to $x_{t_{M-1}}$, then step-by-step to $x_{t_{i+1}}$ and finally $x_{t_i}$
> >
> > If the authors could further provide this comparison, I will raise my score accordingly.

---

> ### Author Response · Authors · 2025-11-26
> **Response part 1**
>
> We thank the reviewer for the clarification, and we appreciate the reviewer's quick responses and constructive engagement.
>
> Based on your specific description—generating $x_t$ via a step-by-step denoising chain starting from $x_T$ rather than one-step diffusion—we have implemented the requested method (Method 4). Moreover, we have implemented a multi-step partial denoising for a guided score matching variant (method 2 above), which we call Method 5. We also provide a detailed analysis of the practical and theoretical challenges of applying Algorithm 1 from [1] to guidance learning.
>
> ## Method 4: Full Trajectory Denoising
>
> We implemented the procedure described in your current comment and based on your previous comments, as follows:
>
> * Sample $x_0 \sim p(x_0)$
>
> * Sample $x_T \sim \mathcal{N}(0, I)$
>
> * Denoise $x_T$ step-by-step to $x_{t_i}$ using DDIM with a churn parameter for a fixed number of steps $T_{steps}$.
>
> * Compute the loss $L_i = \| x_0 - \text{denoiser}(x_{t_i}, t_i, \omega) \|^2$ for each time step
>
> * Compute total loss $L = \sum_{i=1}^{T_{steps}} L_{i}$
>
> We swept over $T_{steps} \in \{5, 10, 20, 40, 50, 100 \}$ and DDIM churn parameters $\in \{ 0.0, 0.25, 0.5, 0.75, 1.0 \}$. We used the same guidance network architecture as in self-consistency method.
>
> **Outcome:** The learned guidance weights converged to **0** in all settings.
>
> **Analysis**: We attribute this to a fundamental objective mismatch. In Eq. 23 (standard diffusion loss with guidance), the loss minimizes the distance between the prediction derived from $x_t$ and the *corresponding* ground truth $x_0$. However, in this method, $x_{t_i}$ is derived from a random noise sample $x_T$ that is independent of the $x_0$ used in the loss function. Since $x_{t_i}$ and $x_0$ are uncorrelated, the optimal prediction for $x_0$ given $x_{t_i}$ is simply the mean of the data distribution, leading to a collapse of the guidance mechanism.
>
> ## Method 5: Multi-step Partial Denoising
>
> To address the correlation issue in Method 4, we implemented a "multi-step" version of the partial denoising approach (an extension of Method 2 from our previous response). This maintains the link between the trajectory and the target $x_0$.
>
> * Sample $(x_0, c)$
>
> * Noise $x_0$ to $x_t$ via forward diffusion
>
> * Partially denoise $x_t$ to $x_s$ (where $s < t$) using $K$ intermediate DDIM steps (instead of 1 step)
>
> * Optimize the loss $\| guideddenoiser(x_s, s, \omega) - x_0 \|^2$
>
> We swept over the same hyperparameters as Method 2, with the addition of $K \in \{ 1, 2, 3, 5, 10 \}$ steps. We also used the same guidance network architecture as in Method 2.
>
> **Outcome**: Increasing $K$ did not improve FID compared to the single-step partial denoising (Method 2) and remained significantly worse (FID \sim 2.61) than our proposed Self-Consistency method (FID \sim 1.99).
>
> **References**:
>
> [1] Towards More Accurate Diffusion Model Acceleration with A Timestep Tuner. Xia et al., CVPR 2024.

---

> > ### Author Response · Authors · 2025-11-26
> > **Response part 2**
> >
> > ## Analysis of Algorithm 1 from [1] for Guidance Learning
> >
> > We believe it is important to clarify why Algorithm 1 from [1], while effective for its original purpose of *timestep tuning for acceleration*, is fundamentally ill-suited and impractical for the task of *learning guidance weights*.
> >
> > 1. **Computational Intractability for Learned Guidance**
> >
> > Algorithm 1 requires backpropagating through a chain of guided denoising steps. When $\omega_{c,(s,t)}$ is a learnable network (as in our method), updating weights at step $t_i$ requires differentiating through the entire sampling chain from $T$ to $t_i$. This is computationally prohibitive and prone to unstable gradients (vanishing/exploding) compared to our efficient per-step Self-Consistency objective.
> >
> > 2. **Rigidity and NFE Incompatibility**
> >
> > Algorithm 1 optimizes a specific schedule for a fixed number of discretization steps (e.g., a specific sequence $t_0, ..., t_M$). If one wishes to change the NFE at inference time—for example, increasing steps for higher quality—the learned schedule is no longer valid and must be retrained.
> >
> > In contrast, our method learns a continuous function $\omega_{c, (s,t)}$. This is NFE-agnostic, allowing the same trained guidance network to be used with any sampler and any number of steps seamlessly.
> >
> > 3. **Theoretical Mismatch**
> >
> > The authors of [1] explicitly note that their method is designed to correct discretization errors in low-NFE regimes: **"the performance improvement is more significant with a small NFE... [because] larger NFE relieves the truncation error."**
> >
> > **Our goal is different:** we aim to correct the distributional mismatch caused by CFG to improve alignment and quality, regardless of NFE.
> >
> > We believe that a worthwhile research direction would be to explore ways of training guidance efficiently in the context of [1], and we will add a corresponding discussion to our paper.
> >
> > ## Conclusion
> >
> > We have exhaustively explored the trajectory-based variants suggested (Methods 1 through 5) and the parallelized adaptation of [1] (Method 3). The empirical evidence is consistent:
> >
> > * **Exact implementation of the request (Method 4)** fails due to the uncoupling of $x_t$ and $x_0$
> >
> > * **Partial denoising variants (Methods 2 & 5) underperform significantly**
> >
> > * **The practical adaptation of [1] (Method 3)** is unstable and underperforms
> >
> > * **Our Self-Consistency method** remains the most robust, efficient, and high-performing approach, offering NFE-agnostic flexibility that trajectory-based optimization lacks.
> >
> > We hope this detailed investigation addresses your concerns regarding the comparison with [1] and demonstrates the robustness of our proposed approach.
> >
> > **References**:
> >
> > [1] Towards More Accurate Diffusion Model Acceleration with A Timestep Tuner. Xia et al., CVPR 2024.

---

> > > ### Comment · Reviewer_A5xC · 2025-11-26
> > >
> > > Thanks for the additional results, together withe the detailed and clear analyses and comparisons. All my concerns are well addressed, and I raise my score accordingly.

---

### Official Review · Reviewer_h7dP · 2025-10-30

**Soundness:** 3
**Presentation:** 3
**Contribution:** 3
**Rating:** 4
**Confidence:** 3

**Summary:**

The authors propose a method for learning the guidance schedule for diffusion model generation. In particular, they train a shallow network to minimize the distributional mismatch between a frozen denoiser and the true distribution. The authors provide solid theoretical motivation for their choice of objective function, and also experimental evidence for class and text-conditional models.

**Strengths:**

- The authors replace a hyper-parameter (guidance strength) with a learned approach
- The objective function is well theoretically motivated

**Weaknesses:**

- The authors train their own diffusion models for use in evaluations. It would greatly strengthen the paper if they also showed FID and reward improvements for existing pre-trained diffusion models from prior work. For example:
     - EDM for cifar-10 (Karras et al, https://github.com/NVlabs/edm)
     - EDM-2 for ImageNet (Karras et al, https://github.com/NVlabs/edm2)
     - MicroDiffusion for text-to-image (Sehwag et al, https://github.com/SonyResearch/micro_diffusion)
- The authors compare their learned approach to simple baselines such as constant or limited-interval guidance. It would strengthen the paper if they also showed improvements over stronger un-trained baselines such as in Wang et al (https://arxiv.org/abs/2404.13040).

My score recommends reject, but I would be happy to raise my score to accept if these additional experiments are included in the paper and my questions below are addressed.

**Questions:**

- What do the authors mean by a high-variance objective?
- What is the training cost of training the learned guidance network as a percentage of diffusion model training cost?
- Why is the FID on COCO for the constant guidance model so high? FID 31 seems incredibly high for a 1B model. How was this model trained?

---

> ### Author Response · Authors · 2025-11-17
> **Rebuttal**
>
> We thank reviewer h7dP for their feedback. Please find our detailed answer below.
>
> > The authors train their own diffusion models for use in evaluations. It would greatly strengthen the paper if they also showed FID and reward improvements for existing pre-trained diffusion models from prior work. For example:
>
> We do not expect that the methodology will perform differently for pre-trained models in other codebases and we will aim to implement our method for these pretrained models for the final version of the paper. However, in our current paper, for the models which we trained ourselves, we do observe consistent improvement over the baselines, such as constant guidance, limited interval guidance and clamp-linear schedule. We do observe it on toy 2D MoG data, image (ImageNet and CelebA) and text-to-image (COCO) settings.
>
> > The authors compare their learned approach to simple baselines such as constant or limited-interval guidance. It would strengthen the paper if they also showed improvements over stronger un-trained baselines such as in Wang et al (https://arxiv.org/abs/2404.13040).
>
> We have compared our method to clamp-linear schedule from this paper, on T2I and on ImageNet, please see our common response to all the reviewers. We also provide detailed metrics and visualization in the Appendix H of the paper. We have also added a comparison of our method to LIG on T2I. The main take-aways are that our method still performs better than LIG and clamp-linear.
>
> > What do the authors mean by a high-variance objective?
>
> We meant that the variance of the estimates of the objective function from eq.15 and its gradient are high (since we will need to sample multiple (x0, c) and then additionally sample corresponding xs for lhs and rhs independently). We will correct this language imprecision.
>
> Please note that we also tried learning guidance weights from this objective but it led to very unstable learning of guidance weights and did not lead to meaningful results. See also our response to reviewer sh9D regarding the variance.
>
> > What is the training cost of training the learned guidance network as a percentage of diffusion model training cost?
>
> Please see our common reply about the training and inference cost.
>
> > Why is the FID on COCO for the constant guidance model so high? FID 31 seems incredibly high for a 1B model. How was this model trained?
>
> We believe that the discrepancy comes from the fact that we used a smaller Flux model (1B) and trained the model on a different dataset from the original Flux paper.
>
> We hope the above addresses the reviewer’s concerns - if we have done so, we would be grateful if the reviewer might consider increasing their score.  We are happy to address any further questions.

---

### Official Review · Reviewer_sh9D · 2025-11-01

**Soundness:** 3
**Presentation:** 3
**Contribution:** 3
**Rating:** 6
**Confidence:** 3

**Summary:**

This paper puts forward learning a classifier-free guidance schedule that varies with both time and the input condition (class or text), enabling the sampler to better match the desired conditional distribution.
Compare with handcrafting guidance strengths, the method optimizes a self-consistency objective: if one adds noise to a real image and then denoises it with guidance, the intermediate result should match what the forward noising process would produce at that timestep.
Following this desin mechnism, this reframes guidance tuning as a distribution-matching problem and naturally extends to reward-guided generation by combining an external reward (e.g., CLIP-based) with the same self-consistency regularizer.
Imortantly, exntensive experiments on standard image datasets and text-to-image benchmarks show consistent quality gains over unguided sampling, fixed guidance, and simple time-limited guidance, with learned schedules that adapt across prompts and timesteps.

**Strengths:**

It reframes classifier-free guidance tuning as a distribution-matching problem, replacing hand-crafted schedules with a learned, time- and prompt-aware policy—an original and practical angle.
The self-consistency objective is elegant and low-variance, making the method easy to implement on top of existing diffusion backbones without architectural changes.
Experiments span multiple datasets and backbones and include sensible ablations, showing consistent gains over unguided, fixed-guidance, and limited-interval baselines.
The learned guidance schedules are interpretable (varying with prompt and timestep), which improves clarity and aids real-world debugging.
Overall, the approach offers a broadly applicable and deployable improvement to conditional diffusion sampling with a favorable quality-to-complexity trade-off.

**Weaknesses:**

1. The paper argues that enforcing self-consistency is stronger and low-variance, but the conditions under which minimizing the proposed loss actually improves conditional sampling are not formalized.

2. On text-to-image, self-consistency plus CLIP reward improves FID yet struggles to surpass strong guided baselines on CLIP score.

3. Learning a guidance network and evaluating the self-consistency loss add additional overhead.

**Questions:**

1. How sensitive are results to the kernel/parameters in the MMD objective versus the simpler L2 variant?

2. What are the training and inference costs attributable to guidance learning and self-consistency evaluation?

3. Since the learned weights vary strongly with prompt, do certain semantic categories systematically demand higher guidance?

4. Can practitioners easily inspect how guidance evolves with time and prompt?

---

> ### Author Response · Authors · 2025-11-17
> **Rebuttal**
>
> We thank reviewer sh9D for their feedback. Please find below our detailed answer.
>
> > The paper argues that enforcing self-consistency is stronger and low-variance, but the conditions under which minimizing the proposed loss actually improves conditional sampling are not formalized.
>
> As for high-variance of marginal consistency, we meant that the variance of the estimates of the objective function from eq.15 and its gradient are high (since we will need to sample multiple (x0, c) and then additionally sample corresponding xs for lhs and rhs independently). We also tried learning guidance weights from this objective but it led to very unstable learning of guidance weights and did not lead to meaningful results.
>
> Self-consistency will have lower variance because we reuse the same (x0,c) for lhs and rhs. Self-consistency implies marginal consistency (since you can integrate over $(x_0, c)$). Marginal consistency condition was also considered in [1] where authors trained a distributional model to get a sample at time s from a sample at time t. Instead of directly optimizing the challenging marginal consistency objective (i.e., eq15), they designed an alternative approach which at convergence guarantees it. They reported strong empirical results which provided motivation for marginal consistency conditions.
>
> Note that marginal consistency condition would be satisfied by the true backwards process, which requires integrating over the intractable posterior distribution x_0 | x_t, c.
>
> Notwithstanding our stronger requirement of self-consistency, we do observe empirically that using self-consistency leads to guidance weights which outperform unguided baseline, as well as a baseline which is guided but with manually selected guidance weights.
>
> [1] Inductive Moment Matching, Linqi Zhou, Stefano Ermon, Jiaming Song, 2025
>
> > On text-to-image, self-consistency plus CLIP reward improves FID yet struggles to surpass strong guided baselines on CLIP score.
>
> Indeed, our approach performs on par with a manually selected guided baseline, leading to similar CLIP score and lower FID. The main advantage comes from including additional CLIP score loss into the objective function. In future work, we will focus on improving performance of our method by considering alternative reward functions which could be added to the objective.
>
> > Learning a guidance network and evaluating the self-consistency loss add additional overhead.
>
> The main computation overhead is to train a guidance network. However, the training cost of this is much lower than training a diffusion model. At inference, since the guidance network is much smaller than a diffusion model, the additional computation overhead is negligible. Please also see our common answer for more details.
>
> > How sensitive are results to the kernel/parameters in the MMD objective versus the simpler L2 variant?
>
> In our paper, we provide in Figure 2, sensitivity analysis of self-consistency and the L2 variant to the hyperparameters such as $\delta$ and $S_{\min}$. We see that a simpler L2 variant is more sensitive to hyperparameters compared to self-consistency.
>
> We added ablations over $\beta$ and number of particles $m$ in the common answer to all the reviewers. Overall, we see that any $m>=4$ works well and the performance is not very sensitive to $m$. As for $\beta$, we see that values  $\beta \in [1. 1.75]$ lead to good results.
>
> > What are the training and inference costs attributable to guidance learning and self-consistency evaluation?
>
> Please see our common answer to all the reviewers.
>
> > Since the learned weights vary strongly with prompt, do certain semantic categories systematically demand higher guidance?
>
> Please refer to our common answer for a more detailed analysis of guidance weights with respect to the prompt.
>
> > Can practitioners easily inspect how guidance evolves with time and prompt?
> Yes, we can visualize the behavior of the guidance network during sampling by plotting its predictions given the conditioning and time inputs. We have provided these visualizations in Figures 1 and 3
>
> We hope the above addresses the reviewer’s concerns - if we have done so, we would be grateful if the reviewer might consider increasing their score.  We are happy to address any further questions.

---

### Official Review · Reviewer_6vA2 · 2025-11-01

**Soundness:** 3
**Presentation:** 3
**Contribution:** 4
**Rating:** 6
**Confidence:** 3

**Summary:**

This paper proposes a method to learn the classifier-free guidance (CFG) weight by setting it as the output of a neural network. While previous CFG approaches relied on heuristics or trial-and-error to find an optimal (often large) weight, which could lead to saturation and degraded sample quality, this work optimizes guidance weights as a continuous function of time and condition. The objective is to minimize the distributional mismatch between the generated guided samples and the noised true conditional samples, using a novel self-consistency loss. This loss successfully mitigates the high-variance problem associated with theoretical approaches, leading to stable training and demonstrated performance gains across various datasets, from toy examples to COCO T2I sampling.

**Strengths:**

- The idea of learning the optimal guidance weight continuously via a neural network, rather than relying on grid search for optimal guidance intervals or empirical adjustments, is highly novel.
- Despite the inherent risk of the guidance weight collapsing to a trivial solution, the successful stabilization of learning via the newly proposed self-consistency loss is a significant achievement. The simplified L2 objective (Eq. 21) also appears surprisingly straightforward to implement.

**Weaknesses:**

The proposed method involves a large number of hyperparameters to determine the optimal guidance weight, and additionally requires performing $m$ rounds of noising and comparison at every denoising step, which introduces significant computational overhead. Given this, I wonder whether the computational cost difference compared to existing guidance distillation methods is actually substantial. It would be valuable to include a cost comparison table or discussion.

Moreover, since the loss function in this paper formalizes the matching between the true sample distribution and the guided one, I suspect that a parameter-efficient fine-tuning (PEFT) approach such as LoRA could optimize a student model directly using the same loss formulation, without needing a separate guidance network at inference time. This would result in a more inference-efficient model, and I’m curious whether the authors have considered or experimented with this alternative.

Regarding the design of the guidance network, the paper focuses heavily on the loss formulation for learning guidance weights but provides limited discussion or justification for the network’s architecture. The rationale behind its input-output design is unclear. Based on Table 1 and Fig. 1, 3, the conditioning input seems to have a stronger influence than expected, suggesting that incorporating the currently sampled image as an additional input could further improve weight estimation. Were such variants or conditioning combinations explored?

In the COCO experiments, the guidance network uses the text encoder output as a conditioning signal, and Fig. 3 shows that the learned guidance weights vary across prompts. Does this imply that the network has learned semantic understanding of the text embeddings? If so, how heavy must the model be to capture such semantics effectively? Additionally, from the qualitative results on COCO, the outputs often resemble those with weak or no guidance, which may indicate that the diffusion model’s original objective still struggles to fully capture the conditional distribution, similar to previous findings.

Finally, in the loss formulation, the authors mention using $m=4$ particles for computing the MMD in the image domain. Even if self-consistency reduces variance, I question whether the MMD loss computed from only four samples can be sufficiently accurate. Was this number chosen primarily to reduce computational cost? If so, it would be helpful to report how performance varies with different particle counts, to justify the design choice.

**Questions:**

- How does this method compare to guidance Interval approaches? The guidance interval method empirically finds an optimal guidance schedule, assigning near-zero guidance weight in early steps and large weights in the final few steps. Since the proposed method can measure how well the estimated distribution matches the true one via its loss, it may be possible to evaluate the optimality of the interval-based scheduling using the same loss metric. If so, this could open new directions for systematic guidance interval optimization beyond the work of LIG.
- Once the optimal guidance weights are obtained through this method, could they be further distilled into a single model to achieve both high performance and inference efficiency?

---

> ### Author Response · Authors · 2025-11-17
> **Rebuttal part 1**
>
> We thank Reviewer 6vA2 for their feedback. Please find our detailed answer below.
>
> > The proposed method involves a large number of hyperparameters to determine the optimal guidance weight, and additionally requires performing  rounds of noising and comparison at every denoising step, which introduces significant computational overhead. Given this, I wonder whether the computational cost difference compared to existing guidance distillation methods is actually substantial. It would be valuable to include a cost comparison table or discussion.
>
> We acknowledge the need for training a guidance network and tuning hyperparameters. However, this cost is only incurred once. Given that the network is lightweight and training is brief (50K steps), the total resource consumption is insignificant compared to the base diffusion model. Please see our common response for more details.
>
> The guidance network is a small MLP, and the training procedure only requires evaluating a pre-trained diffusion model on some data points. Moreover, as we show in our paper (see Table 1 and Table 2), using our procedure leads to overall better performance in image generation and is comparable to baseline performance on T2I benchmarks. The main hyperparameters are $\delta$ and $S_{\min}$ which control the time distribution $p(s,t)$. We observe that using $\delta~ 0.1/0.2$ overall works better, and the performance is not very sensitive to $S_{\min}$. See also Figure 2 and Section 5.1.
>
> In image space experiments, we also compare constant guidance and limited interval guidance (LIG) which were extensively tuned. Tuning LIG requires selecting both the interval and the guidance weight, which requires a large number of possible hyperparameters. As we show, this actually leads to worse performance than our method. Please also see our common response to all the reviewers where we added comparisons to clamp-linear schedule and LIG on T2I.
>
> In future work, we will focus on further improving the performance of our method such that the gap between baselines becomes larger, especially in text-to-image. Following your concerns, we will add a few sentences highlighting that our approach requires training a guidance network while the baseline approaches require only to sweep over corresponding hyperparameters.
>
> > ...loss function in this paper formalizes the matching between the true sample distribution and the guided one, I suspect that a parameter-efficient fine-tuning (PEFT) approach such as LoRA could optimize a student model directly using the same loss formulation...
>
> This is an excellent suggestion. While we focused on a lightweight auxiliary network to keep the base model frozen (preserving its original distinct capabilities), formulating this as a LoRA adaptation is a viable alternative implementation for future work. However, the methodological contribution of the self-consistency loss remains the same regardless of the specific implementation.
>
> > ...the paper focuses heavily on the loss formulation for learning guidance weights but provides limited discussion or justification...
>
> For our experiments, we mainly used a simple architecture for guidance networks, with a motivation that this should be an additional light-weight module. In Appendix E, we provide more details on what exact architecture was used for different experiments. The main rationale behind input-output design of our guidance network is the ability to take conditioning and time as input. Since classifier-free guidance (CFG) is conditioning-dependent, adding this information to guidance weights seems natural. Moreover, while fixed conditioning schedules were explored in the previous works (see, for example [1] which was suggested by Reviewer h7dP), our experiments suggest (see Table 1 and Table 2), that adding conditioning information leads to better results. Finally, above in the common answer we provide ablation over different types of conditionings for guidance networks. Overall, we see that adding more conditioning helps.
>
> _Variation with the size of network and its architecture_:
> We ran two ablations to explore design choices of MLP. First, we increased the size of the Dense layers in the MLP from 128 to 1024. This larger MLP has 12M parameters and we refer to this MLP as MLP-Wide. Second, we also introduced residual connections in the above MLP. We refer to this MLP  architecture as Residual MLP-Wide. We summarize the results below. We do not observe significant improvements in metrics upon increasing the number of parameters.
>
> | MLP | Dense layer size | Network size | FID 10K | CLIP 10K|
> | -- | -- | -- | -- | -- |
> | MLP-Small | 128 | 6M | 28.37 | 0.306 |
> | MLP-Wide | 1024 | 12M | 28.41 | 0.306 |
> | Residual MLP-Wide | 1024 | 12M | 28.68 | 0.306 |

---

> > ### Author Response · Authors · 2025-11-17
> > **Rebuttal Part 2**
> >
> > > ...the conditioning input seems to have a stronger influence than expected, suggesting that incorporating the currently sampled image as an additional input could further improve weight estimation. Were such variants or conditioning combinations explored?
> >
> > This is an interesting suggestion. We thought about such a variant, but this would make the guidance network much bigger. We will explore such variants in future work.
> >
> > > In the COCO experiments, the guidance network uses the text encoder output as a conditioning signal, and Fig. 3 shows that the learned guidance weights vary across prompts. Does this imply that the network has learned semantic understanding of the text embeddings? If so, how heavy must the model be to capture such semantics effectively?
> >
> > It is difficult to assess whether the network has learned semantic understanding of the text embeddings as we do not have a good evaluation technique. However, our network already receives CLIP and T5 embeddings, which come from pretrained models. These pretrained models already capture a degree of semantic understanding of the text.
> >
> > We have run an ablation on what kind of conditioning we could provide to the guidance network, see above in the common answer. We see that adding more information, i.e., both CLIP and T5 embeddings, leads to better performance.
> >
> > > Additionally, from the qualitative results on COCO, the outputs often resemble those with weak or no guidance, which may indicate that the diffusion model’s original objective still struggles to fully capture the conditional distribution, similar to previous findings.
> >
> > It is very common for T2I models to have weak performance without classifier-free guidance or with weak guidance. We only observe such outputs for our method when we do not include CLIP rewards in the training. All images in the 4th row in Figures 4-7, use both self-consistency and CLIP reward loss. CLIP reward directly encourages the alignment between the image and the conditioning. We see that our method can take advantage of such rewards.
> >
> > >  ...I question whether the MMD loss computed from only four samples can be sufficiently accurate. Was this number chosen primarily to reduce computational cost? If so, it would be helpful to report how performance varies with different particle counts, to justify the design choice....
> >
> > Our method draws inspiration from [2-3], where authors used MMD-based loss to train a distributional diffusion model on images. They also used $m=4$, number of particles. Please note that we use $m=4$ particles for every element of a batch (with a batch size n) which means that for every gradient step we use an $n * m$ amount of data.
> >
> > Moreover, we ran an ablation over $m$, see our common answer for more details. We see that performance is similar for $m >= 4$. Since increasing $m$ also increases computational complexity – $O(n * m^2)$, we tended to use a small $m$. We will include these ablations in the paper.
> >
> > > How does this method compare to guidance Interval approaches...
> >
> > In Table 1 and Table 2, we compare performance of our approach to Limited Interval Guidance (LIG). In Figure 1, we visualize the learned guidance weights of our approach. We see that it bears some similarities to limited interval guidance, but the shape is more complicated. We also include additional comparisons to LIG on T2I in the common response to all the reviewers, see above.
> >
> > > Since the proposed method can measure how well the estimated distribution matches the true one via its loss, it may be possible to evaluate the optimality of the interval-based scheduling using the same loss metric. If so, this could open new directions for systematic guidance interval optimization beyond the work of LIG.
> >
> > This is an interesting suggestion, we will explore this in the future work.
> >
> > > Once the optimal guidance weights are obtained through this method, could they be further distilled into a single model to achieve both high performance and inference efficiency?
> >
> > Yes, this is possible but we have not explored this direction yet. We thank you for the suggestion and we will explore it in future work.
> >
> > We hope the above addresses the reviewer’s concerns - if we have done so, we would be grateful if the reviewer might consider increasing their score.  We are happy to address any further questions.
> >
> >
> > **References**:
> >
> > [1] Analysis of Classifier-Free Guidance Weight Schedulers, Xi Wang, Nicolas Dufour, Nefeli Andreou, Marie-Paule Cani, Victoria Fernandez Abrevaya, David Picard, Vicky Kalogeiton, TMLR 2024
> >
> > [2] Distributional Diffusion Models with Scoring Rules, Valentin De Bortoli, Alexandre Galashov, J. Swaroop Guntupalli, Guangyao Zhou, Kevin Murphy, Arthur Gretton, Arnaud Doucet, ICML 2025
> >
> > [3] Inductive Moment Matching, Linqi Zhou, Stefano Ermon, Jiaming Song, ICML 2025

---

### Author Response · Authors · 2025-11-17
**Common answer part 1**

We thank all the reviewers for their feedback. In this section, we provide a common answer for all the reviewers which we will refer to. It contains additional ablations and experiments reviewers have asked for.

## Training costs and inference costs of guidance network

Training the guidance network involves training a lightweight MLP with a few layers. The cost of training it is insignificant compared to the cost of training a diffusion model. For example, in ImageNet experiments, we train a U-Net with ~ 260M parameters for around 7M iterations with a batch size of 128, while the guidance network with ~ 600K parameters is trained for up to 100K with batch size 256. In text-to-image, the model has about ~1B parameters and is trained for 1.6M steps with batch size of 2048, while guidance network has about 6M parameters which we train for 50K iterations with batch size 256.

At inference, we use a guidance network to produce a scalar $\omega$ which involves evaluating the MLP on precomputed $(t,s)$ and conditioning embeddings $c$. The computation cost is minimal compared to unrolling the diffusion model.

We have added a section in Appendix H.1 of the revised version of the paper about it.

## Ablation over $\beta$ and $m$ in the MMD computation

We ran an ablation study over $\beta$, the exponent applied to the $L_2$ norm, and $m$, the  number of particles used for the MMD computation for the self-consistency loss. We did this ablation on ImageNet.

The results are given in the following tables.

Take-aways: We see that the performance is not very sensitive to $m$, we see that $m>=4$ works well.  As for $\beta$, we see that values  $\beta \in [1. 1.75]$ lead to good results.

| $\beta$ | FID | IS |
| -- | -- | -- |
| 0.1 | 2.07 | 78.18 |
| 0.5 | 2.04 | 77.72 |
| 1.0 | 1.98 | 73.33 |
| 1.5 | 2.0 |  71.17 |
| 1.75 | 1.99 | 73.69 |

| m | FID | IS |
| -- | -- | -- |
| 2 | 2.00 | 70.66 |
| 4 | 1.99 | 73.69 |
| 8 | 1.99 | 73.02 |
| 16 | 2.00 | 77.11 |

We have added it to Appendix H.2 of the revised version of the paper.

## Comparison to Clamp-linear schedule and limited interval guidance (LIG)

We compare our method to the Clamp-linear schedule $w_t = max(c0,w_0 2t/T)$, see [1], and to Limited Interval Guidance (LIG) on T2I and ImageNet. For LIG on ImageNet, we take results from Table 1.

On ImageNet, for clamp-linear schedule, we swept over parameter $w_0$ and $c$ (see Appendix H.3 for more details).

We report the results in a table below. Our method is the same as the one reported in Table 1, i.e., self consistency with all the conditioning information. We observe that clamp-linear works worse than our method and baselines.

| Method | FID | IS |
| -- | -- | -- |
| Ours | 1.99 | 73.62 |
| LIG | 2.11 | 71.60 |
| Clamp-linear | 2.24 |  76.34 |
| Constant guidance | 2.4 | 66.72 |

We added this to Appendix H.3 in our paper.

On T2I, for a clamp-linear schedule, we swept over $w_0 \in \\{7.5,10.0,14.0,16.0\\}$. For $w_0 = 14.0$ and over clamp parameter $c_0 \in \\{1.0,2.0,4.0\\}$. The best parameters we found are $w_0 = 14, c_0=4$. For LIG, we swept over the guidance scales in $\{7.5, 15.0\}$ and intervals $[0.1,0.9]$, $[0.1,0.8]$, $[0.2,0.9]$, $[0.2,0.8]$. The best parameters are the guidance scale of $15.0$, interval $[0.1,0.9]$. We report performance of the best parameters.

The results are given below. We refer to our method as the one using self-consistency loss, clip reward and all conditioning information, see Table 3. We see that both LIG and clamp-linear underperform wrt our method and constant guidance in terms of CLIP score. We also provide detailed quantitative results for both clamp-linear schedule and LIG as well as qualitative results for these methods in the Appendix H.4 of our paper. Interestingly, we observe that in some cases, the guidance schedules predicted by our method resemble those of clamp-linear as shown in Figure 11 and Figure 15 in our paper.

| Method | CLIP | FID |
| -- | -- | -- |
| Ours | 0.306 | 28.37 |
| LIG | 0.304 | 24.95 |
| Clamp-linear | 0.305 | 27.14|
| Constant guidance | 0.306 | 31.2 |

[1] Analysis of Classifier-Free Guidance Weight Schedulers Xi Wang, Nicolas Dufour, Nefeli Andreou, Marie-Paule Cani, Victoria Fernandez Abrevaya, David Picard, Vicky Kalogeiton, TMLR, 2024

## Ablation over conditioning information in T2I experiments

We ablated the choice of conditioning embeddings for the MLP in our T2I experiments, specifically comparing T5-only and CLIP-only embeddings against our default method which uses both. In all the experiments we also use time conditioning. We found that while T5-only conditioning yields a slightly lower CLIP score than the other two configurations, it achieves slightly better FID. See table below. We also report qualitative results in the Appendix H.5 of the paper.

| Conditioning | FID | CLIP |
| -- | -- | -- |
| T5 | 28.28 | 0.305 |
| CLIP | 28.63 | 0.306|
| T5 and CLIP | 28.37 | 0.306|

---

> ### Author Response · Authors · 2025-11-17
> **Common answer part 2**
>
> ## Analysis of guidance weights on T2I and Imagenet
>
> We analysed learned guidance weights on ImageNet and T2I benchmarks. We have included these results in Appendix H.6 of the paper.
>
> For the analysis of guidance weights, we evaluate guidance network $\omega^\phi_{c, (t-dt,t)}$ on all 10k prompts from COCO or 1000 classes from ImageNet, using $T=128$ denoising steps for COCO, and $T=100$ steps for ImageNet. We then compute average guidance weight per conditioning $\bar{\omega_{c}} =  \frac{1}{T} \sum_{t=1}^{T} \omega^\phi_{c, (t-dt,t)}$ and also a sum of guidance weights per conditioning, i.e. $\hat{\omega_{c}} = \sum_{t} \omega^\phi_{c, (t-dt,t)}$.
>
> We then select top, median and bottom K (for K=10) prompts using $\hat{\omega}_{c}$.
>
> ### T2I analysis of learned guidance weights
>
> We visualize $\omega^\phi_{c, (t-dt,t)}$ for these top, median and bottom K prompts, see the link https://drive.google.com/file/d/1k-W05EZUTTpMwiDTP4zvjLhcBIbvnNk0/view?usp=drive_link
> (which is a link towards a png image on anonymous google drive). Here, t = 0 indicates data and t = 1 indicates noise.
>
> The results indicate that the guidance weights behave quite differently for these 3 categories. For bottom K, we see that guidance weights are high close to the noise and low otherwise. For the median K guidance weights, we see that they are increasing as t approaches 0. For top K guidance weights, we see that they are generally high. This suggests that there is a very different behavior depending on the prompt.
>
> Moreover, in https://drive.google.com/file/d/1d_z7ioWAAbjdhIiCFm9xMwSsf_yQszke/view?usp=drive_link, we show mean and standard deviation of $\bar{\omega}_{c}$ across prompts for a given prompt length. Overall, it seems that prompt length does not affect the magnitude of guidance weights that much.
>
> Finally, in https://drive.google.com/file/d/1xbzqhsc-k0gNDECQT5cnC_z5a_SExwxq/view?usp=drive_link, we visualize the histogram $\bar{\omega}_{c}$ across all 10k prompts. We also a category of prompts, which we call “human”, which contain keywords such as 'person', 'human', 'man', 'woman', 'men', 'women', 'boy', 'girl', 'child', 'baby', 'kid', 'guy'. Moreover, we select a category of prompts which we call “animal”, which contain keywords 'cat', 'dog', 'kitten', 'puppy', 'puppies'. We see that the distribution of human prompts category is similar to that of one on the full dataset, while the distribution of animal prompts category overall seems to have lower guidance.
>
> Below we show the prompts for top, median and bottom K categories. It is hard to see if there is a specific category of prompts which leads to high or low guidance, so we hypothesize that the discrepancy is due to how the original diffusion model was trained.
>
> **Top K**
> | k | Prompt |$\bar{\omega}_{c}$ |
> |--|--|--|
> |0| an old rusty refrigerator is abandoned in a field.  | 12.975 |
> |1| An old truck with several years of paint schemes sits parked as if artwork. | 12.936 |
> |2| A sky filled with lots of colorful flying kites. | 12.881 |
> |3| A pile of junk sitting next to a curb on green grass. | 12.756 |
> |4| Street art painted on the wall in an asian country.  | 12.733 |
> |5| OLD RUSTED BOXCARS STILL SITTING ON A SET OF TRACKS | 12.727 |
> |6| A field full of people flying kites in a cloudy blue sky. | 12.630 |
> |7| A crowded city filled with lots of people and traffic. | 12.607 |
> |8| a large clock tower towering above a city under a cloudy blue sky. | 12.544 |
> |9 | Some roll down doors  at a building painted with graffiti. | 12.494 |
>
> **Median K**
> | k|Prompt|$\bar{\omega}_{c}$ |
> |--|--|--|
> |0 |A group of men standing around a table of food in wooden room. | 6.477 |
> |1 |A room with a toilet, a door and shoes in it.  | 6.475 |
> |2 |A boy in black shirt doing a trick on a skateboard. | 6.474 |
> |3 |a man stands in front of a flipped skate boarder  | 6.474 |
> |4 |Oddly shaped homemade pizza about to be cut with pizza cutter | 6.474 |
> |5 |a silver toilet and some toilet paper and a green switch | 6.471 |
> |6 |Two cakes sitting on a class table near a candle. |6.471|
> |7 |A counter top in a kitchen topped with fruits and vegetables. |6.470|
> |8 |A park with many trees and benches at night.  | 6.470|
> |9 |A duck floating on a lake with gray and black feathers. | 6.470|
>
> **Bottom K**
> |k| Prompt|$\bar{\omega}_{c}$ |
> |--|--|--|
> |0 | A group of kids that are playing soccer on a field. |0.217|
> |1 | A group of men playing a game of soccer on a field. |0.243|
> |2 | A group of people playing a game of soccer on top of a field. |0.280|
> |3 | A group of men who are playing basketball together. |0.335|
> |4 | Small boys are playing a softball game on grass. |0.347|
> |5 | A group of men on a field playing baseball. |0.358|
> |6 | Brown horse grazing alone in an open grassy field.  |0.392|
> |7 | Men on a baseball diamond playing baseball, a man just swung |0.393|
> |8 | A group of soccer players scramble against each other to get the ball.  |0.413|
> |9 | A group of children playing soccer on a field. |0.415|

---

> > ### Author Response · Authors · 2025-11-17
> > **Common answer part 3**
> >
> > ### ImageNet analysis of learned guidance weights
> >
> > We visualize $\omega^\phi_{c, (t-dt,t)}$ for top, median and bottom K classes on ImageNet, see https://drive.google.com/file/d/1a7ydr5tZuKC_sdMVZtmmqDqKjhJ-HkIw/view?usp=drive_link.
> >
> > Unlike in T2I, we see that guidance weights seem to have two distinct behaviors – for bottom K, they are generally small, while for median and top K they are generally large. We also see that they look like a form of Limited Interval Guidance (LIG). The discrepancy to T2I could be attributed to a few things. First, in T2I, we add clip score reward to the loss which encourages prompt alignment. Second, in T2I, the diffusion model was trained on a vast amount of text-image pairs, such that each prompt receives an unequal number of images. In ImageNet, each class is seen the same amount of time.
> >
> > Below, we show the class labels and their corresponding $\bar{\omega}_{c}$ for bottom, median and top K.
> >
> > **Bottom K**
> > | k | Class id | Class name | $\bar{\omega}_{c}$|
> > |--|--|--|--|
> > | 0 | 640 | manhole_cover | 0.000 |
> > | 1 | 138 | bustard | 0.000 |
> > | 2 | 83 | prairie_chicken | 0.000 |
> > | 3 | 353 | gazelle | 0.009 |
> > | 4 | 974 | geyser | 0.012 |
> > | 5 | 44 | alligator_lizard | 0.012 |
> > | 6 | 106 | wombat | 0.020 |
> > | 7 | 352 | impala | 0.021 |
> > | 8 | 276 | hyena | 0.024 |
> > | 9 | 209 | Chesapeake_Bay_retriever | 0.026 |
> >
> > **Median K**
> > | k | Class id | Class name | $\bar{\omega}_{c}$ |
> > | -- | -- | -- | -- |
> > | 0 | 345 | ox | 0.445 |
> > | 1 | 735 | poncho | 0.445 |
> > | 2 | 765 | rocking_chair | 0.444 |
> > | 3 | 607 | jack-o'-lantern | 0.444 |
> > | 4 | 51 | triceratops | 0.447 |
> > | 5 | 660 | mobile_home | 0.447 |
> > | 6 | 565 | freight_car | 0.444 |
> > | 7 | 108 | sea_anemone | 0.447 |
> > | 8 | 200 | Tibetan_terrier | 0.443 |
> > | 9 | 255 | Leonberg | 0.443 |
> >
> > **Top K**
> > | k | Class id | Class name | mean $\omega$ |
> > | -- | -- | -- | -- |
> > | 0 | 768 | rugby_ball | 0.727 |
> > | 1 | 650 | microphone | 0.735 |
> > | 2 | 788 | shoe_shop | 0.735 |
> > | 3 | 886 | vending_machine | 0.738 |
> > | 4 | 560 | football_helmet | 0.743 |
> > | 5 | 513 | cornet | 0.745 |
> > | 6 | 592 | hard_disc | 0.748 |
> > | 7 | 541 | drum | 0.751 |
> > | 8 | 613 | joystick | 0.752 |
> > | 9 | 681 | notebook | 0.777 |

---

### Meta-Review · Area_Chair_2D3v · 2026-01-06

**Summary:**

This paper received two borderline accept (6) and two borderline reject (4) scores. I believe that the majority of the reviewers’ concerns have been adequately addressed in the rebuttal. One remaining issue, raised by reviewer h7dP, is the lack of a fair comparison with existing pre-trained diffusion models. I agree that such a comparison would further strengthen the paper. As noted in the rebuttal, the authors have committed to including additional experimental results using publicly available pre-trained models, which I expect those addition in the final version.

Overall, after considering the rebuttal, the paper appears to offer useful insights to the community. As there are no strong objections from the reviewers that would clearly argue against acceptance, I am inclined to recommend acceptance.

**Reviewer Concerns:**

See the summary section.

**Reviewer Scores:**

See the summary section.

---

### Decision · Program_Chairs · 2026-01-26

Accept (Poster)